# Inhibition of lung microbiota-derived proapoptotic peptides ameliorates acute exacerbation of pulmonary fibrosis

Corina N. D'Alessandro-Gabazza[1,2,3,18], Taro Yasuma[1,4,18], Tetsu Kobayashi[5,18], Masaaki Toda[1,2,18], Ahmed M. Abdel-Hamid [3,6,18], Hajime Fujimoto[5], Osamu Hataji[7], Hiroki Nakahara[5], Atsuro Takeshita[1,4], Kota Nishihama[4], Tomohito Okano[5], Haruko Saiki[5], Yuko Okano[1,4], Atsushi Tomaru[5], Valeria Fridman D'Alessandro[1], Miyako Shiraishi [3], Akira Mizoguchi[8], Ryoichi Ono[9], Junpei Ohtsuka[9,10], Masayuki Fukumura[9,10], Tetsuya Nosaka [9], Xuenan Mi[11], Diwakar Shukla[11], Kensuke Kataoka[12], Yasuhiro Kondoh[12], Masaki Hirose[13], Toru Arai[13], Yoshikazu Inoue [13], Yutaka Yano[4], Roderick I. Mackie[3,14,15], Isaac Cann [3,14,15,16,17,19 ✉] & Esteban C. Gabazza [1,2,3,19 ✉]

Idiopathic pulmonary fibrosis is an incurable disease of unknown etiology. Acute exacerbation of idiopathic pulmonary fibrosis is associated with high mortality. Excessive apoptosis of lung epithelial cells occurs in pulmonary fibrosis acute exacerbation. We recently identified corisin, a proapoptotic peptide that triggers acute exacerbation of pulmonary fibrosis. Here, we provide insights into the mechanism underlying the processing and release of corisin. Furthermore, we demonstrate that an anticorisin monoclonal antibody ameliorates lung fibrosis by significantly inhibiting acute exacerbation in the human transforming growth factorβ1 model and acute lung injury in the bleomycin model. By investigating the impact of the anticorisin monoclonal antibody in a general model of acute lung injury, we further unravel the potential of corisin to impact such diseases. These results underscore the role of corisin in the pathogenesis of acute exacerbation of pulmonary fibrosis and acute lung injury and provide a novel approach to treating this incurable disease.

[1] Department of Immunology, Mie University Faculty and Graduate School of Medicine, Tsu, Mie, Japan. [2] Center for Intractable Diseases, Mie University, Tsu, Mie, Japan. [3] Carl R. Woese Institute for Genomic Biology (Microbiome Metabolic Engineering), University of Illinois at Urbana–Champaign, Urbana, IL, USA. [4] Department of Diabetes and Endocrinology, Mie University Faculty and Graduate School of Medicine, Tsu, Mie, Japan. [5] Department of Pulmonary and Critical Care Medicine, Mie University Faculty and Graduate School of Medicine, Tsu, Mie, Japan. [6] Department of Botany and Microbiology, Faculty of Science, Minia University, El-Minia, Egypt. [7] Respiratory Center, Matsusaka Municipal Hospital, Matsusaka, Mie, Japan. [8] Department of Neural Regeneration and Cell Communication, Mie University Graduate School of Medicine, Tsu, Mie, Japan. [9] Department of Microbiology and Molecular Genetics, Mie University Graduate School of Medicine, Tsu, Mie, Japan. [10] BioComo Incorporation, Komono, Mie, Japan. [11] Department of Chemical and Biomolecular Engineering, University of Illinois at Urbana-Champaign, Urbana, IL, USA. [12] Department of Respiratory Medicine and Allergy, Tosei General Hospital, Seto, Aichi, Japan. [13] Clinical Research Center, National Hospital Organization Kinki-Chuo Chest Medical Center, Sakai City, Osaka, Japan. [14] Department of Microbiology, The University of Illinois at Urbana–Champaign, Urbana, IL, USA. [15] Center for East Asian & Pacific Studies, the University of Illinois at Urbana–Champaign, Urbana, IL, USA. [16] Department of Animal Science, The University of Illinois at Urbana–Champaign, Urbana, IL, USA. [17] Division of Nutritional Sciences, The University of Illinois at Urbana–Champaign, Urbana, IL, USA. [18]These authors contributed equally: Corina N. D'Alessandro-Gabazza, Taro Yasuma, Tetsu Kobayashi, Masaaki Toda, Ahmed M. Abdel-Hamid. [19]These authors jointly supervised this work: Isaac Cann, Esteban C. Gabazza. ✉email: icann@illinois.edu; gabazza@doc.medic.mie-u.ac.jp

diopathic pulmonary fibrosis (IPF) is a chronic and incurable disease of unknown etiology. Patients with the disease have a life expectancy of only 2 to 3 years after diagnosis[1]. Recent epidemiological studies suggest that globally there are more than 5 million IPF patients and that the number of cases is further growing worldwide[2]. Recurrent injury and apoptosis of the alveolar epithelium and aberrant tissue repair caused by increased secretion of profibrotic factors (e.g., transforming growth factorβ1) and excessive recruitment of myofibroblasts and deposition of extracellular matrix in the lungs play central roles in the disease pathogenesis[1]. The process ultimately culminates in lung tissue scarring, lung structural destruction, and respiratory failure[2]. The natural history of IPF is unpredictable and variable[3]. Patients may have a slowly progressive disease, a rapidly progressive course, or a sudden clinical deterioration referred to as acute exacerbation (AE)[4,5]. The most frequent cause of death in IPF is AE occurring in ~46% of the cases[6,7]. AE predicts high mortality (of up to 50%) during the episode and a short-term survival after the acute event[6]. Patients live only 3 to 4 months after the diagnosis of AE[5,6]. The mortality increases by up to 90% in cases requiring mechanical ventilation[6]. Infections and diagnostic or surgical procedures can trigger AE, although the precise underlying mechanism is unknown[5]. No effective therapy is currently available for AE[8].

A growing amount of evidence suggests a causative role of the lung microbiome in IPF. Dysbiotic or abundant lung microbiome is associated with persistent altered expression of genes involved in host defense, decreased innate immune response, impaired lung epithelial integrity, abnormal fibroblast responsiveness, the severity of lung function abnormality, deterioration of chest radiograph findings, disease progression, and less survival in IPF patients[9–14]. Alteration and bacterial burden of the lung microbiome become even worse in IPF with AE[15,16]. However, the mechanistic pathways linking lung microbial dysbiosis with pulmonary fibrosis remain unclear. We recently identified a proapoptotic peptide termed corisin in the lung microbiome that may explain the pathogenic role of microbial alteration in pulmonary fibrosis[17]. Corisin is a peptide conserved in diverse staphylococci and strains of different pathogenic bacteria[17]. The sequence of corisin corresponds to a segment of bacterial transglycosylases[17]. Intrapulmonary administration of corisin or corisin-harboring bacterium causes apoptosis of alveolar epithelial cells and AE in mice with established pulmonary fibrosis[17]. Also, IPF patients with slowly progressive disease exhibit increased lung levels of corisin compared to a healthy population, and patients with AE show strikingly much-elevated lung corisin levels compared to stable patients[17]. Therefore, we hypothesized that treatment with a monoclonal antibody against corisin would directly inhibit or ameliorate the AE of pulmonary fibrosis (AE-IPF) by blocking the apoptotic activity of corisin.

In the present study, we demonstrate the presence and apoptotic activities of other corisin-like peptides from known pathogenic bacteria and therefore show a wider distribution of this toxic peptide. We then identify a cleavage activity in a putative serine protease secreted by corisin-containing Staphylococcus spp. to unravel the mechanism by which the transglycosylase housing the corisin peptide is activated to yield the proapoptotic agent and further demonstrate that corisin activates the intrinsic pathway of apoptosis in lung epithelial cells. We also develop and characterize several monoclonal antibodies to identify monoclonal antibody 21A with the capacity to neutralize the toxic activity of corisin and related peptides. By harnessing this knowledge, we test the efficacy of the neutralizing monoclonal antibody on two different lung fibrosis models and demonstrate that the anticorisin antibody ameliorates lung fibrosis by significantly inhibiting AE in the human transforming growth factorβ1 model and

acute lung injury in the bleomycin model. We further show the inhibitory activity of the anticorisin monoclonal antibody in a general acute lung injury model to provide the potential impact of corisin on acute lung injury-associated diseases. These results underscore the role of corisin and corisin-like peptides in the pathogenesis of AE in pulmonary fibrosis and perhaps other acute lung injury-associated diseases and provide a novel approach to treating this incurable disease.

## Results

**A broad group of pathogens carries proapoptotic corisin-like peptides.** We previously demonstrated that the culture supernatants from Staphylococcus nepalensis strain CNDG and a mixture of Staphylococcus spp. that grew in medium inoculated with lung fibrotic tissue specimens from mice with terminal stages of lung fibrosis induce apoptosis of lung alveolar epithelial cells[17]. Also, we reported that a transglycosylase fragment, termed corisin, released from Staphylococcus nepalensis, is the cause of the lung epithelial cell apoptosis and triggers AE of chronic pulmonary fibrosis in transgenic mice with established lung fibrotic disease[17]. The sequence of corisin or its derivatives is highly conserved in diverse members of the genus staphylococcus[17]. We hypothesized that, in addition to Staphylococcus nepalensis, other Staphylococcus species also release corisin-like peptides from transglycosylases to induce apoptosis of lung epithelial cells. To test this hypothesis, we performed streak plating of a mixture of Staphylococcus spp. grown in medium inoculated with fibrotic lung specimens to isolate individual bacterial colonies. Screening of twelve colonies indicated that three colonies designated strain 1, strain 7, and strain 12, secreted apoptotic factors, while in the remaining isolates, the apoptotic activity was not detected. These results suggested that the culture originally designated strain 6 in our earlier discovery was a mixture of pro- and non-apoptotic bacteria[17]. The whole-genome sequences revealed that the three apoptotic bacterial isolates were distinct strains of Staphylococcus haemolyticus. The complete genome sequences of the bacteria designated S. haemolyticus strain 1 (accession No: CP071512-CP071515), S. haemolyticus strain 7 (accession No: CP071508-CP071511) and S. haemolyticus strain 12 (accession number: CP071505-CP071507) have been deposited at the Genbank database (https://www.ncbi.nlm.nih.gov/genbank/). A search of the genomes revealed the presence of a corisin-like sequence in putative IsaA transglycosylases in strain 1 (transglycosylase_1bp_00353), strain 7 (transglycosylase_7bp_00350) and strain 12 (transglycosylase_12bp_00350) (Supplementary Fig. 1). The corisin-like peptide sequence was identical in the corresponding transglycosylase[17] from each strain (Supplementary Fig. 1).

We then cultured A549 alveolar epithelial cells in the presence of the bacterial culture supernatant from each bacterial strain to further evaluate their apoptotic characteristics. In addition to inducing apoptosis during culture with A549 alveolar epithelial cells (Fig. 1a, b), the culture supernatant from each Staphylococcus haemolyticus strain also triggered significantly increased cleavage of caspase-3 in the alveolar epithelial cells (Fig. 1c, d). Furthermore, transmission electron microscopy confirmed the presence of increased apoptotic cells after treatment with the culture supernatant of S. haemolyticus strain 12 (Fig. 1e–n). A synthetic peptide of the S. haemolyticus corisin-like sequence, which differs only at a single position compared to the reported sequence of S. nepalensis strain CNDG[17], prepared by a commercial manufacturer (ThermoFisher Scientific), recapitulated the proapoptotic effect of the supernatant of each strain (strain 1, 7, and 12) on A549 lung alveolar epithelial cells (Supplementary Fig. 2a–c). In addition, intratracheal instillation of Staphylococcus haemolyticus strain 12 resulted in a significant increase of lung neutrophil infiltration, pulmonary fibrosis score and lung area of epithelial cell apoptosis in TGFβ1 TG mice

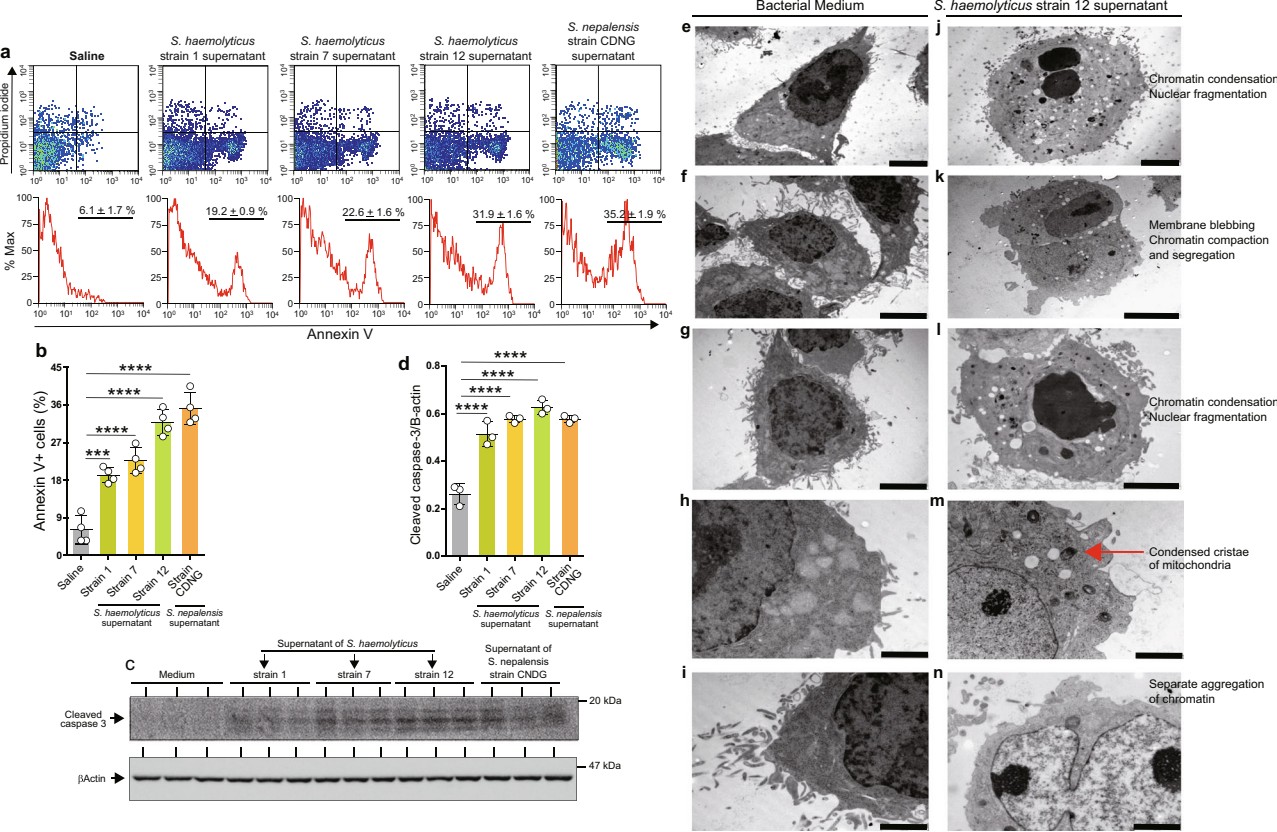

**Fig. 1 The culture supernatants from three strains of _S. haemolyticus_ induce apoptosis of alveolar epithelial cells. a** A549 alveolar epithelial cells were cultured in the presence of the (1/10 dilution) culture supernatants from strains 1, 7, and 12 of _S. haemolyticus_ for 48 h. A549 cells treated with the culture supernatant from _S. nepalensis_ strain CNDG were the positive controls and cells treated with saline were the negative controls. **b** The percentage of apoptotic cells was determined by flow cytometry and quantified. $N = 4$ in each group. Data are expressed as the mean ± S.D. Statistical analysis was performed using ANOVA with a post hoc Newman–Keuls test. ***$p < 0.001$; ****$p < 0.0001$. **c, d** Western blotting of A549 cells cultured in the presence of the culture supernatants of the three _S. haemolyticus_ strains and _S. nepalensis_. In (**d**) $n = 3$ in each group. Representative blots from two independent experiments with similar results are shown. Data are expressed as the mean ± S.D. Statistical analysis was performed using ANOVA with a post hoc Newman–Keuls test. ****$p < 0.0001$. **e–n** Transmission electron photo-micrograph of alveolar epithelial cell apoptosis induced by the culture supernatant from _Staphylococcus haemolyticus_ strain 12. A549 cells were cultured in the presence of the bacterial medium (uninoculated) as control and the culture supernatant from _S. haemolyticus_ strain 12 grown for 48 h, and the cells were evaluated by transmission electron microscopy as described under Methods. _S._, _Staphylococcus_. Scale bars are 5 μm in **e, f, g, j, l**, 10 μm in **k** and 2 μm in **h, i, m, n**. Representative images from two independent experiments with similar results are shown. The source data underlying (**b, d**) are provided in the Source Data file.

compared to intratracheal instillation of _Staphylococcus epidermidis_, which contains transglycosylases lacking the corisin sequence (Supplementary Fig. 3a–i).

_Staphylococcus haemolyticus_ is a commensal bacterium; however, it is also a well-known opportunistic and multidrug-resistant pathogen[18]. The bacterium grows optimally in the presence of oxygen and salty (10%NaCl) conditions[19]. The infections caused by _Staphylococcus haemolyticus_ in humans include sepsis, peritonitis, endocarditis, meningitis, and infections of wound, bone, and urinary tract[18,19]. Interestingly, the corisin-like sequence of the transglyco-sylase homolog from the strains of _S. haemolyticus_ isolated in the present report is identical to the sequence observed in transglyco-sylases from _Mycobacteroides abscessus subsp. abscessus_ (Genbank protein accession no. SKR69498.1) and a strain of _Listeria monocytogenes_ (Genbank protein accession no. ECO1693478.1) (Supplementary Fig. 4). _Listeria monocytogenes_ is the most virulent foodborne pathogen[20] and _Mycobacteroides abscessus subsp. abscessus_[21] is a multidrug-resistant non-tuberculous mycobacterium that commonly causes chronic lung infections. Homologous transglycosylases from other strains of _Listeria monocytogenes_ and _Mycobacteroides abscessus subsp. abscessus_ and a hypothetical protein from _Weissella confusa_, an opportunistic pathogen[22], also contain

corisin-like sequences with proapoptotic activity (Supplementary Fig. 2a–c). These observations suggest that a broad group of pathogens carry derivatives of this toxic peptide, a likely component of their arsenal of pathogenicity.

**Activation of intracellular apoptotic pathways by corisin and bacterial culture supernatant.** Apoptosis occurs by activation of the extrinsic or intrinsic pathway[23]. The extrinsic pathway is mediated by membrane-anchored death receptors that relay intracellular death signals through their cytoplasmic death domains after ligand binding, leading to cleavage of procaspase-8 to caspase-8 (Fig. 2a)[23]. The intrinsic pathway is regulated by the mitochondrion. The proapoptotic stimulus causes mitochondrial perturbation that increases mitochondrial membrane perme-ability leading to the release of suppressors of baculoviral inhi-bitors of apoptosis repeat-containing (BIRC) proteins and proapoptotic factors, which contribute to apoptosome formation to cleave procaspase-9 to caspase-9[24,25]. The cleaved forms of caspase-8 and caspase-9 activate caspase-3, the executor of apoptosis (Fig. 2a). We found that the synthetic corisin peptide (Fig. 2b, c) and the culture supernatant of _S. haemolyticus_ strain 12 (Supplementary Fig. 5a, b) significantly increased the number

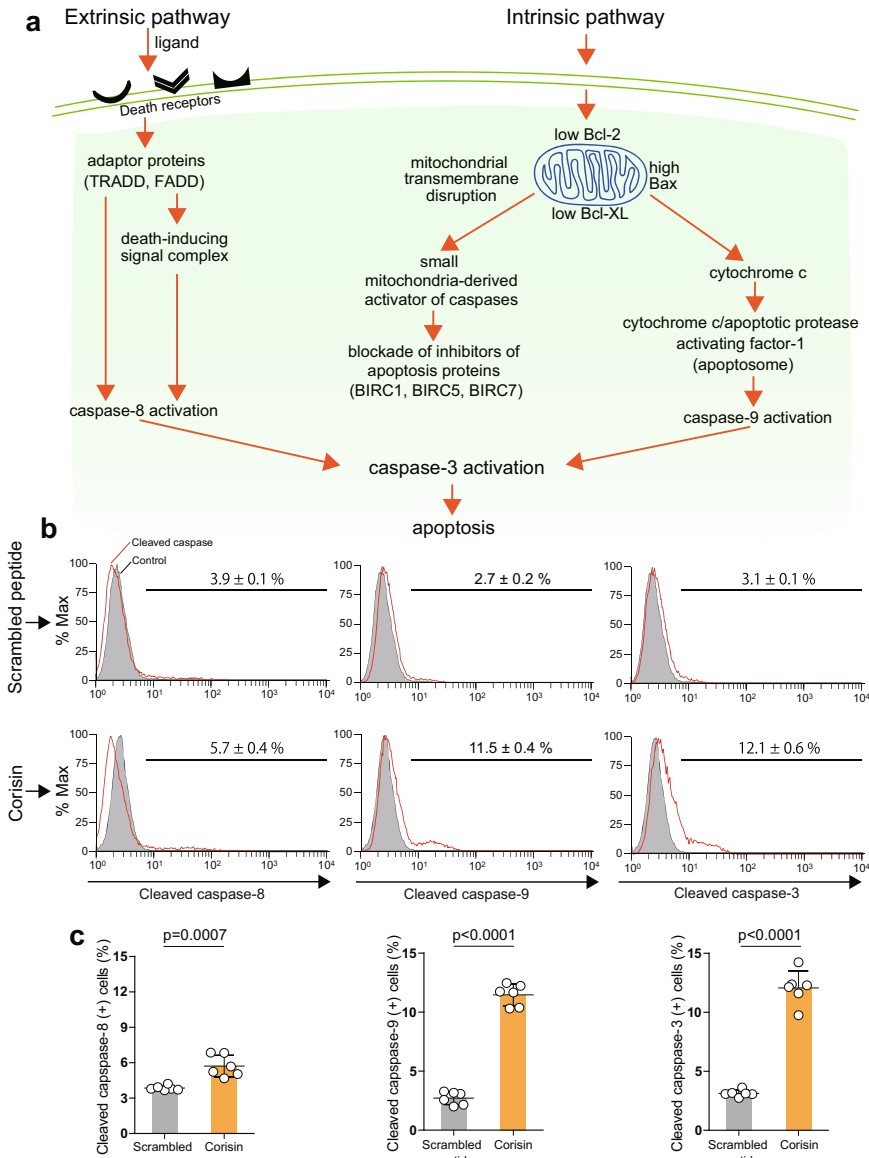

**Fig. 2 Activation of cleaved caspases by corisin. a** Apoptosis occurs by the extrinsic and intrinsic pathways. The extrinsic pathway is initiated by binding of ligands to membrane-bound death receptors that results in activation of intracellular signaling and cleavage of procaspase-8 to caspase-8. The intrinsic pathway is regulated by the mitochondrion. In the presence of the stimulus, perturbation of the mitochondrial membrane increases its permeability and causes the release of suppressors of baculoviral inhibitors of apoptosis repeat-containing (BIRC) proteins and proapoptotic factors, which contribute to apoptosome formation to cleave procaspase-9 to caspase-9. Both caspase-8 and caspase-9 can activate caspase-3, the effector of apoptosis. **b**, **c** A549 alveolar epithelial cells were cultured in the presence of 5 μM corisin or scrambled peptide for 48 h. The percentage of cells positive for cleaved caspase-8, cleaved caspase-9 and cleaved caspase-3 was determined by flow cytometry and quantified. $N = 6$ in each group. Data are expressed as the mean ± S.D. Statistical analysis was performed using two-sided unpaired $t$ test. The source data underlying (**c**) are provided in the Source Data file.

of alveolar epithelial cells with activated caspase-9 and activated caspase-3 compared to control stimulation, as shown by cytometry analysis. Western blotting showed that the culture supernatants of the different strains of *S. haemolyticus* significantly induced activation of caspase-9 in alveolar epithelial cells (Supplementary Fig. 5c, d).

Consistent with findings in other types of cells, we also found a significantly increased reactive oxygen species accumulation in alveolar epithelial cells treated with corisin compared to scrambled peptide-treated cells, and loss of mitochondrial membrane integrity during apoptosis induced by corisin (Supplementary Fig. 6a–d)[26]. In addition, corisin significantly reduced the mRNA expression of the antiapoptotic factors BIRC5, BIRC7, Bcl2, Bcl-xL, cyclin D1, proliferating cell nuclear antigen, and significantly increased the

mRNA expression of proapoptotic factors including caspase-3 and apoptotic protease activating factor-1 (APAF-1) compared to the scrambled peptide (Supplementary Fig. 7). Screening of phosphorylated proteins by Western blotting disclosed an increment in p53 S392 phosphorylation after synthetic corisin stimulation compared to control (Supplementary Fig. 8a, b). These results suggest that the microbes-derived peptides preferentially activate the mitochondrion-mediated intrinsic pathway of apoptosis.

**A bacterial secreted factor degrades a corisin-containing transglycosylase to induce apoptosis of lung epithelial cells.** We previously reported that the apoptotic peptide corisin is likely cleaved from an *S. nepalensis* transglycosylase to induce AE of

pulmonary fibrosis[17]. The factor involved in the cleavage is unknown. Here, we hypothesized that corisin-shedding bacteria secrete a protease that cleaves the apoptotic peptides from their respective transglycosylase (Supplementary Fig. 9a). To interrogate this hypothesis, we first partially purified a protease fraction from the culture supernatant of *S. nepalensis* strain CNDG. We successively concentrated the culture supernatant of *S. nepalensis* using >50 kDa-filter and then concentrated the flowthrough (<50 kDa) fraction using >10 kDa-filter (Supplementary Fig. 9b). We loaded the concentrated fraction onto a Sephracryl S-300 column and then measured the proteolytic activity of collected fractions on a corisin-containing recombinant transglycosylase after resolving the products by gel electrophoresis (Supplementary Fig. 9c). Fractions with high proteolytic activity were pooled and concentrated before loading for a second time onto a Sephracryl S-300 column to separate fractions with high transglycosylase degrading activity (Supplementary Fig. 9d).

We then incubated the recombinant transglycosylase in the presence of varying concentrations of the protease fraction for 2 h or incubated the recombinant transglycosylase in the presence of a definite concentration of the protease fraction over increasing time periods before resolving the products by gel electrophoresis. Silver staining of the gel showed a dose- and time-dependent degradation of the recombinant transglycosylase (Fig. 3a, b). Importantly, in a separate experiment, we showed that the culture supernatant of *S. haemolyticus* also cleaves recombinant transglycosylase (Supplementary Fig. 10). We then cultured A549 alveolar epithelial cells to subconfluency and treated the cells with the degradation products prepared by incubating the recombinant transglycosylase in the culture supernatant from *S. nepalensis* for 2 h. Flow cytometry analysis performed after 48 h of cell culture revealed significantly increased apoptosis of alveolar epithelial cells cultured in the presence of the degradation products compared to controls (Fig. 3c, d). These results suggest that a bacterial secreted factor, likely a protease, cleaves the transglycosylase, yielding the corisin peptide to induce apoptosis of lung epithelial cells.

**The bacterial secreted factor that cleaves transglycosylase is a putative serine protease**. To determine whether the transglycosylase cleaving factor is a protease, we added varying doses of the serine protease inhibitors pefabloc SC or diisopropyl fluorophosphate (DFP), the cysteine protease inhibitor E-64, or the chelating agent ethylenediaminetetraacetic acid (EDTA) to a reaction mixture containing culture supernatant from *S. nepalensis* and its recombinant transglycosylase and incubated at 37 °C for different time intervals. After resolving the products on a protein gel through electrophoresis and silver staining, we found that the serine protease inhibitors pefabloc SC and DFP suppress transglycosylase degradation (Supplementary Fig. 11a–d). However, neither E-64 nor EDTA suppressed transglycosylase degradation (Supplementary Fig. 11e–h). Overall, these findings suggest that a putative serine protease secreted by *S. nepalensis* degrades its corisin-containing transglycosylase into smaller peptides, likely including corisin.

**An anticorisin neutralizing monoclonal antibody inhibits lung cell apoptosis induced by the transglycosylase degradation products**. Based on our findings above, we hypothesized that an anticorisin neutralizing antibody would block the proapoptotic activity of the transglycosylase degradation products derived from the proteolytic activity of the putative bacterial serine protease. To test our hypothesis, we first developed and characterized anticorisin monoclonal antibodies. The hybridoma clones 2A, 3Aa, 4A, 9A, and 21A producing respectively the immunoglobulins (Ig) IgG2bκ, IgMκ, IgG2aκ, IgG2aκ, and IgG2bκ were established

(Supplementary Fig. 12a, b). Western blotting performed using increasing amounts of recombinant transglycosylase revealed that all monoclonal antibody clones recognize and bind to the corisin-containing transglycosylase from *S. nepalensis* (Supplementary Fig. 12c). We also observed that the monoclonal antibody (mAtb) produced by each hybridoma clone differentially binds to a corisin-coated surface at different dilution levels. In particular, clones 9A and 21A showed strong binding activity (Supplementary Fig. 12d).

To identify the binding site of the anticorisin mAtbs in the corisin molecule, we performed epitope mapping. To this end, the binding of each mAtb to a microarray of small peptides with different lengths of corisin sequences was evaluated. We used hemagglutinin peptides and anti-hemagglutinin antibody as controls to validate the overall peptide microarray integrity and assay quality. The staining of the hemagglutinin peptides with the control antibody alone showed a well-defined spot pattern in green color with no background interaction at any scanning intensity level (Supplementary Fig. 13a, b). Incubation of the corisin peptide microarray with each anticorisin mAtb showed a moderate and obvious antibody response against epitope-like spot patterns in red color containing the consensus motif PESSGNP for 9A, 21A, and 2A mAtb clones (Supplementary Fig. 13c–h), and the consensus motif NPAGY for 4 A mAtb clone with high signal-to-noise ratios (Supplementary Fig. 13i, j; https://zenodo.org/record/5803063#.YcWY-WDP2ck).

We then evaluated whether monoclonal antibodies with the high binding ability to corisin (2A, 9A, and 21A) can inhibit the proapoptotic activity of corisin on cultured A549 alveolar epithelial cells. Flow cytometry analysis revealed that only the clone 21A mAtb significantly neutralizes corisin-induced apoptosis of alveolar epithelial cells in vitro (Supplementary Fig. 14a, b). The antiapoptotic activity of the anticorisin mAtb clone 21A was confirmed by immunostaining and DNA laddering assay (Supplementary Fig. 14c–e). Furthermore, the clone 21A neutralizing mAtb also significantly inhibits the apoptotic activity of the culture supernatants from both *S. nepalensis* and *S. haemolyticus* on A549 alveolar epithelial cells (Supplementary Fig. 15a, b). Similar to the results observed with the A549 cells, the corisin-like peptide from *S. haemolyticus* induces apoptosis in normal human bronchial epithelial (NHBE) cells (Supplementary Fig. 16a, b), and the 21A mAtb blocks the apoptotic activity of this bacterium's culture supernatant on these cells (Supplementary Fig. 16c, d).

In a subsequent experiment, we used the clone 21A mAtb to evaluate whether the anticorisin mAtb can inhibit the proapoptotic activity of the degradation products of the *S. nepalensis* corisin-containing transglycosylase. Flow cytometry analysis showed that the anticorisin mAtb clone 21A significantly blocks the apoptotic activity of the transglycosylase degradation products on lung alveolar epithelial cells compared to the control rat IgG (Supplementary Fig. 17a, b). It is of note, however, that gel electrophoresis and silver staining revealed that the anticorisin neutralizing mAtb clone 21A could not inhibit the proteolytic degradation of the recombinant transglycosylase (Supplementary Fig. 17c). Overall, these observations suggest that secretion of a putative serine protease by fibrotic lung tissue-associated *Staphylococcus* spp. effectuates transglycosylase proteolysis, and that the proteolysis of the transglycosylase releases corisin to elicit apoptosis of lung alveolar epithelial cells, and that binding of the anticorisin neutralizing mAtb to the transglycosylase does not impede transglycosylase degradation.

**Significant biological activity of native corisin in bacterial culture supernatant and BALF from IPF patients**. We then evaluated the concentration of native corisin that is biologically

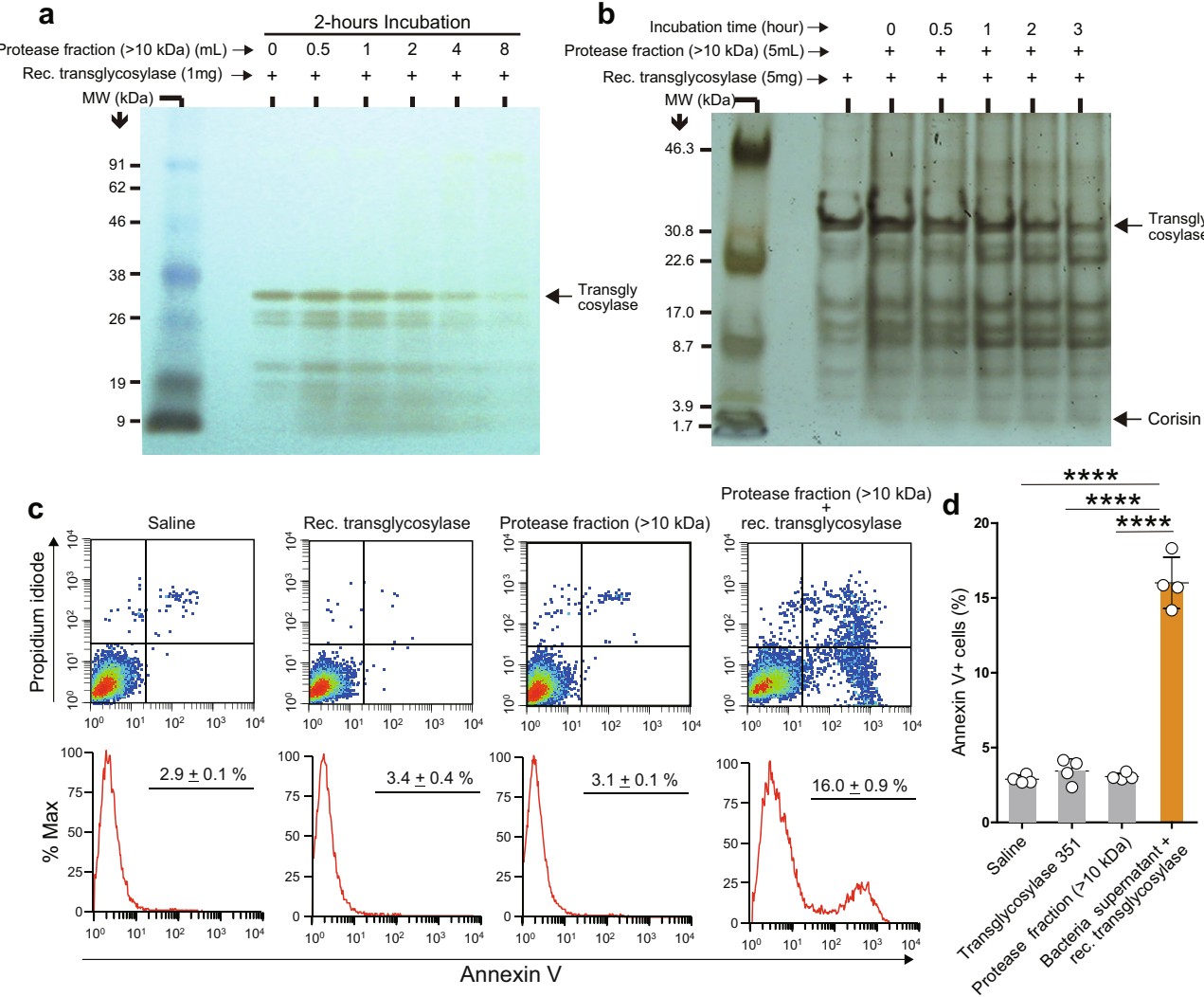

**Fig. 3 A bacterial secreted product degrades transglycosylase to induce apoptosis of lung epithelial cells. a** A sodium dodecyl sulfate-polyacrylamide gel stained with silver staining after completing the electrophoresis of a reaction mixture containing digestion buffer, recombinant transglycosylase, and varying doses of the protease fraction (>10 kDa) prepared from *Staphylococcus nepalensis* culture supernatant and incubated at 37 °C for 2 h. A representative image from two independent experiments with similar results is shown. **b** A sodium dodecyl sulfate-polyacrylamide gel stained with silver staining after completing the electrophoresis of a reaction mixture containing digestion buffer, recombinant transglycosylase, and a dose of the protease fraction (>10 kDa) prepared from *Staphylococcus nepalensis* culture supernatant incubated at 37 °C at different time intervals. A representative image from two independent experiments with similar results is shown. **c, d** Flow cytometry analysis to evaluate apoptosis of A549 alveolar epithelial cells after treatment with the indicated reaction mixture. $N = 4$ in each group. Data are the mean ± S.D. Statistical analysis by ANOVA with a post hoc Newman–Keuls test. ****$p < 0.0001$. The source data underlying (**d**) are provided in the Source Data file.

active in the bacterial supernatant with an ELISA system prepared using rabbit antitransglycosylase polyclonal antibody and biotinylated anticorisin mAtb clone 9A. We first confirmed the apoptotic activity of the bacterial supernatant. Bacterial supernatant from *S. nepalensis* was added to the medium of A549 alveolar epithelial cells at 1:10 dilution in the presence or absence of the neutralizing anticorisin A21 mAtb (20 μg/ml). The A549 cells were cultured for 48 h, and then apoptosis was evaluated by flow cytometry. Apoptosis induced by the diluted bacterial supernatant was significantly and almost completely inhibited by anticorisin mAtb (Fig. 4a, b; Supplementary Fig. 15b). Immunoassay of corisin showed that the mean concentration of native corisin in undiluted bacterial supernatants from *S. nepalensis* was 1042 ± 174.6 pg/ml and from *S. haemolyticus* was 802.4 ± 57.7 pg/ml (Fig. 4c). As the bacterial supernatant was used at 1:10 dilution in the apoptosis assay, we estimate that native corisin has strong biological activity at a concentration of ≥100 pg/ml.

We also conducted a similar experiment to evaluate the proapoptotic activity of native corisin present in BALF from IPF patients (Supplementary Table 1). The BALF from IPF patients has been shown to cause apoptosis of lung epithelial cells[27], and we have also demonstrated that corisin levels are increased in BALF from IPF patients with AE compared to patients with stable disease[17]. BALF collected from IPF patients with AE ($n = 14$), and healthy subjects ($n = 5$) was added to the medium of A549 alveolar epithelial cells at 1:2 dilution in the presence of anticorisin mAtb clone 21A or control IgG (20 μg/ml), and apoptosis was assessed after 48 h. Apoptosis of A549 cells in the presence of BALF from IPF patients was significantly inhibited by the anticorisin mAtb (Fig. 4d, e). The mean concentration of native corisin in undiluted BALF from IPF patients ($n = 14$) was 650.9 ± 58.3 pg/ml and from healthy subjects ($n = 5$) was 420.1 ± 10.7 pg/ml, suggesting that native corisin also has strong biological activity at low concentrations in BALF from IPF

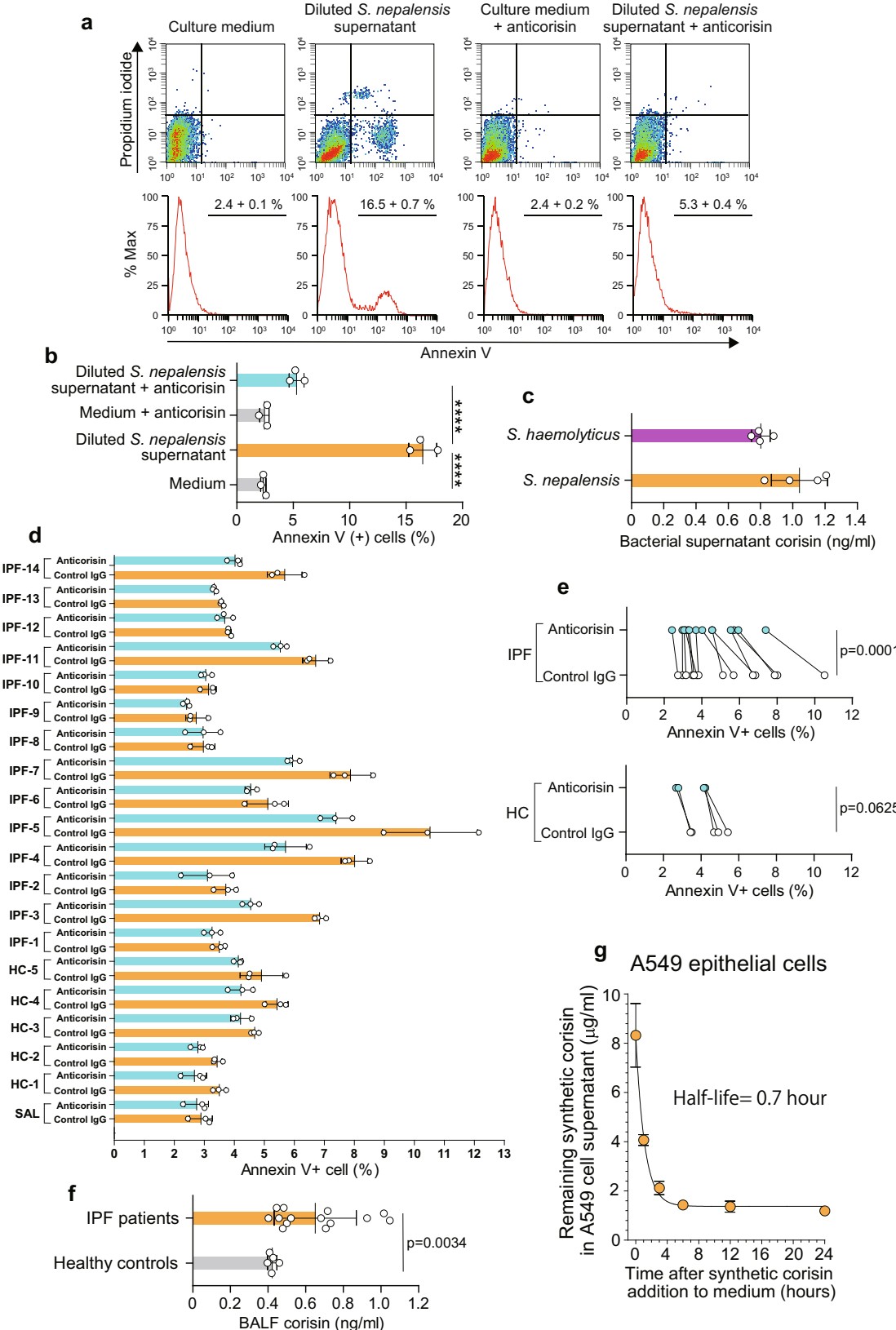

patients with AE (Fig. 4f). However, it is worth noting that the half-life of (synthetic) corisin in solution is <1 h in a human alveolar cell culture system (Fig. 4g). Therefore, the concentration of native corisin in vivo during an AE event may differ depending on the sample collection time.

**Dose of synthetic corisin for in vivo experiments**. Unlike native corisin, a high concentration of synthetic corisin (10 µg/ml or 5 µM) was required to induce significant apoptosis in A549 cells (current study Supplementary Fig. 14b)[17]. While native corisin peptides probably conserve their folded structure after cleavage

**Fig. 4 Significant biological activity of native corisin in bacterial culture supernatant and BALF from IPF patients. a, b** A549 alveolar epithelial cells were cultured in the presence of culture diluted supernatants (1:10 dilution) from *Staphylococcus* (*S.*) *nepalensis* (*n* = 3) in the presence or absence of anticorisin neutralizing mAtb clone 21A (20 μg/ml) for 48 h, and apoptosis was evaluated by flow cytometry. Bars indicate the mean ± S.D. Statistical analysis by ANOVA with a post hoc Newman–Keuls test. ****$p$ < 0.0001. **c** Concentration of native corisin in undiluted culture supernatant from *S. nepalensis* and *S. haemolyticus*. $N$ = 4 in each group examined over two independent experiments. Bars indicate the mean ± S.D. **d** A549 alveolar epithelial cells were cultured in the presence of diluted bronchoalveolar lavage fluid (BALF) (1:2 dilution) from healthy controls (HC, *n* = 5) and idiopathic pulmonary fibrosis (IPF, *n* = 14) patients with acute exacerbation and the anticorisin neutralizing mAtb clone 21A or control IgG for 48 h, and apoptosis was evaluated by flow cytometry. Bars indicate the mean ± S.D. Three replicates performed for the study of each patient. **e** The mean percentage of apoptotic (Annexin V+) cells induced by diluted BALF from all healthy controls (*n* = 5) and IPF patients (*n* = 14) was compared between anticorisin mAtb and control IgG. Statistical analysis by two-sided Wilcoxon signed rank test. **f** Concentration of native corisin in undiluted BALF from healthy subjects (*n* = 5) and IPF patients (*n* = 14). Bars indicate the mean ± S.D. Statistical analysis by two-sided Mann–Whitney *U* test. **g** A549 cells were cultured up to subconfluency, synthetic corisin was added to cell culture medium at a final concentration of 10 μg/ml and medium samples were collected after 0, 1, 3, 6, 12, and 24 h to measure corisin levels and calculate the in vitro half-life of corisin in a cell system. $N$ = 4 in each time point examined over two independent experiments. The half-life of synthetic corisin was calculated using an exponential decay equation model available in the GraphPad Prism version 7. Bars indicate the mean ± S.D. The source data underlying (**b**, **c**, **d**, **e**, **f**, **g**) are provided in the Source Data file.

from transglycosylase, most synthetic corisin peptides are probably unfolded, and therefore high concentrations of synthetic corisin are required to induce biological activity[28,29]. We used the biologically active in vitro concentration (5 μM) of the synthetic peptide as a reference to estimate the approximate dose of synthetic corisin required to induce AE in our disease mouse models using a molarity equation and a molarity calculator software as described under Methods. The estimated dose was about 240 μg, and in a preliminary study using varying intranasal doses of synthetic corisin (100, 150, and 300 μg), we found that a dose of 300 μg induces significant lung radiological changes. Therefore, we used 300 μg of synthetic corisin to induce AE in our subsequent in vivo experiments.

**An anticorisin neutralizing antibody inhibits corisin-induced acute exacerbation (AE) of pulmonary fibrosis in TGFβ1 TG mice.** We previously demonstrated that corisin induces AE of pulmonary fibrosis in TGFβ1 TG mice[17]. Based on this finding, we sought to determine whether the anticorisin neutralizing antibody would suppress the AE of the fibrotic disease. We first assessed changes in the circulating levels of the anticorisin mAtb clone 21A after intraperitoneal administration and calculated its half-life. The average half-life of the anticorisin neutralizing mAtb 21A in plasma was 3.8 ± 2.5 days (Supplementary Fig. 18). We then allocated TGFβ1 TG mice with lung fibrosis into two groups with matched CT scores (Supplementary Fig. 19a, b). A group of TGFβ1 TG mice with no fibrosis receiving intratracheal corisin was used as controls.

One group with lung fibrosis received intraperitoneal injections of anticorisin mAtb clone 21A, and the other group with lung fibrosis received intraperitoneal injections of an irrelevant control rat IgG once a day every 3 days during 2 weeks before intratracheal instillation of synthetic corisin to induce AE (Supplementary Fig. 19a). We performed CT before and after corisin or saline intratracheal instillation for comparison. Control TGFβ1 TG mice with no fibrosis treated with anticorisin mAtb showed no changes on treatment with corisin (Supplementary Fig. 19d, e). TGFβ1 TG mice, with fibrosis, receiving corisin and treated with control IgG had deterioration of CT findings (Supplementary Fig. 19f, g), whereas counterparts treated with anticorisin mAtb had significant amelioration of CT score of lung fibrosis (Supplementary Fig. 19h, i).

We compared the cell count in bronchoalveolar lavage fluid (BALF) between the treatment groups. TGFβ1 TG mice treated with anticorisin mAtb had significantly decreased total number of all cells and the total number of lymphocytes in BALF compared to controls and mouse counterparts treated with control IgG (Fig. 5a, b). The markers of lung injury, including the plasma

levels of surfactant protein-D (SP-D), MUC5B, matrix metalloproteinase-1 (MMP-1), and the BALF levels of SP-D, MUC5B, MUC-1, were also significantly decreased in mice treated with the anticorisin mAtb compared to controls and mice treated with control antibody (Fig. 5c). The Ashcroft fibrosis score and the collagen (trichrome) stained area were decreased in mice treated with anticorisin mAtb compared to the control group, although the decrease was not statistically significant. The lung hydroxyproline content, a generally used marker of collagen tissue deposition, was significantly lower in mice treated with anticorisin mAtb than in mice treated with control IgG. The lung hydroxyproline content was significantly associated with the number of lung inflammatory cells (Supplementary Fig. 20a–f). These observations point to corisin as a potential therapeutic target for AE in pulmonary fibrosis.

**Corisin induces AE of bleomycin-induced pulmonary fibrosis.** Lung fibrosis induced by bleomycin (BLM) in mice or rats is the most characterized and commonly used preclinical model for IPF studies[30]. However, whether corisin can exacerbate the disease in BLM-induced lung fibrosis is unknown. To address this question, we developed the model in WT mice by infusing BLM through subcutaneous osmotic mini-pumps during 7 days and administered corisin by intratracheal instillation on day 20 during the fibrotic phase of BLM-induced lung injury (Supplementary Fig. 21a)[31]. We first confirmed lung fibrosis development by chest CT before allocating the mice into two groups with a matched CT score using in-house CT fibrosis score criteria (Supplementary Fig. 21b, c). We then intratracheally administered corisin or scrambled peptide. CT was performed before euthanizing the mice a day after the corisin intratracheal instillation. The controls were WT mice receiving an infusion of saline through osmotic mini-pumps and subsequently receiving intratracheal instillation of corisin or scrambled peptide. The CT fibrosis score remained unchanged in mice without lung fibrosis receiving either intratracheal scrambled peptide or corisin. Although the CT score remained unchanged in mice with lung fibrosis receiving intratracheal instillation of a scrambled peptide, the CT score of mice with lung fibrosis significantly worsened after receiving intratracheal corisin (Supplementary Fig. 21d–k).

Mice with BLM-induced lung fibrosis receiving intratracheal corisin showed a significantly increased BALF total number of lymphocytes and enhanced BALF levels of MUC-1, periostin, osteopontin, and collagen I compared to counterpart mice receiving intratracheal scrambled peptide (Supplementary Fig. 22a–c). In addition, the cleavage of caspase-3 was significantly increased in mice receiving intratracheal corisin compared to mouse counterparts receiving scrambled peptide (Supplementary Fig. 22d, e). The

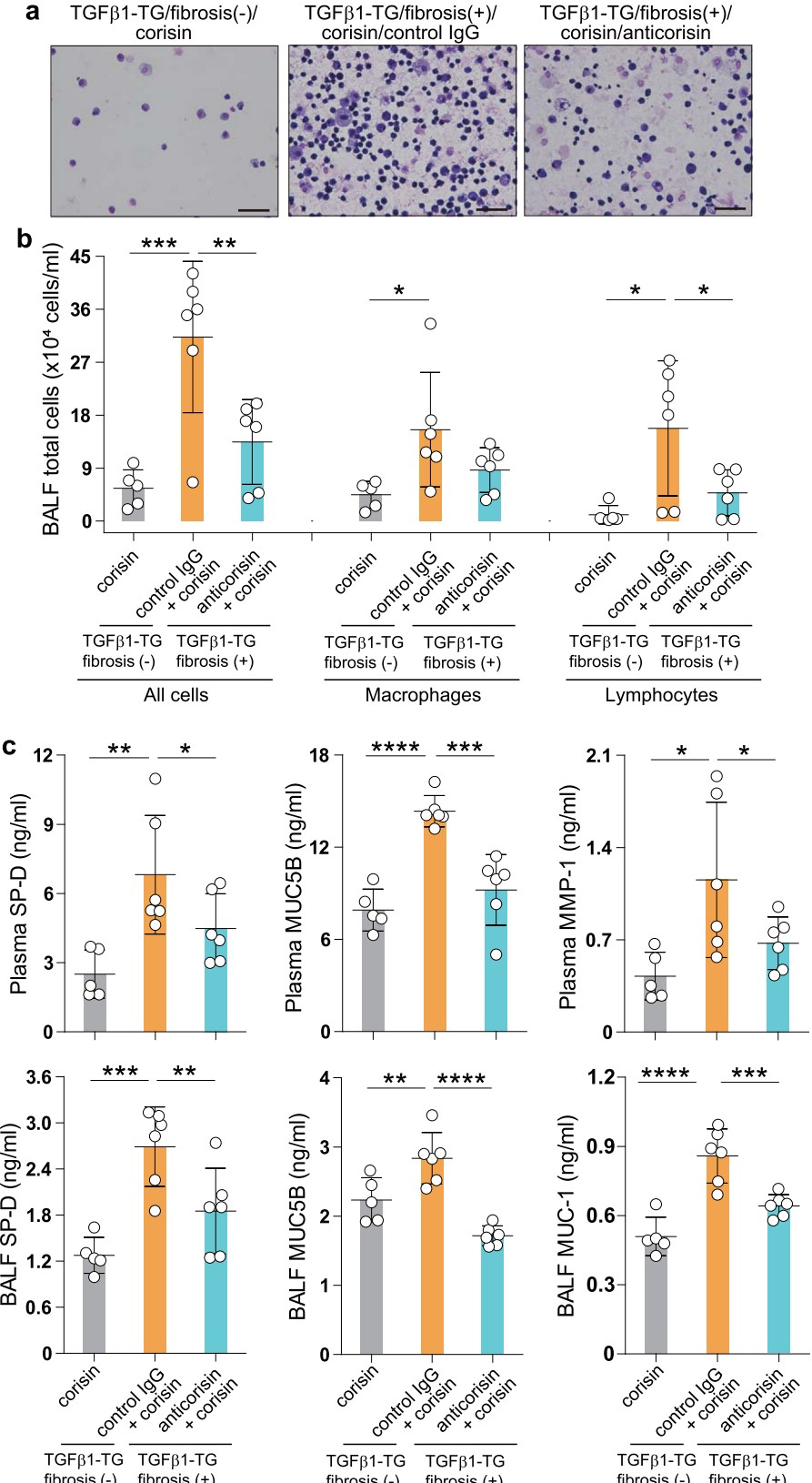

Ashcroft fibrosis score, the lung collagen deposition assessed by Trichrome stained area, and the lung hydroxyproline content were also significantly enhanced in mice receiving corisin compared to mice treated with the scrambled peptide (Supplementary Fig. 23a–e).

Mice receiving saline through osmotic mini-pumps and treated with corisin or scrambled peptide showed no significant changes. These findings showed that corisin can also exacerbate the disease in BLM-induced lung fibrosis.

**Fig. 5 Monoclonal anticorisin antibody inhibits acute exacerbation of pulmonary fibrosis in TGFβ1 TG mice.** TGFβ1 transgenic (TG) mice were randomly allocated into two groups with a matched grade of lung fibrosis and one group without lung fibrosis. A group of TGFβ1 TG mice ($n = 6$) with lung fibrosis received an intraperitoneal injection of anticorisin monoclonal antibody (mAtb) and another group ($n = 6$) with lung fibrosis received control IgG five times every 2 days before intratracheal instillation of corisin. The group of TGFβ1 TG mice without fibrosis ($n = 5$) received only intratracheal corisin. After euthanasia by an overdose of anesthesia, bronchoalveolar lavage fluid (BALF) was drawn from mice of each group, the BALF fluid was centrifuged and the pellet was used to evaluate the total cell count and differential cells. **a** Giemsa staining for differential cell count. Scale bars indicate 50 μm. **b** Count of total number of cells and neutrophils in BALF (total lymphocytes, macrophages and all cells). Data are the mean ± S.D. Statistical analysis was performed by ANOVA with a post hoc Neuman–Keuls test. *$p < 0.05$; **$p < 0.01$; ***$p < 0.001$. **c** The levels of surfactant protein-D (SP-D), MUC5B protein, matrix metalloproteinase-1 (MMP-1), and MUC-1 were measured by commercially available enzyme immunoassay kits. Data are the mean ± S.D. Statistical analysis was performed by ANOVA with a post hoc Neuman–Keuls test. *$p < 0.05$; **$p < 0.01$; ***$p < 0.001$; ****$p < 0.0001$. TGFβ1 transforming growth factor β1. The source data underlying (**b**, **c**) are provided in the Source Data file.

**Mice receiving intranasal corisin at early stages of BLM-induced lung injury develop advanced pulmonary fibrosis.** Mice received BLM infusion through osmotic mini-pump for 7 days to induce lung injury/fibrosis, and treated with intranasal corisin (100 μg) or scrambled peptide (100 μg) on days 3, 4, 5, 7, 9, 10, and 11 and euthanized on day 22 after starting BLM infusion. Mice with lung fibrosis receiving intratracheal corisin showed significantly increased Ashcroft score, collagen (trichrome) stained area, and lung hydroxyproline content compared to mice receiving scrambled peptide (Supplementary Fig. 24a–f). No difference was observed in Ashcroft score, trichrome stained area, or hydroxyproline content between mice without lung fibrosis receiving corisin or scrambled peptide.

**Longitudinal changes of circulating native corisin during BLM-induced pulmonary fibrosis.** Mice were infused BLM ($n = 5$) or saline ($n = 4$) through osmotic mini-pumps for 7 days, and blood samples were collected from the mouse tail vein on days 0, 3, 6, 11, 14, 18, 21, and 25 to measure the concentration of corisin. The plasma corisin levels gradually increased from day 3 to reach significant levels on days 14 and 18, suggesting that circulating corisin increases during the acute phase and remains high during the chronic phase of lung injury (Supplementary Fig. 25a, b).

**Anticorisin mAtb suppresses BLM-induced pulmonary fibrosis.** As corisin exacerbates the BLM-induced pulmonary fibrosis, we hypothesized that the deadly peptide plays a role in this experimental model. To test this hypothesis, we infused BLM in mice through subcutaneous osmotic mini-pumps for 7 days and treated them with the anticorisin neutralizing mAtb or control IgG by intraperitoneal route three times a week for 3 weeks during the acute and chronic phase of BLM-induced lung injury before sacrifice on day 22 (Supplementary Fig. 26a). Mice receiving saline (SAL) through osmotic mini-pumps were the control mice prepared to rule out the secondary effects. Evaluation of inflammatory cells in BALF showed that mice with BLM-induced lung fibrosis treated with anticorisin mAtb have a significantly low number of all cells, total lymphocytes, and neutrophils compared to mice treated with control IgG (Supplementary Fig. 26b, c).

Mice with BLM-induced lung fibrosis treated with anticorisin mAtb showed low plasma and BALF levels of osteopontin and MUC-1, reduced BALF MUC5B level, decreased number of apoptotic lung cells, and reduced cleavage of caspase-3 (Fig. 6a±e). We performed a chest CT on day 21 before sacrificing mice to evaluate CT fibrosis score. Mice with BLM-induced lung fibrosis treated with anticorisin mAtb showed significantly reduced CT fibrosis scores compared to their counterpart mice treated with the control Atb. There were no radiological changes in mice receiving SAL infusion and then treated with anticorisin mAtb or control IgG. In addition, the Ashcroft fibrosis scores, lung collagen deposition assessed by trichrome stained area, and the lung hydroxyproline content decreased compared to counterpart

mice treated with control IgG (Fig. 7a–g). Overall, these findings suggest that corisin also plays a role in the pathogenesis of BLM-induced pulmonary fibrosis.

**Treating the acute phase of BLM-induced lung injury with the anticorisin mAtb ameliorates pulmonary fibrosis.** In general, lung injury induced by BLM administered subcutaneously through osmotic mini-pumps is characterized by an acute phase of lung inflammation that peaks on day 12, followed by lung fibrosis in the chronic phase[31]. To determine whether the anticorisin mAtb ameliorates lung fibrosis by inhibiting lung injury induced by BLM in the early acute phase, we treated mice with anticorisin neutralizing mAtb or control IgG in the acute phase (days 2, 4, 6, 9, and 11) after BLM mini-pump infusion. We then stopped the treatment until sacrifice on day 22 (Supplementary Fig. 27a). Mice treated with anticorisin mAtb revealed a significant reduction in the BALF number of lymphocytes during the chronic phase of the disease (day 22 after starting BLM infusion) compared to mice receiving control IgG. The plasma levels of SP-C, SP-D, periostin, osteopontin, the BALF levels of MUC-1, total TGFβ1 and the cleavage of caspase-3 on day 22 were significantly lower in mice treated with anticorisin mAtb than in mice treated with control IgG (Supplementary Fig. 27b–f). The plasma levels of MUC-1 and collagen I also decreased in mice treated with anticorisin mAtb compared to mice treated with control IgG, although the reduction was not statistically significant.

In addition, mice treated with anticorisin mAtb showed significant amelioration of the CT fibrosis score and significantly reduced Ashcroft fibrosis score and hydroxyproline content in the lungs compared to mice treated with control IgG. The collagen (trichrome) stained area was also reduced in mice treated with anticorisin mAtb, but the reduction was not statistically significant (Supplementary Fig. 28a–g). These observations suggest that corisin-associated acute lung injury during the acute phase is an important determining factor of lung fibrosis in the disease's chronic phase.

**Anticorisin mAtb inhibits apoptosis during the acute phase of BLM-induced lung injury.** We administered anticorisin mAtb or control IgG by intraperitoneal route to mice with acute lung injury induced by BLM delivered once by intratracheal instillation and compared the grade of apoptosis. Mice treated with the anticorisin mAtb showed a significant reduction in apoptosis of lung epithelial cells compared to mice treated with control IgG (Supplementary Fig. 29a–c).

**Anticorisin mAtb prolongs the survival of mice with AE of pulmonary fibrosis.** Human TGFβ1 transgenic mice with a CT score-matched lung fibrosis (Supplementary Fig. 30a, b) received BLM or SAL through osmotic mini-pumps and then treated with anticorisin mAtb or irrelevant IgG three times a week, and the mouse

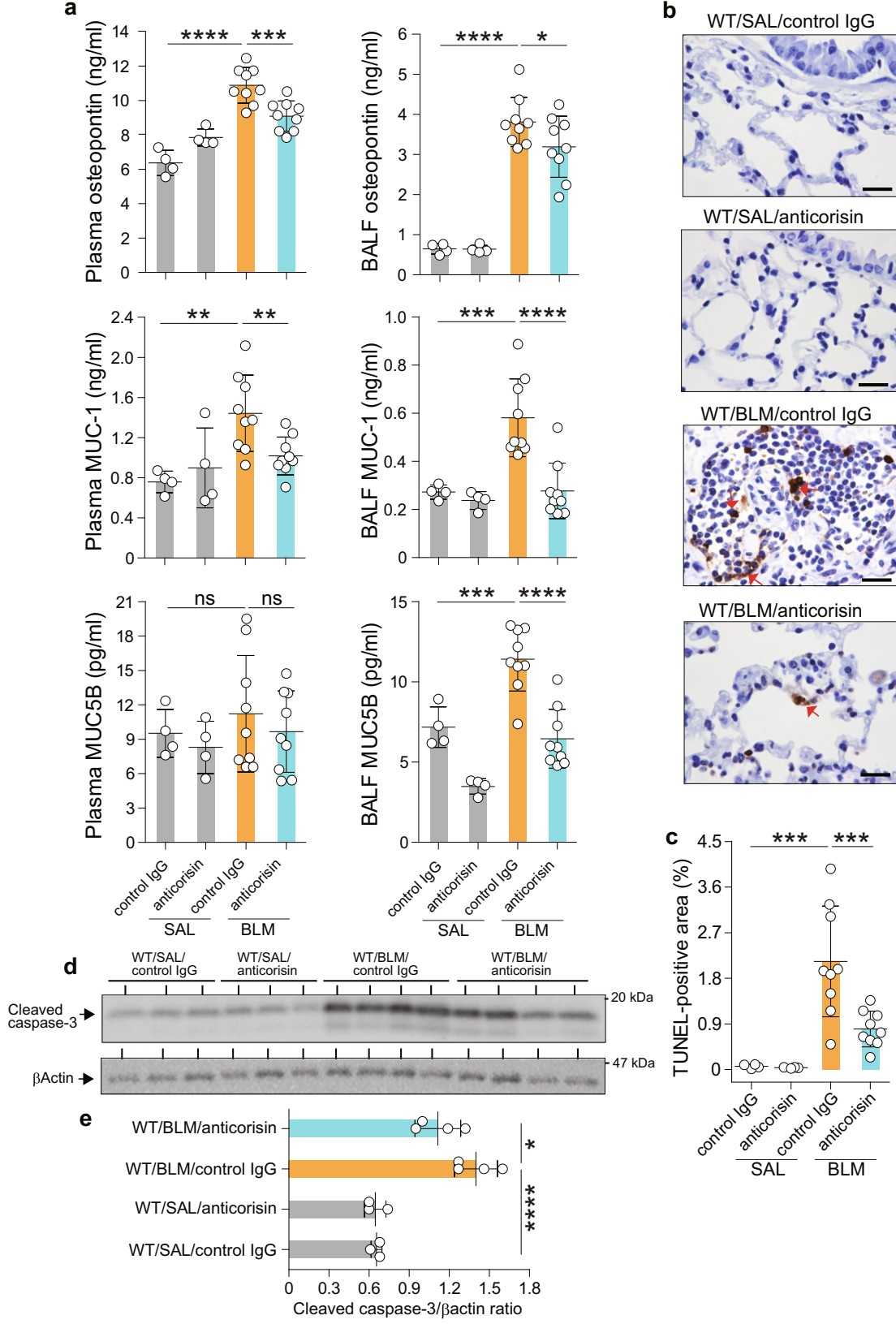

survival was followed. The TGFβ1 TG mice with lung fibrosis and BLM-induced AE treated with the anticorisin mAtb showed significantly longer survival than those treated with the control IgG (Supplementary Fig. 30c). As expected, the control groups receiving saline through osmotic mini-pumps showed no changes in survival.

**Anticorisin mAtb suppresses lipopolysaccharide-induced acute lung injury**. Based on the preceding results, we decided to determine whether corisin may also impact the pathogenesis of acute lung injury caused by other agents such as lipopolysaccharide (LPS). We first treated the mice with 20 mg/kg of anticorisin mAtb or

**Fig. 6 Amelioration of acute tissue injury, and apoptosis in the lungs of mice with bleomicin-induced pulmonary fibrosis treated with anticorisin monoclonal antibody.** Wild-type (WT) mice received bleomycin (BLM) by osmotic mini-pumps and treated with anticorisin monoclonal antibody (mAtb) (WT/BLM/anticorisin) or control IgG (WT/BLM/control IgG) by intraperitoneal route three times a week for 3 weeks. WT mice receiving saline (SAL) by osmotic mini-pumps and treated with anticorisin mAtb (WT/SAL/anticorisin) or control IgG (WT/SAL/control IgG) by intraperitoneal route three times a week for 3 weeks were the control mice. **a** The levels of osteopontin, MUC-1 and MUC5B were measured by enzyme immunoassays using commercial kits. $N = 4$ in WT/SAL/control IgG and WT/SAL/anticorisin groups and $n = 9$ in WT/BLM/control IgG and WT/BLM/anticorisin groups. Bars indicate the mean ± S.D. Statistical analysis by ANOVA with a post hoc Newman–Keuls test. *$p < 0.05$; **$p < 0.01$; ***$p < 0.001$; ****$p < 0.0001$. ns not significant. **b, c** DNA fragmentation was evaluated by staining through terminal deoxynucleotidyl transferase dUTP Nick-End Labeling (TUNEL). Scale bars indicate 20 μm. $N = 4$ in WT/SAL/control IgG and WT/SAL/anticorisin groups and $n = 9$ in WT/BLM/control IgG and WT/BLM/anticorisin groups. Bars indicate the mean ± S.D. Statistical analysis by ANOVA with a post hoc Newman–Keuls test. ***$p < 0.001$. **d, e** Cleavage of caspase-3 confirmed by western blotting and quantified by an image software. Representative blots from two independent experiments with similar results are shown. $N = 3$ in WT/SAL/control IgG and WT/SAL/anticorisin groups and $n = 4$ in WT/BLM/control IgG and WT/BLM/anticorisin groups. Bars indicate the mean ± S.D. Statistical analysis by ANOVA with a post hoc Newman–Keuls test. *$p < 0.05$; ****$p < 0.0001$. The source data underlying (**a**, **c**, **e**) are provided in the Source Data file.

control IgG by intraperitoneal route once a day every other day for a week before administering a high dose of LPS (150 μg) by intratracheal instillation (Fig. 8a). Mice with LPS-induced acute lung injury pretreated with anticorisin mAtb showed reduced lung CT opacity, less number of lung infiltrating neutrophils, and significantly decreased plasma levels of lactate dehydrogenase A (LDHA), SP-D, and MMP-1, and significantly reduced BALF levels of LDHA, MMP-1, MUC-1, and tumor necrosis factor-α (TNFα) compared to mice pretreated with control IgG. In addition, the plasma level of chemokine (C-C motif) ligand 2 (CCL2) and the BALF level of SP-D were also reduced in mice with acute lung injury pretreated with anticorisin mAtb compared to their counterparts pretreated with the control IgG, although the reduction was not statistically significant. Overall, these observations suggest the involvement of corisin in the acute inflammatory response to LPS (Fig. 8b–f).

In a separate experiment using less amount (75 μg) of intratracheal LPS to induce a less severe acute lung injury, we found a significantly decreased number of lung infiltrating neutrophils and significantly reduced plasma levels of SP-D, CCL2, TNFα and a significant reduction in the BALF levels of SP-D and CCL2 in mice pretreated with anticorisin mAtb compared to mice pretreated with control IgG (Supplementary Fig. 31a–d). In addition, the plasma levels of LDHA, MUC5B, and the BALF levels of TNFα, LDHA, and MMP-1 were also reduced in mice pretreated with anticorisin mAtb compared to mice pretreated with control IgG, although the reduction was not statistically significant. Overall, these observations further support the implication of corisin in the pathogenesis of acute lung injury.

**Corisin increases the expression of pro-inflammatory factors and enhances the proapoptotic activity of BLM and LPS in alveolar epithelial cells.** The significant amelioration of lung inflammation in mice with BLM-induced pulmonary fibrosis and LPS-induced acute lung injury treated with anticorisin mAtb suggests the participation of corisin in the mechanism of both disease models. To further interrogate this observation, we cultured A549 cells for 24 h in the presence of corisin and assessed the expression of cytokines and chemokines. Corisin significantly increased the mRNA expression of CCL2, CXCL1, CXCL2, IL-8, the secretion of CCL2, CCL3, CXCL1, IL-8, and the activation of the NFκB pathway in A549 alveolar epithelial cells compared to controls (Supplementary Fig. 32a, b). Activation of the NFκB pathway is probably secondary to increased levels of chemokines in the culture supernatant of cells treated with corisin. In a separate experiment, we cultured A549 cells in the presence of corisin alone or in the presence of a combination of corisin and BLM or LPS and assessed apoptosis by flow cytometry. Corisin in combination with BLM or LPS significantly increased the percentage of apoptotic cells compared to BLM or LPS alone (Supplementary Fig. 33a–c). These observations suggest that corisin

per se stimulates the secretion of cytokines and chemokines and enhances LPS-induced and BLM-induced apoptosis of alveolar epithelial cells.

**The plasma level of corisin is a potential biomarker of pulmonary fibrosis.** The occurrence and frequency of AE predict accelerated lung fibrosis and unfavorable clinical outcome in IPF patients[1]. Therefore, we reasoned that circulating corisin may be a biomarker of pulmonary fibrosis. We designed an enzyme immunoassay to investigate this premise using an antitransglycosylase polyclonal antibody and biotinylated anticorisin mAtb. We measured the plasma levels of corisin in wild-type (WT) mice and TGFβ1 transgenic mice (TG) with pulmonary fibrosis caused by lung-specific overexpression of the full-length human TGFβ1 encoding gene and evaluated correlation with fibrosis markers (Supplementary Fig. 34a). We also measured the lung tissue levels of corisin in mice with BLM-induced lung fibrosis and control mice (described in Figs. 6, 7) collected on day 22 after starting bleomycin infusion and evaluated correlation with fibrosis markers. As expected, the CT fibrosis score, Ashcroft fibrosis score, and the lung hydroxyproline content were significantly increased in TGFβ1 TG mice with fibrosis compared to WT mice (Supplementary Fig. 34b). TGFβ1 TG mice showed significantly elevated plasma concentration of corisin compared to WT mice, and the plasma concentration of corisin significantly correlated with the CT fibrosis score, Ashcroft fibrosis score, and the lung hydroxyproline content (Supplementary Fig. 34b).

WT mice with BLM-induced lung fibrosis treated with control IgG showed significantly higher levels of corisin in lung tissue homogenate than WT mice without fibrosis. However, WT mice with lung fibrosis, induced by BLM, treated with anticorisin mAtb showed significantly low lung tissue levels of corisin compared to mouse counterparts treated with control IgG (Supplementary Fig. 34c). In addition, the lung tissue levels of corisin were significantly and proportionally correlated with the CT fibrosis score, Ashcroft fibrosis score, lung collagen (trichrome) stained area, and the lung hydroxyproline content. Overall, these observations support the potential application of corisin as a biomarker of pulmonary fibrosis.

**High circulating levels of corisin in IPF patients with AE.** Based on the above results, we compared the serum levels of corisin among healthy subjects, IPF patients with stable disease, and AE. The serum concentration of corisin was significantly increased in IPF patients with the stable disease compared to heathy subjects and in IPF patients with AE compared to IPF patients with stable disease, further supporting the potential usefulness of corisin as a biomarker in IPF patients (Fig. 9a, b).

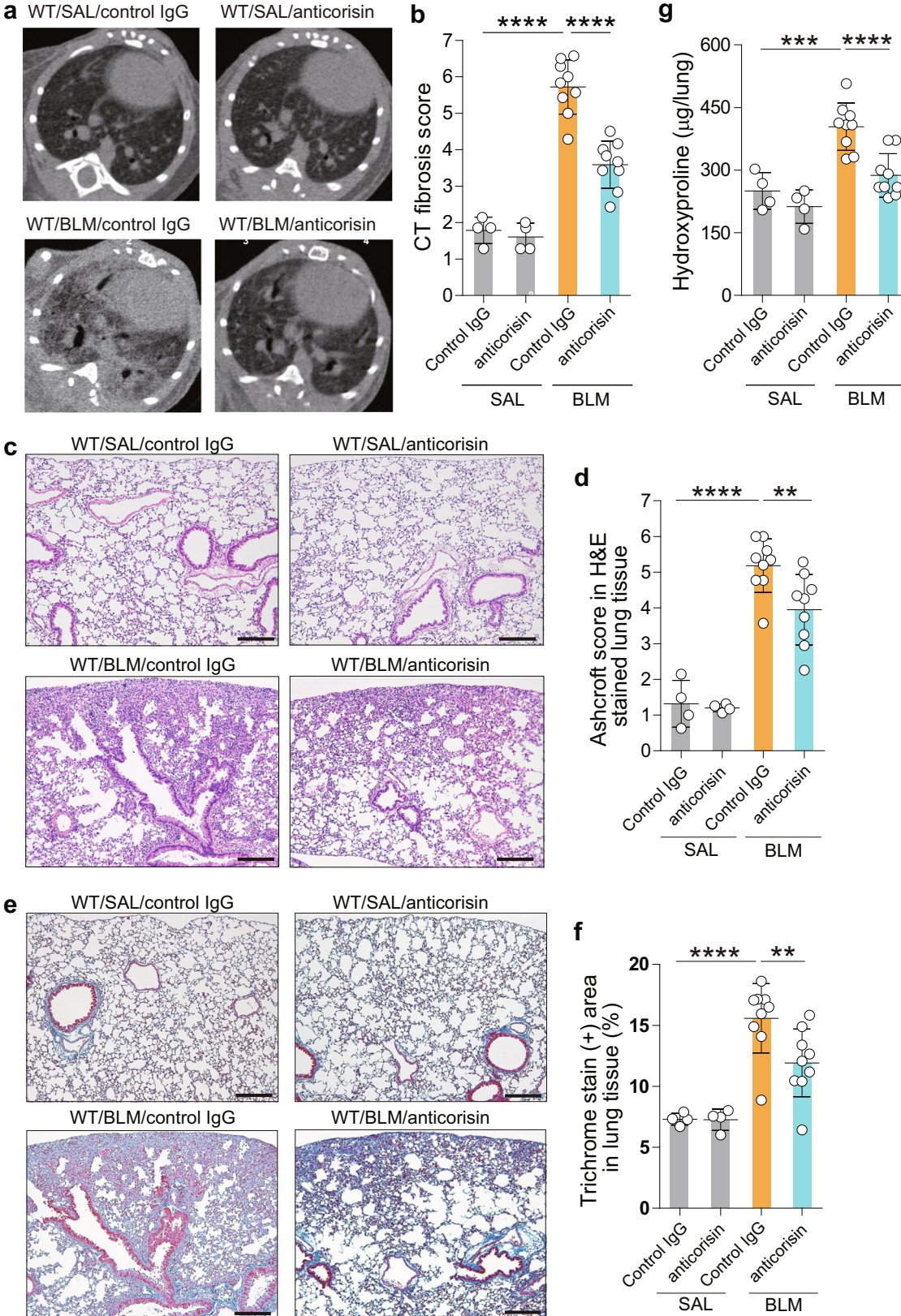

## Discussion

This study showed that a putative serine protease secreted by a bacterium cleaves and releases corisin from an IsaA-like trans-glycosylase and that treatment with an anticorisin neutralizing mAtb ameliorates AE of pulmonary fibrosis and LPS-induced acute lung injury.

Injury and apoptosis of alveolar epithelial cells play a significant role in the pathogenesis of IPF[32]. The injured lung epithelium releases profibrotic growth factors and cytokines such as TGFβ1 that promote epithelial-mesenchymal transition, recruitment, and activation of fibroblasts that cause aberrant lung tissue repair and excessive matrix deposition[33,34]. Enhanced apoptosis

**Fig. 7 Amelioration of tissue fibrosis in mice with bleomycin-induced pulmonary fibrosis treated with anticorisin monoclonal antibody.** Wild-type (WT) mice received bleomycin (BLM) by osmotic mini-pumps and treated with anticorisin monoclonal antibody (mAtb) (WT/BLM/anticorisin) or control IgG (WT/BLM/control IgG) by intraperitoneal route three times a week for 3 weeks. WT mice receiving saline (SAL) by osmotic mini-pumps and treated with anticorisin mAtb (WT/SAL/anticorisin) or control IgG (WT/SAL/control IgG) by intraperitoneal route three times a week for 3 weeks were the control mice. **a, b** Computed tomography (CT) was performed 1 day before mouse euthanasia. The radiological findings of lung fibrosis were evaluated using a CT fibrosis score as described under Methods. $N = 4$ in WT/SAL/control IgG and WT/SAL/anticorisin groups and $n = 9$ in WT/BLM/control IgG and WT/BLM/anticorisin groups. Data are the mean ± S.D. Statistical analysis by ANOVA with a post hoc Newman–Keuls test. ****$p < 0.0001$. **c, d** The grade of lung fibrosis in hematoxylin & eosin (H&E) stained lung tissue were evaluated using the Ashcroft score. Scale bars indicate 200 μm. $N = 4$ in WT/SAL/control IgG and WT/SAL/anticorisin groups and $n = 9$ in WT/BLM/control IgG and WT/BLM/anticorisin groups. Bars indicate the mean ± S.D. Statistical analysis by ANOVA with a post hoc Newman–Keuls test. **$p < 0.01$; ****$p < 0.0001$. **e, f** Lung collagen deposition was evaluated by Masson's trichrome staining, and the percentage of trichrome stain (+) area was measured using the WinRoof Image Processing Software. Scale bars indicate 200 μm. $N = 4$ in WT/SAL/control IgG and WT/SAL/anticorisin groups and $n = 9$ in WT/BLM/control IgG and WT/BLM/anticorisin groups. Bars indicate the mean ± S.D. Statistical analysis by ANOVA with a post hoc Newman–Keuls test. **$p < 0.01$; ****$p < 0.0001$. **g** The lung tissue content of hydroxyproline was measured by a colorimetric assay using a commercially available kit following the manufacturer instructions. $N = 4$ in WT/SAL/control IgG and WT/SAL/anticorisin groups and $n = 9$ in WT/BLM/control IgG and WT/BLM/anticorisin groups. Bars indicate the mean ± S.D. Statistical analysis by ANOVA with a post hoc Newman–Keuls test. ***$p < 0.001$; ****$p < 0.0001$. The source data underlying (**b**, **d**, **f**, **g**) are provided in the Source Data file.

leads to denudation of the alveolar lining epithelium and replacement by the overgrowing fibroblasts (Fig. 10)[32]. The initial factor triggering apoptosis in IPF remains unknown. However, we recently reported that corisin, a 19-residue peptide produced by *Staphylococcus nepalensis* strain CNDG, isolated from lung tissue with advanced fibrosis, causes apoptosis of alveolar epithelial cells and AE of pulmonary fibrosis in experimental animal models[17]. In the present study, we demonstrate that a corisin-like peptide derived from three *Staphylococcus haemolyticus* strains, isolated from the same fibrotic tissue as *S. nepalensis* strain CNDG, recapitulates the apoptotic activity of corisin. Furthermore, the significantly high concentration of corisin in serum and BALF from IPF patients with AE compared to stable patients indicates the involvement of bacterial released proapoptotic peptides in the pathogenesis of the human fibrotic disease[17].

Corisin is a component of an IsaA-like transglycosylase that is highly conserved among members of the genus *Staphylococcus*[17]. The cleavage mechanism of corisin from the transglycosylase has been unclear. However, here we demonstrated that incubation of recombinant transglycosylase in the presence of the culture supernatant of *S. nepalensis* results in degradation of the recombinant protein, releasing by-products that have apoptotic activity. Importantly, while it requires large amounts of synthetic peptides (μg/ml) to observe the deadly effects of corisin on lung cells either in vitro or in vivo, we demonstrate that the native peptide, i.e., proteolytically released corisin in the BALF of patients or in the bacterial culture supernatant or from recombinant transglycosylase is a highly potent proapoptotic peptide (effective at the picogram/ml level). It is our hypothesis that the proteolytically cleaved corisin peptide assumes a native state that is poised to elicit its apoptotic activity on lung alveolar epithelial cells. In contrast, the artificial peptide, synthesized in a linear form, is likely a mix of peptides in diverse folded states, with a significant portion failing to achieve the functional corisin fold and therefore requiring larger amounts to elicit apoptotic activity. The eventual purification of the corisin-releasing protease, reported in the present study, will allow release of larger amounts of corisin from recombinant transglycosylases, embedded with the proapoptotic peptide, to gain a deeper understanding of their roles in pulmonary fibrosis. In silico analysis of the genomes of the *S. nepalensis* and *S. haemolyticus* strains, demonstrated to elicit apoptosis of lung alveolar epithelial cells, unveiled several serine proteases, cysteine proteases, and metalloproteases that may be potentially responsible for the release of corisin. We performed immunoassays using a panel of protease inhibitors and found that the protease cleaving corisin from the transglycosylase is a putative serine protease. We then developed and

characterized anticorisin mAtbs and found that an anticorisin neutralizing mAtb inhibits the apoptotic activity of the transglycosylase degradation products. However, the anticorisin mAtb was unable to block the degradation of the transglycosylase.

Overall, from these in vitro experiments, we concluded that a putative serine protease cleaves corisin from its full-length transglycosylase to elicit its apoptotic activity. The newly developed anticorisin mAtb inhibits the activity of corisin released during the transglycosylase degradation without affecting the proteolytic activity of the protease. As we previously demonstrated, corisin is pathogenic in AE, and in subsequent experiments, we evaluated whether the anticorisin neutralizing mAtb improves the AE of pulmonary fibrosis in two different animal models.

Increased apoptosis of alveolar epithelial cells plays a central role in the pathogenesis of IPF[35]. Studies performed in experimental animal models demonstrated that the induction of apoptosis in alveolar epithelial cells is sufficient to cause pulmonary fibrosis and that apoptosis inhibitors ameliorate the disease[35,36]. Apoptosis of alveolar epithelial cells in IPF may occur by the extrinsic or intrinsic pathway[35]. Activation of the intrinsic pathway by mitochondrial dysfunction is a major contributor to enhanced apoptosis in IPF[37–40]. Mitochondrial injury causes mitochondrial outer membrane permeabilization inducing the release of cytochrome c and second mitochondria-derived activator of caspase[35]. In the cytosol, cytochrome c activates apoptosis-protease activating factor-1, an activator of caspase-9, and the second mitochondria-derived activator of caspase blocks the activity of inhibitors of apoptosis proteins[37]. The mitochondrial outer membrane permeabilization is enhanced by several factors, including the Bcl2-associated X apoptosis regulator (Bax), Bcl2-associated death promoter (BAD), and p53-upregulated binding component (PUMA), and inhibited by Bcl2 apoptosis inhibitors including Bcl2 and Bcl-xL[38]. Damaged mitochondria release several damage-associated molecular patterns (DAMPs), including peptides, lipids, metabolites, and mitochondrial DNA, which are potent activators of the inflammatory response[37]. A recent clinical study showing that the serum levels of mitochondrial DNA can predict the risk of AE and progression of IPF[41] supports the critical role of mitochondria in the pathogenesis of IPF[41]. Consistent with the mitochondrial dysfunction in IPF, in the present study, we found that corisin and the culture supernatant of bacteria expressing corisin-harboring transglycosylases predominantly induce activation of caspase-9, and that corisin stimulation significantly affects the expression of factors that regulate the mitochondrial outer membrane permeabilization (Bcl2, Bcl-XL, Bax, cyclin D1) and inhibitors of apoptosis proteins (BIRC1, BIRC5, BIRC7), and p53 activation.

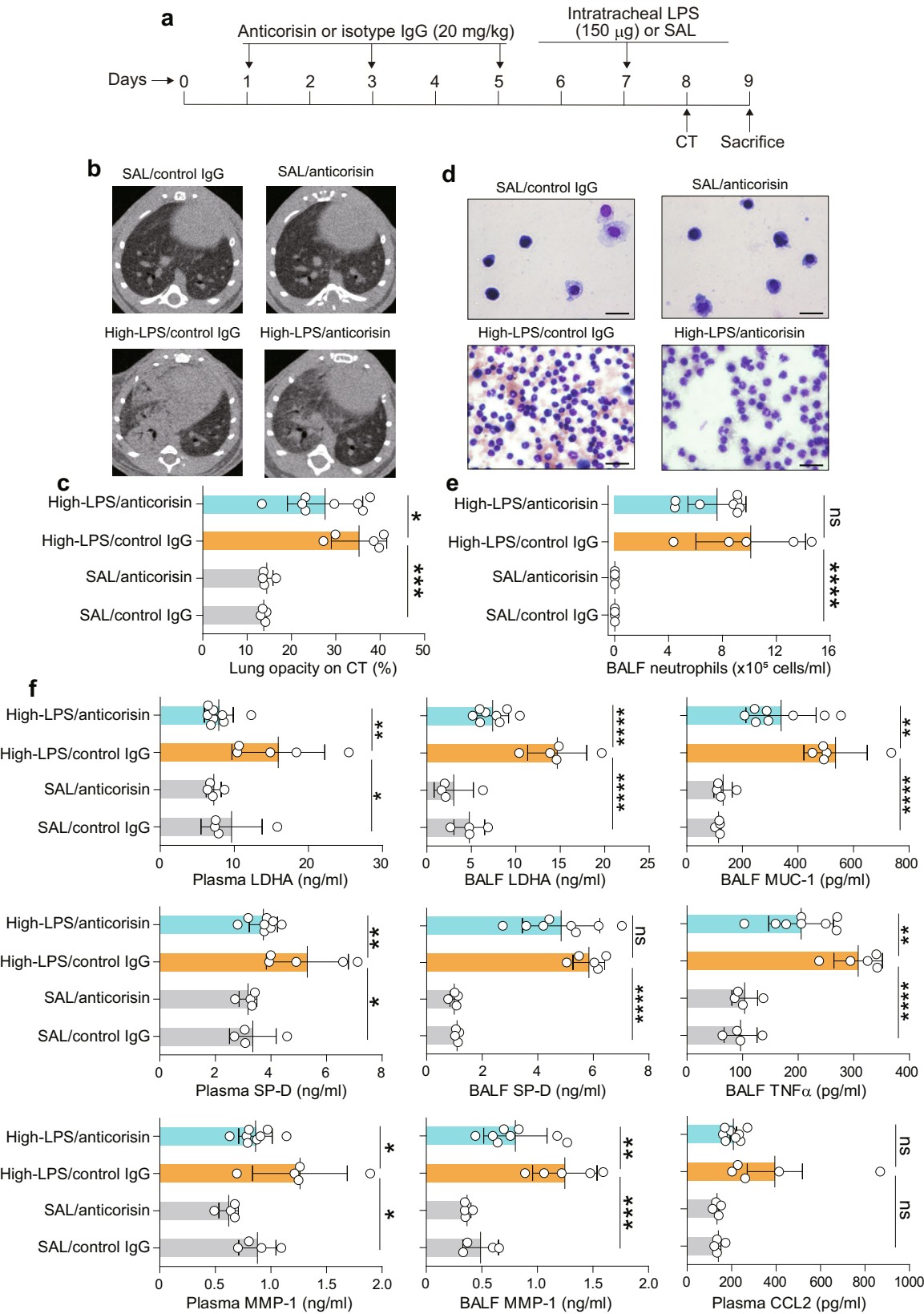

AE is the most frequent and fatal complication of IPF[5,7,42]. AE's characteristic clinical presentation includes a rapid and severe deterioration of clinical symptoms, worsening of lung radiological findings, and acute respiratory failure[5]. Patients with AE have a poor prognosis[5]. AE accounts for almost half of all IPF-related deaths, and the life expectancy of IPF patients with AE is no more than 4 months[6,43,44]. Therapy for AE-IPF is currently unavailable. International guidelines recommend the use of glucocorticosteroids[5,45]. However, there is no proven evidence of the therapeutic efficacy of glucocorticosteroids[46]. The main obstacle to developing a treatment for AE-IPF is its uncertain etiopathogenesis. AE may be idiopathic or triggered by

**Fig. 8 Monoclonal anticorisin mAtb attenuates severe lipopolysaccharide-induced acute lung injury. a** Wild-type mice were treated three times with control IgG or anticorisin monoclonal antibody at a dose of 20 mg/kg by intraperitoneal route once a day every other day. Mice received intratracheal instillation of a high-dose (150 μg) of lipopolysaccharide (LPS) or saline (SAL) 2 days after the last treatment with antibody and sacrificed 2 days after LPS instillation. Mice receiving intratracheal saline (SAL) and treated with control IgG. (SAL/control IgG) or anticorisin antibody (SAL/anticorisin) were the control mice. **b, c** Computed tomography (CT) was performed 1 day after the intratracheal instillation of lipopolysaccharide (LPS). $N = 4$ in WT/SAL/control IgG and WT/SAL/anticorisin groups, $n = 5$ in WT/LPS/control IgG group, and $n = 8$ in WT/LPS/anticorisin group. The radiological findings of LPS-acute lung injury were evaluated by measuring lung opacity on axial CT images using the WinRoof Image Processing Software as described under Methods. Data are the mean ± S.D. Statistical analysis by ANOVA with a post hoc Newman–Keuls' test. *$p < 0.05$; ***$p < 0.001$. **d, e** Mice were sacrificed on day 2 after intratracheal LPS instillation and bronchoalveolar lavage fluid (BALF) was collected. BALF cells were counted using a nucleocounter and stained with Giemsa for differential cell counting as described under Methods. Scale bars indicate 20 μm. $N = 4$ in SAL/control IgG and SAL/anticorisin groups, $n = 5$ in LPS/control IgG group, and $n = 8$ in LPS/anticorisin group. Data are the mean ± S.D. Statistical analysis by ANOVA with a post hoc Neuman–Keuls test. ****$p < 0.0001$. ns not significant. **f** The levels of lactate dehydrogenase A (LDHA), surfactant protein-D (SP-D), matrix metalloproteinase-1 (MMP-1), MUC-1, tumor necrosis factorα (TNFα) and chemokine (C-C motif) ligand 2 (CCL2) were measured using commercially available immunoassay kit following the manufacturer instruction. $N = 4$ in SAL/control IgG and SAL/anticorisin groups, $n = 5$ in LPS/control IgG group, and $n = 8$ in LPS/anticorisin group. Data are the mean ± S.D. Statistical analysis by ANOVA with a post hoc Neuman–Keuls test. *$p < 0.05$; **$p < 0.01$; ***$p < 0.001$; ****$p < 0.0001$. ns not significant. The source data underlying (**c**, **e**, **f**) are provided in the Source Data file.

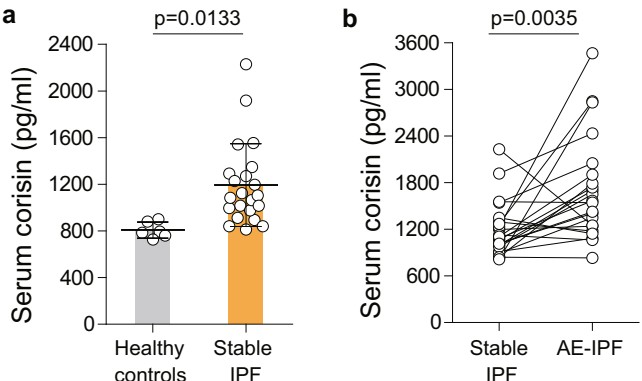

**Fig. 9 Significant increase in serum corisin level in idiopathic pulmonary fibrosis patients with acute exacerbation. a**, **b** Corisin was measured using an enzyme immune assay using rabbit polyclonal antitransglycosylase antibody as coating antibody and rat monoclonal anticorisin antibody as secondary antibody. Healthy controls, $n = 6$; all stable idiopathic pulmonary fibrosis (IPF) patients, $n = 22$; IPF patients with acute exacerbation, $n = 22$. Bars indicate the means ± S.D. Statistical difference between healthy controls and stable IPF patients was evaluated by two-sided unpaired $t$ test, and statistical difference between stable IPF and acute exacerbation-IPF patients by two-tailed paired $t$ test. The source data underlying (**a**, **b**) are provided in the Source Data file.

a secondary event (infection, aspiration)[5] (Fig. 10); however, in either case, the clinical features and prognosis are similar, and the causative factor is unknown[5,47].

Studies showing that IPF patients with AE have a four-fold lung bacterial burden, a dramatic alteration in the lung bacterial community, and a significant increase of antibacterial immunoglobulins compared to patients with stable disease implicate the lung microbiome in the pathogenesis of AE[15,16]. However, the mediating factor and triggering mechanism are unclear. The increased apoptosis of alveolar epithelial cells in the fibrotic lung tissue with AE compared to the fibrotic lung tissue with stable disease points to apoptosis as a central mechanism for triggering AE[32,35,48,49]. We previously demonstrated that the lung microbiome-derived factor corisin triggers AE of lung fibrosis by inducing apoptosis of alveolar epithelial cells[17]. In the present study, we found that treatment with a neutralizing anticorisin mAtb improves abnormal radiological findings and lung inflammation induced by corisin in a mouse model of TGFβ1-associated lung fibrosis. Also, in mice with BLM-induced lung fibrosis, we found that corisin exacerbates the disease and that treatment with the anticorisin mAtb during the acute and chronic

phases of the disease ameliorates pulmonary fibrosis compared to mice treated with a control Atb. Moreover, additional experiments showed that treating mice with anticorisin mAtb during the acute phase of BLM-induced injury is sufficient to suppress lung fibrosis development in the chronic phase, further supporting the pathogenic role of corisin in acute inflammatory responses (Fig. 10). Inhibition of the apoptotic activity of BALF from IPF patients with AE on alveolar epithelial cells by the anticorisin mAtb highlights the clinical relevance of corisin in the pathogenesis of AE in IPF. Overall, these beneficial effects of the neutralizing anticorisin mAtb in vivo indicate that corisin plays a causative role in AE, thereby pointing to corisin as a potential molecular target for the therapy of this devastating complication of IPF.

Another question addressed in the present study is whether the pathogenic role of corisin is specific to pulmonary fibrosis. Whether the corisin-releasing lung microbiota of IPF patients or their peptides are similar to microbiota of normal subjects is currently unknown. However, the detectable levels of corisin in BALF from healthy subjects and in tissue from mice with no fibrosis, albeit at lower levels, suggest that the microbiota releases corisin (or corisin-like peptides) even under physiological conditions[17]. Therefore, the involvement of corisin in other diseases associated with lung injury, such as acute respiratory distress (ARDS) and other diseases is predictable[50,51]. ARDS is an acute inflammatory lung injury associated with high morbidity and mortality[51]. Pulmonary or extrapulmonary infections, secondarily affecting the lungs, may cause ARDS[52]. AE-IPF and ARDS share many pathophysiological characteristics[53]. Diffuse alveolar damage, increased pro-inflammatory cytokine concentration, and enhanced neutrophils' recruitment in the lungs are common features of both pathological entities[54]. Importantly, similar to AE-IPF, apoptosis of lung cells also occurs in ARDS[55]. Injury and apoptosis of lung epithelial cells and endothelial cells result in increased pulmonary microvascular permeability, alveolar edema, and disruption of the alveolar epithelial cell barrier that ultimately causes hypoxemia in ARDS patients[53,55]. The most commonly used animal model of ARDS is LPS-induced acute lung injury[56].

LPS is a component of Gram-negative bacteria outer cell wall that may cause lung injury by systemic or intrapulmonary administration[57]. Here, we administered a low or high dose of LPS to mice by the intratracheal route and evaluated the preventive effect of anticorisin mAtb against acute lung injury. Irrespective of the dose of LPS, mice treated with the anticorisin mAtb showed less radiological abnormalities and neutrophil infiltration in the lungs and reduced biological markers of acute lung injury compared to untreated mice, suggesting the potential

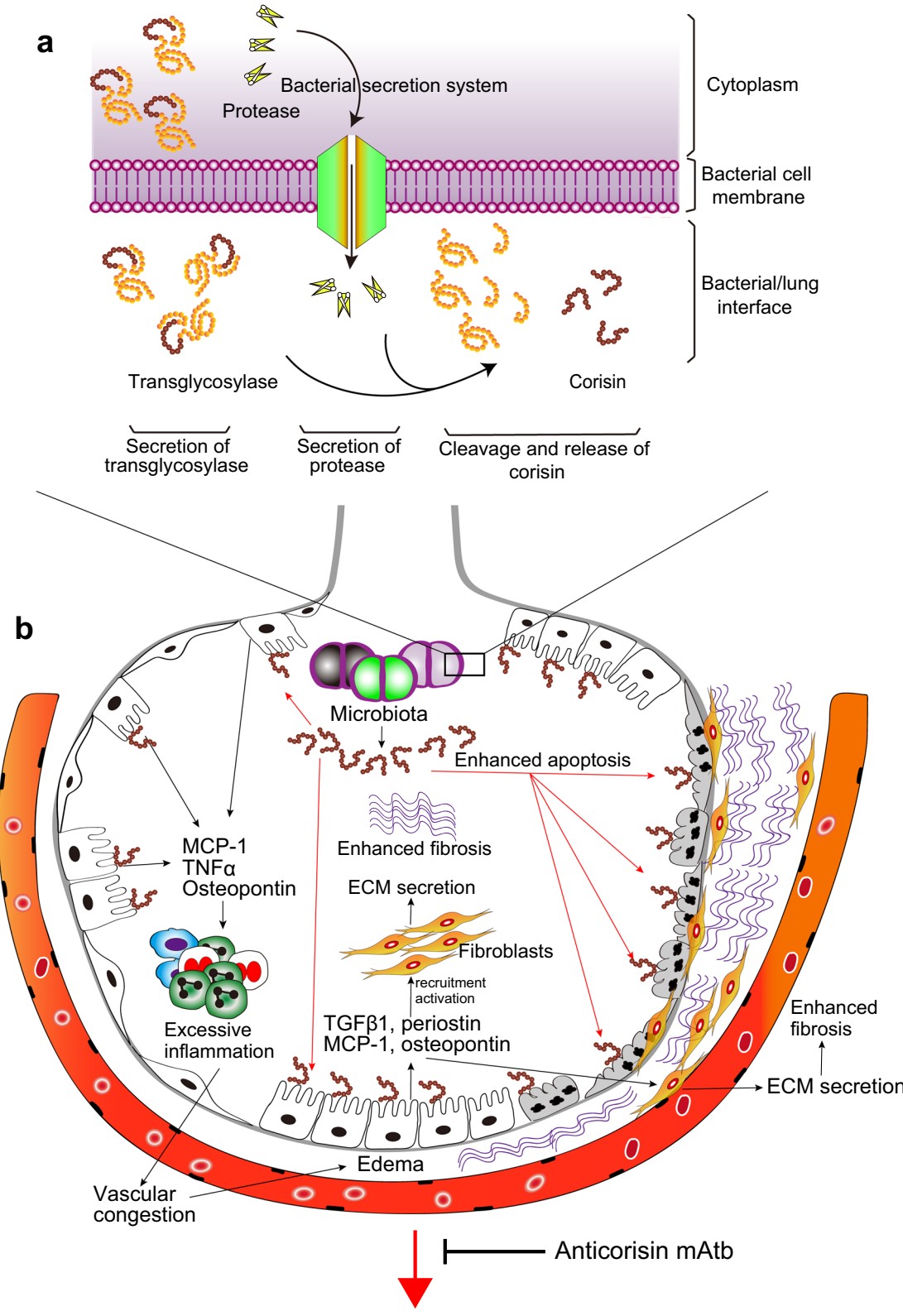

implication of corisin in the pathogenesis of ARDS. Accumulating evidence indicates that a dysregulated response of host cells to acute lung injury causes "pro-inflammatory cytokine storm" in ARDS and "profibrotic cytokine storm" in AE-IPF[1,55]. Therefore, it is possible to speculate that lung injury coupled with an aberrant host repair that leads to a fibrotic environment remodel the

microbiome into a community enriched with organisms such as the *Staphylococcus* spp, described in the present study, and other pathogenic bacteria that harbor and release corisin and its derivative peptides. In a cellular milieu, where the alveolar epithelial cells have sustained acute injury and therefore predisposition to the mitochondrial-targeting and damaging corisin, the enhanced

**Fig. 10 A model of the process of corisin-induced acute alveolar damage during acute exacerbation of pulmonary fibrosis and the protective role of the anticorisin monoclonal antibody. a** Environmental factors that trigger enhanced fibrosis and injury (e.g., drugs, viral and bacterial infections, radiotherapy, or unknown factors), remodel the lung environment to enrich for a salt-tolerant microbial community. Within this community are Staphylococcus spp (e.g., *S. nepalensis* and *S. haemolyticus*) harboring transglycosylases with corisin and corisin-like peptides embedded in their C-terminal region. The corisin-containing bacteria secrete their transglycosylases together with a protease that cleaves the proapoptotic peptides from the transglycosylases. **b** Increased intra-alveolar concentration of the deadly peptides (corisin and corisin-like peptides) in the predisposed host stimulates secretion of inflammatory cytokines (TNFα, osteopontin) chemokines (MCP-1), profibrotic cytokines (TGFβ1), and periostin from alveolar epithelial cells that enhance the inflammatory response, the recruitment of fibroblasts, and the deposition of extracellular matrix in the lungs. Lung areas with enhanced apoptosis of alveolar wall lining epithelial cells caused by corisin or corisin-like peptides are replaced by lung fibrotic tissue. The process accelerates the clinical progression (acute exacerbation) of the disease that ultimately has a fatal outcome. The anticorisin monoclonal antibody binds to the peptides to block the pro-inflammatory and proapoptotic activity of the deadly peptides, ameliorating acute exacerbation of the disease. mAtb monoclonal antibody, TNFα tumor necrosis factor-α, MCP-1 monocyte chemoattractant protein-1, TGFβ1 transforming growth factor-β1.

secretion of the proapoptotic peptide (due to the enriched corisin-producing bacteria) then leads to a concomitant release of potent activators that exacerbate the apoptotic activity on the lung alveolar epithelial cells. Enhancement of LPS- and BLM-induced apoptosis and expression of cytokines and chemokines in alveolar epithelial cells observed in the present study supports this assumption. It is thus reasonable to propose that the "first hit" or lung injury increases the vulnerability of the lung to the proapoptotic activity of corisin that likely constitutes the "second hit" or "final hit" for triggering AE-IPF or the most severe form of acute lung injury termed ARDS. Importantly, the results of the present study suggest that the anticorisin neutralizing mAtb may prevent AE-IPF or ARDS by mitigating the "final hit" of lung injury.

The development of a biomarker is another unmet clinical need for improving the health care of IPF patients[2]. Identifying, developing, and validating biomarkers for diagnosis and predicting the clinical outcome or the response to therapy in IPF patients is challenging. There are many reports of potentially useful markers, including products secreted by alveolar type II epithelial cells (Krebs von Lung-6, surfactant protein-A, surfactant protein-D), extracellular matrix proteins (periostin, fibronectin), metalloproteinases (MMP-1, MMP-7), coagulation factors (antithrombin, plasminogen activator inhibitor-1), inflammatory chemokines (CCL-18, CXCL-13) and peripheral-blood transcriptome and proteome signatures[58–65]. However, most of the reported biomarkers remain incompletely validated. The ideal biomarker would be the level of the factor causing the disease[66]. As corisin plays an active role in the AE of pulmonary fibrosis, we posit that the level of corisin would be a marker of fibrosis. To interrogate this hypothesis, we evaluated the relationship between the levels of corisin and fibrosis markers in TGFβ1 TG and BLM-induced mouse models of pulmonary fibrosis. We found that the significant increase in the level of corisin was proportional to the degree of lung fibrosis assessed by radiological and histological examination and by the lung hydroxyproline content. These findings suggest the potential of corisin to become a useful biomarker of pulmonary fibrosis. The significantly high concentration of corisin in serum and BALF from IPF patients with the stable or exacerbated disease compared to a healthy population further supports this assumption[17].

In the present study, we have confined our analyses to IPF, with some extension to ARDS. However, it is likely that other lung diseases with underlying injury and aberrant lung tissue repair, including the current Covid-19 outbreak, can remodel the lung microbiome and predispose the host to the organisms harboring and shedding the proapoptotic factor corisin and its derivative peptides. In fact, the literature indicates that the previous outbreak of the corona-virus disease designated severe acute respiratory syndrome or SARS (caused by SARS-COV) subsequently led to the development of pulmonary fibrosis in many

survivors after recovery[67], an observation supported by follow up radiographic findings with survivors of Middle East respiratory syndrome[68], and by extension, the current Covid-19 infections[69].

## Methods

**Reagents.** We purchased the human lung epithelial cell line A549 and hypersaline media (ATCC media 1097, 2168) from the American Type Culture Collection (Manassas, VA), Dulbecco's Modified Eagle Medium (DMEM) from Sigma-Aldrich (Saint Louis, MO), and fetal bovine serum (FBS) from Bio Whittaker (Walkersville, MD). L-glutamine, penicillin, and streptomycin were from Invitrogen (Carlsbad, CA). Normal human bronchial epithelial (NHBE) cells were from Clonetics (Walkersville, MD). Peptide Institute Incorporation (Osaka, Japan) and ThermoFisher Scientific (Waltham, MA, USA) prepared and provided synthetic corisin and the corresponding synthetic scrambled peptide. BLM was from Nihon Kayaku (Tokyo, Japan) and Alzet osmotic mini-pumps (model 2001) from Alza Corporation (Palo Alto, CA).

**Subjects.** This study comprised 36 Japanese patients with IPF and 6 Japanese male healthy volunteers (Supplementary Table 1). All consecutive IPF patients consulted or were referred to the health care institutions (Tosei General Hospital, Aichi, Japan or the National Hospital Organization Kinki-Chuo Chest Medical Center, Osaka, Japan) for breathlessness. Diagnosis of IPF and AE was following accepted international criteria[5]. A bronchoscopy study was performed following the American Thoracic Society guidelines, and BALF samples were collected from 14 IPF patients with AE and 5 healthy volunteers[70]. Serum samples collected during stable disease and during AE were available in 22 IPF patients.

**Animals.** We purchased female WT C57BL/6J mice from Nihon SLC (Hamamatsu, Japan) to induce lung injury, pulmonary fibrosis, and AE. WT mice weighing 20–22 g and 8–9 weeks of age were used for the BLM-induced lung fibrosis experiment. We also used male and female transforming growth factorβ1 (TGFβ1) TG mice in a C57BL/6J background that develop spontaneously progressive and fatal pulmonary fibrosis and their WT littermates in the experiments[17,33,71]. The TG mice and WT littermates weighed 23–26 g and aged 8–10 weeks. We bred and maintained the mice in a specific pathogen-free environment at a temperature of 21 °C, relative humidity of 50–70% and under a constant 12-h light/dark cycle in the facility for experimental animals of Mie University. The plastic cage of the mice was supplied with wood-wool nesting material, and mice had access to water and food (standard bait, CE-7 pellet, CLEA Japan Incorporation) *ad libitum*. TG mice were genotyped by standard PCR analysis, using DNA isolated from the tail of mice and specific primer pairs[17].

**Ethics approval.** All subjects participating in the clinical investigation provided written informed consent, and the study protocol was approved by the Ethical Committees for Clinical Investigation of Mie University (approval No: H2019064, date: 25/04/2019; approval No H2021-073; date 2021/4/27), Matsusaka Municipal Hospital (approval date: 11/06/2014) and conducted following the Principles of the Declaration of Helsinki. The Recombinant DNA Experiment Safety Committee (approval No: I-614 (henko1); date: 2013/15/12; approval No: I-708, date: 13/02/2019) and the Committee for Animal Investigation of Mie University approved the experimental protocols (approval No: 25-20-hen1-sai1, date: 23/07/2015; approval No: 29-23, date: 15/-01/2019). We performed all experimental procedures following internationally approved laboratory animal care principles published by the National Institute of Health (https://olaw.nih.gov/). The research followed the ARRIVE Guidelines for animal investigation, and variables were measured blindly of the treatment groups.

**CT examination.** CT of the lungs was performed with a micro-CT Latheta LCT-200 purchased from Hitachi Aloka Medical (Tokyo, Japan). After anesthesia with isoflurane inhalation, mice were placed in a prone position for data acquisition[17,71].

The radiological findings of lung fibrosis in both TGFβ1-induced (TGFβ1 TG) and BLM-induced lung fibrosis models were evaluated using a CT fibrosis score as follows: score 1, normal radiological lung findings; 2, intermediate findings; 3, slight lung fibrosis; 4, intermediate findings; 5, moderate lung fibrosis; 6, intermediate findings; and 7, advanced lung fibrosis[17]. The radiological findings of LPS-acute lung injury were evaluated by measuring lung opacity on axial CT images using the WinRoof Image Processing Software. The average percentage of the total lung opacity area of all axial (transverse) CT images (56 slides) from each individual mouse was calculated, and the mean percentage of lung opacity area was compared among all groups.

**Screening of phosphoproteins by immunoblotting**. A549 alveolar epithelial cells were cultured up to subconfluency, and after overnight serum starvation, the cells were treated with 100 μM of corisin or scrambled peptide for 30 min. The cells were then washed with ice-cold phosphate-buffered saline and scraped into lysis buffer (20 mM 3-(N-morpholino) propane sulfonic acid (MOPS), pH 7.0, 2 mM EGTA, 5 mM ethylenediaminetetraacetic acid (EDTA), 30 mM sodium fluoride, 40 mM β-glycerophosphate, 10 mM sodium pyrophosphate, 2 mM sodium orthovanadate, 1 mM phenylmethanesulfonyl fluoride, 3 mM benzamidine, 5 μM pepstatin, 10 μM leupeptin, and 0.5% Triton X-100; final pH 7.0), and sonicated for 15 s. The cells were centrifuged at 100,000 rpm for 30 min at 4 °C. Protein concentration was determined by the Bradford assay. Phospho-kinase screening was performed by Kinexus™ Corporation.

**Preparation of protease fraction**. We prepared and serially concentrated 100 ml of *Staphylococcus nepalensis* culture supernatant using >50 kDa filters and concentrated the resulting flow-through fractions using >10 kDa filters. Then, we loaded the >10 kDa-concentrated samples onto a Sephacryl S-300 column to separate several fractions (1 ml each). The products of the proteolytic activity of each fraction on a corisin-containing recombinant transglycosylase were resolved by sodium dodecyl sulfate-polyacrylamide gel electrophoresis and silver staining. Silver staining was performed using the Daiichi 2-D Silver Staining Kit (Daiichi, Tokyo, Japan). The reaction mixture consisted of 5.5 μL 1x digestion buffer (20 mM Tris-HCl, 140 mM NaCl, pH 7.5), 0.5 μL of 2 mg/ml recombinant transglycosylase and 6 μL of each fraction. We then prepared a pool of the fractions with high digesting activity and concentrated to 0.5 ml. We then loaded the concentrated sample onto a Sephacryl S-300 column, separated in several fractions, and measured absorbance at 280 nm. The digesting activity of each fraction on recombinant transglycosylase 351 was then assessed. We used the fractions with high proteolytic activity in the experiments.

**Degradation assay of recombinant transglycosylase**. We prepared a 50-μl reaction mixture containing 12.5 μl of recombinant transglycosylase (2 μg/ml), 5 μl of 10× digestion buffer (Tris-Cl, 1.4 M NaCl, pH 7.5), 25 μl of *Staphylococcus nepalensis* culture supernatant, and 7.5 μl of distilled water. The reaction mixture was incubated at 37 °C overnight, and then 10 μl of the reaction mixture was run on sodium dodecyl sulfate-polyacrylamide gel electrophoresis (SDS-PAGE), and products were visualized with silver staining. To determine if the mAtb will block the protease degradation of the transglycosylase, the reaction mixture containing the recombinant transglycosylase was first reacted/incubated with the mAtb for 30 min and then the protease-containing culture supernatant was added and the degradation products examined by silver staining.

**Assay for transglycosylase degradation inhibition**. We prepared a reaction mixture with a digestion buffer containing recombinant transglycosylase (2 mg/ml), *Staphylococcus nepalensis* culture supernatant, varying concentrations (0.1, 0.3, 1 mM) of a serine protease inhibitor (pefabloc SC or diisopropyl fluorophosphate [DFP]), a cysteine protease inhibitor (E-64) or a chelating agent (EDTA). We then incubated the mixture for 0.5, 1, and 2 h before running on an SDS-PAGE and silver staining.

**Cell culture**. Culture of the human A549 alveolar epithelial cells and normal human bronchial epithelial (NHBE) cells was performed in DMEM supplemented with 10% fetal calf serum, 0.03% (w/v) L-glutamine, 100 IU/ml penicillin, and 100 μg/ml streptomycin in a humidified, 5% CO$_2$ atmosphere at 37 °C. In most in vitro studies, we used A549 cell lines due to their higher potential growth and because previous studies have shown that they mimic more the phenotype of alveolar type II cells than primary NHBE cells[72,73].

**Apoptosis assay**. A549 cells ($4 \times 10^5$ cells/well) were cultured in 12-well plates up to subconfluency, and after serum starvation for 12 h, they were treated with corisin or the corisin-like peptides, or a scrambled peptide based on the corisin sequence to evaluate apoptosis after 48 h. Apoptosis was assessed by flow cytometry (FACScan, BD Biosciences, Oxford, UK) after staining with fluorescein-labeled annexin V and propidium iodide (FITC Annexin V Apoptosis Detection Kit with PI, Biolegend, San Diego, CA). The percentage of cells positive for cleaved caspase-8, cleaved caspase-9, and cleaved caspase-3 was evaluated by flow cytometry using antibodies from Cell Signaling Technology (Beverly, MA). The gating strategy for

the flow cytometry study of apoptosis is described in (Supplementary Fig. 35 and Supplementary Fig. 36). Apoptosis of A549 cells was also evaluated by DNA laddering assay, TUNEL assay performed at MorphoTechnology Corporation (Sapporo, Hokkaido, Japan) or by using a commercial TUNEL apoptosis assay kit (AAT Bioquest, Sunnyvale, CA) following the manufacturer's instructions. To evaluate apoptosis of lung epithelial cells induced by BALF from IPF patients with AE, A549 alveolar epithelial cells were cultured in the presence of BALF from healthy controls, and IPF patients with AE treated with the anticorisin neutralizing mAtb clone 21 A or control IgG for 48 h, and apoptosis was evaluated by flow cytometry analysis. To conduct apoptosis inhibition assay, A549 alveolar epithelial cells were cultured up to subconfluency, serum-starved overnight, and pre-treated with anticorisin mAtb clone 2A, 9A, or 21A, added corisin or bacterial supernatant, and continued culturing for an additional 48 h. The number of apoptotic cells was evaluated by flow cytometry as described above. To evaluate the proapoptotic activity of transglycosylase degradation products, A549 alveolar epithelial cells ($2 \times 10^5$ cells/wells) were cultured in a 12-well plate up to subconfluency. After overnight serum starvation, a reaction mixture containing 5 μM recombinant transglycosylase, 1/10 volume of *Staphylococcus nepalensis* supernatant, and 5 μM anticorisin mAtb or control IgG was added to the cell culture and incubated for 48 h. Apoptosis was evaluated, as described above.

**Mitochondrial Membrane Potential Assay**. We measured the loss of mitochondrial membrane integrity using the fluorescent dye JC-1 (5,5′, 6,6′-tetrachloro-1,1′, 3,3′-tetraethyl tetrethyl benzimidalyl carbocyanine iodide) from Dojindo (Kumamoto, Japan). The ratio of red (585 nm) to green (530 nm) fluorescence represents the changes in mitochondrial membrane potential. A549 cells were stained with 2 μM JC-1 for 20 min at 37 °C after treatment with 5 μM corisin or scrambled peptide for 48 h. JC-1 fluorescence was detected using BD FACScan flow cytometry and data were analyzed using BD CellQuant software.

**Detections of Intracellular reactive oxygen species**. Cultured A549 cells ($2 \times 10^5$ cells/well) were treated with 5 μM corisin or scrambled peptide for 48 h. After washing with phosphate buffer solution (PBS), the cells were incubated with RPMI-1640 medium containing 20 μM 2′,7′-dichlorofluorescin diacetate (DCFDA) (R&D Systems, Minneapolis, MN, USA) for 20 min in the dark at 37 °C. Fluorescence intensity of DCF was detected using a BD FACScan flow cytometer. Data were analyzed using the BD CellQuest software.

**Induction of cytokines and chemokines by corisin in A549 alveolar epithelial cells**. A549 cells were cultured for 24 h in the presence of corisin and the expression of cytokines and chemokines were assessed by PCR and immunoassays. In a separate experiment, A549 cells were cultured in the presence of corisin alone or in the presence of a combination of corisin and BLM or LPS and apoptosis was assessed by flow cytometry.

**Transmission electron microscopy of apoptotic cells**. A549 cells ($10 \times 10^4$ cells/ml) were plated on collagen-coated 8-well chamber slides (BD Bioscience, San Jose, CA) and cultured until semi-confluency. Cells were serum-starved for 6 h and stimulated with the proapoptotic peptide (5 μM) for 16 h. Cells were fixed with 2% fresh formaldehyde and 2.5% glutaraldehyde in 0.1 M sodium cacodylate buffer (pH 7.4) for 2 h at room temperature. After washing with 0.1 M cacodylate buffer (pH 7.4), they were postfixed with 1% OsO$_4$ in the same buffer for 2 h at 4 °C. The samples were rinsed with distilled water, stained with 1% aqueous uranyl acetate for 2 h or overnight at room temperature, dehydrated with ethanol and propylene oxide, and embedded in epon (Epon 812 resin, Nakalai). After removing the cells from the glass, ultra-thin sections (94 nm) were cut, stained with uranyl acetate and Reynolds's lead citrate, and viewed with a transmission electron microscope (JEM-1010, JEOL, Tokyo, Japan).

**Genomic DNA sequencing and genome annotation**. As in our earlier report, genome sequencing was carried out using Oxford Nanopore Sequencing and Illumina Miseq nano sequencing as in our earlier report[17]. Briefly, genomic DNA from each bacterial strain (400 ng) was converted into a Nanopore library with the Rapid Barcoding library kit SQK-RAD004. The library was sequenced on a SpotON R9.4.1 FLO-MIN106 flowcell for 48 h on a GridION sequencer. The majority of the reads were 6 kb to 30 kb in length, although reads as long as 94 kb were also obtained. The Illumina Miseq sequencing was carried out by preparing shotgun genomic libraries with the Hyper Library construction kit from Kapa Biosystems (Roche). The library was quantitated by qPCR and sequenced on one MiSeq Nano flowcell for 251 cycles. Fastq files were generated and demultiplexed with the bcl2fastq v2.20 Conversion Software (Illumina).

Initial assembly of the Oxford Nanopore data was carried out using Canu[74], followed by polishing using Nanopolish[75] and Pilon (utilizing the Illumina MiSeq reads)[76], and finally, the genome was re-oriented using Circlator[77]. Another hybrid genome assembly was carried out using SPAdes[78], followed by reorienting the genome using Circlator. A hybrid genome assembly was also carried out using Unicycler[79]. The final hybrid genome assembly was generated using Unicycler, with the Canu assembly as the assembly backbone.

All assemblies were quality-assessed with BUSCO[80] and QUAST[81] and compared to a relevant reference genome using MUMmer[82]. The genome assemblies were annotated using the tool Prokka[83], and after evaluation, the best overall assembly was determined by combining the best overall BUSCO scores with the overall assembly metrics.

**Western blotting.** Tissue or sample was washed twice with ice-cold phosphate-buffered saline and then lysed in radioimmunoprecipitation assay (RIPA) buffer (10 mM Tris-Cl (pH 8.0), 1 mM EDTA, 1% Triton X-100, 0.1% sodium deoxycholate, 0.1% SDS, 140 mM NaCl, 1 mM phenylmethylsulfonyl fluoride) supplemented with protease/phosphatase inhibitors (1 mM orthovanadate, 50 mM β-glycerophosphate, 10 mM sodium pyrophosphate, 5 µg/mL leupeptin, 2 µg/mL aprotinin, 5 mM sodium fluoride). After centrifugation at 17,000 × g for 10 min at 4 °C, the protein concentration was measured using the Pierce BCA protein assay kit (Thermo Fisher Scientific Incorporation, Waltham, MA). Samples containing equal amounts of protein were mixed with Laemmli sample buffer and separated by sodium dodecyl sulfate-polyacrylamide gel electrophoresis (SDS-PAGE). Western blotting was then performed using nitrocellulose membranes and anti-cleaved caspase-3 (dilution 1:1000; cat. 9664 S; clone D175 (5A1E); lot. 13), anti-cleaved caspase-9 (dilution 1:1000; cat. 20750 S; clone Asp315 (D8I9E); lot. 1), anti-total IkB (dilution 1:000; cat. 4812 S; clone D44D4; lot. 6) or anti-phosphorylated IkB (dilution 1:000; cat. 2859 S; clone 14D4; lot. 8), anti-total p65 (dilution 1:000; cat. 8242S; clone D14E12XP(R); lot. 0004) or anti-phosphorylated p65 (dilution 1:1000; cat. 3033S; clone S536 (93H1); lot. 0014), or anti-βactin antibody (dilution 1:2000; cat. 4970 T; clone 13E5; lot. 18) from Cell Signaling (Danvers, MA), or mouse monoclonal anti-rat IgG2b (dilution 1:1000; cat. 106750; clone KT98 (HRP); lot. GR3324821-11) (abcam, Cambridge, UK), anti-rat IgG2a (dilution 1:1000; cat. A110; clone 109 P; lot. 34) (Bethyl Laboratories, Montgomery, TX) or anti-rat IgG, IgA, IgM (dilution 1:1000; cat.800-656-7625; clone #m612-103-130; lot. 36609) (Rockland Immunochemicals, Gilbertsville, PA), antibody. Band intensity on the blots was quantified using the public domain NIH ImageJ vs. 1.53e program (wayne@codon.nih.gov; Wayne Rasband, NIH, Research Service Branch).

**Expression of recombinant transglycosylase 351.** To prepare recombinant transglycosylase 351, the gene encoding the protein was synthesized using *Escherichia coli* optimized codons, amplified to add terminal A and cloned into the TA-cloning vector pGEM-T Easy (Promega, Madison, WI). We then excised the gene to clone into a modified pET28a vector, transformed into *E. coli* BL21 DE3 cells, and then expressed and purified as a 6-Histidine tagged (His-tag) protein[17].

**Preparation of antitransglycosylase polyclonal antibody.** Protein-A purified rabbit polyclonal antibody against the transglycosylase was developed by Eurofins Genomics (Tokyo, Japan) using the recombinant transglycosylase 351 prepared as described above.

**Preparation of monoclonal antibodies against the proapoptotic peptide IVMPESSGNPNAVNPAGYR.** The monoclonal antibodies were prepared at Eurofins Genomics Inc. (Tokyo, Japan). Briefly, the peptide used for immunization (NH2-C + IVMPESSGNPNAVNPAGYR-COOH) was synthesized, purified by high-performance liquid chromatography, and then conjugated to the keyhole limpet hemocyanin, a commonly used carrier protein, by the maleimide method[84,85]. For experimental animal immunization, two female Wistar rats (6 weeks of age) received one subcutaneous injection of the immunogen on the 0, 2nd, 4th, 6th week, and one intravenous injection of the same peptide 8th week after the first immunization. Blood samples were drawn on the 5th week after the first immunization to measure antibody titer by immunoassay. One rat's splenocytes were used for the cell fusion with myeloma cells (P3U1 and Sp2). The cell fusion procedure was performed by the polyethylene glycol method using the hypoxanthine-aminopterin-thymidine medium[86]. Cloning of monoclonal antibody-producing hybridomas was performed by limiting dilution[86]. Hybridomas were stored in liquid nitrogen for subsequent use. Isotyping was performed using Rapid RAT Monoclonal Antibody Isotyping XpressCard purchased from Antagen Pharmaceuticals, Inc. (Boston, MA).

**Epitope mapping of monoclonal antibodies against the proapoptotic peptide IVMPESSGNPNAVNPAGYR.** Mapping was performed at PEPperPRINT Incorporation (Heidelberg, Germany). Briefly, the peptide IVMPESSGNPNAVNPAGYR was first elongated with neutral GSGSGSG linkers at the C- and N-terminus to avoid truncated peptides. Then, the elongated peptide sequence was translated into 5, 10, and 15 amino acid peptides with peptide-peptide overlaps of 4, 9, and 14 amino acids. The resulting peptide microarrays contained 72 different linear peptides printed in duplicate (144 spots) and additional hemagglutinin (YPYDVPDYAG, 17 spots) control peptides. Pre-staining of a peptide microarray copy was done with secondary (goat anti-rat IgG [H + L] DyLight680) and control (mouse monoclonal anti-HA [12CA5] DyLight800) antibodies in incubation buffer (washing buffer [phosphate-buffered saline, pH 7.4 with 0.05% Tween 20 with 10% Rockland blocking buffer MB-070) to determine background interactions with the 72 different peptides of the microarray. Subsequent incubation of other peptide microarray copies with the rat monoclonal antibodies 9A, 21A, 2A, and 4A at a

concentration of 1 µg/ml in incubation buffer was followed by staining with secondary and control antibodies, and read-out at scanning intensities of 7/7 (red/green) using the LI-COR Odyssey Imaging System. The additional hemagglutinin peptides framing the peptide microarrays were simultaneously stained as internal quality control to confirm the assay quality and the peptide microarray integrity.

Quantification of spot intensities and peptide annotation were based on the 16-bit gray-scale tiff files at scanning intensities of 7/7 that exhibit a higher dynamic range than the 24-bit colorized tiff files. Microarray image analysis was performed using a PepSlide® Analyzer, and the results are available at https://zenodo.org/record/5803063#.YcWY-WDP2ck. A software algorithm breaks down fluorescence intensities of each spot into a raw, foreground, and background signal and calculates the averaged median foreground intensities and the spot-to-spot deviations of spot duplicates. Based on the averaged median foreground intensities, intensity maps were generated, and an intensity color code highlighted the interactions in the peptide maps with red for high and white for low spot intensities. We tolerated a maximum spot-to-spot deviation of 40%; otherwise, the corresponding intensity value was zeroed. We further plotted the assays' averaged spot intensities with the rat antibodies against the antigen sequence from the N- to the C-terminus of the peptide IVMPESSGNPNAVNPAGYR to visualize the overall spot intensities and signal-to-noise ratios. The intensity plots were correlated with the peptide and intensity maps and the microarrays' visual inspection to identify the epitopes of rat antibodies 9A, 21A, 2A, and 4A. If it was not clear if a particular amino acid contributed to antibody binding, the corresponding letters were depicted in gray color.

**Half-life of corisin in vitro.** We assessed the half-life of corisin in a cell culture system to determine the elimination rate of corisin from a solution in a cell culture system. A549 alveolar epithelial cells were cultured up to confluence in DMEM supplemented with 10% heat-inactivated FBS. The cells were then washed and cultured in FBS (−) DMEM. Corisin was then inoculated into the cell culture medium at a final concentration of 10 µg/ml, and the cell supernatant was sampled after 0, 1, 3, 6, 12, and 24 h. The concentration of corisin was measured by enzyme immunoassay, and the half-life of corisin was determined.

**Dose of synthetic corisin for in vivo experiments.** In our in vitro assays, corisin showed strong proapoptotic activity on A459 alveolar epithelial cells at a concentration of 10 µg/ml (5 µM). We used this concentration as a reference to estimate the approximate dose of corisin required to induce cell apoptosis and AE in disease mouse models. We used the following molarity equation and the molarity calculator software provided by GraphPad (https://www.graphpad.com/quickcalcs/molarityform/) to calculate the corisin mass:

$$M = C \times V \times MW$$

where M is the mass or amount of corisin in gram, C is the biologically active concentration of corisin in mol/L, V is the body weight of a mouse expressed in volume (1 g = 0.001 L), and MW is the molecular weight of corisin in Daltons. As the mean body weight of a TGFβ1 TG mouse is ~24.5 g (0.0245 L), the calculator provides the following result:

$$\text{Mass of corisin} = 0.000005 \, \text{mol/L} \times 0.0245 \, \text{L} \times 1974 \, \text{Daltons} = 0.0002408 \, \text{grams} = 240.8 \, \mu g$$

Based on this calculation, we performed a preliminary experiment using varying amounts of corisin. Mice were treated with 100, 150, and 300 µg of corisin dissolved in 50 µl of physiological saline by the intranasal route, and lung radiological changes were evaluated by CT 1 day after intranasal instillation. Lung radiological changes were detected only in mice receiving 300 µg of corisin.

**Treatment of AE of pulmonary fibrosis in TGFβ1 TG mice with anticorisin mAtb.** We performed a chest CT study in male TGFβ1 TG mice (bodyweight 26–29 g) with lung fibrosis and allocated them in two CT score-matched groups. We treated by intraperitoneal route a group of TGFβ1 TG mice with anticorisin mAtb clone 21A and another group with irrelevant antibody every 3 days five times before both groups received intratracheal instillation of corisin to induce AE. We performed CT before and after the corisin administration for comparison before sacrifice to take samples of blood, BALF, and the lungs. TGFβ1 TG mice without fibrosis that received only intratracheal corisin were the control mice. In a separate experiment, to evaluate the efficacy of the anticorisin mAtb to prolong survival, we administered BLM to TGFβ1 TG mice with lung fibrosis through subcutaneously implanted osmotic mini-pumps, treated them with the anticorisin mAtb or control antibody and followed the mouse survival. TGFβ1 TG mice without lung fibrosis treated with the anticorisin mAtb or control IgG were the controls.

**Induction of AE by intratracheal corisin in BLM-induced pulmonary fibrosis.** Lung fibrosis was induced in WT mice by infusing BLM through subcutaneous osmotic mini-pumps for 7 days. Lung fibrosis was confirmed by CT scanning, and mice were allocated into two groups with a matched CT score using in-house CT fibrosis score criteria. A group of mice received intratracheal corisin, and another group intratracheal scrambled peptide on day 20 during the fibrotic phase of BM-induced lung injury. The negative control mice were mice receiving saline through

subcutaneous osmotic mini-pumps and corisin or scrambled peptide by the intratracheal route.

**Intranasal administration of corisin at early stages of BLM-induced pulmonary fibrosis.** Lung fibrosis was induced in WT mice by infusing BLM (90 mg/kg mouse weight) through subcutaneous osmotic mini-pumps for 7 days. Mice receiving physiological saline in a similar manner were the controls. Mice with lung fibrosis and control mice were allocated into two groups. A group of mice with lung fibrosis received intranasal synthetic corisin (100 µg in 40 µl of physiological saline), and another group scrambled peptide (100 µg in 40 µl of physiological saline) on days 3, 5, 7, 9, 11 after starting BLM infusion. Control mice receiving saline through subcutaneous mini-pump were also allocated into two groups and treated with corisin or scrambled peptide in a similar manner. All groups were sacrificed on day 22 of BLM or saline treatment.

**Longitudinal changes of circulating native corisin during BLM-induced pulmonary fibrosis.** Mice were infused BLM ($n = 5$) or saline ($n = 4$) through osmotic mini-pumps for 7 days, and then blood samples were collected from the tail vein on days 0, 3, 6, 11, 14, 18, 21, and 25 to measure the concentration of corisin by enzyme immunoassay as described below.

**Treatment of BLM-induced lung fibrosis with anticorisin mAtb.** To induce lung injury and fibrosis, we dissolved BLM in physiological saline and administered it to WT mice at a dose of 100 mg/kg of mouse weight (total volume 200 µL) by constant (7 days) subcutaneous infusion through osmotic mini-pumps[87]. The osmotic mini-pumps deliver their contents at 0.5 µL/h for 7 days. Mice receiving an infusion of saline through subcutaneous osmotic mini-pumps were used as controls. To evaluate whether anticorisin mAtb can block BLM-induced lung fibrosis, mice receiving BLM were treated with anticorisin mAtb (WT/BLM/anticorisin) or control IgG (WT/BLM/control IgG) at a dose of 20 mg/kg of mouse weight by intraperitoneal route, three times a week during 3 weeks. Mice receiving saline (SAL) by osmotic mini-pumps and treated with anticorisin mAtb (WT/SAL/anticorisin) or antibody isotype (WT/SAL/control IgG) at a dose of 20 mg/kg of mouse weight by intraperitoneal route, three times a week during 3 weeks were used as control mice. After performing chest CT, all mouse groups were sacrificed on day 22 of BLM infusion.

**Assessment of pulmonary fibrosis.** We anesthetized the mice to collect BALF for biochemical analysis and cell counting. Briefly, we cannulated the trachea with a 20-gauge needle and infused saline solutions into the lungs[88]. After centrifugation, the supernatants were stored at −80 °C until analysis, and the cell pellets were resuspended in physiological saline solution to count the number of cells and determine the differential cell count. We used the nuclei counter ChemoMetec (Allerød, Denmark) to determine the total cell count and stained the cells with May–Grünwald–Giemsa (Merck, Darmstadt, Germany) to count differential cells. We then euthanized the mice by an overdose of anesthesia before resecting the lungs to fix in formalin, embedd in paraffin, and prepare for hematoxylin-eosin and Masson trichrome staining. We used the Olympus BX50 microscope with a Plan objective combined with an Olympus DP70 digital camera (Tokyo, Japan) to evaluate histopathological findings. The grade of lung fibrosis was assessed using the following parameters: (1) CT fibrosis score using a score system[17], (2) Ashcroft fibrosis score using lung tissue stained with hematoxylin & eosin[71,89], (3) Masson trichrome (or collagen) stained area expressed in percentage using the WinRoof image processing software[71], and (4) lung hydroxyproline content measured using a colorimetric assay[71]. For the Ashcroft fibrosis score, microphotographs of five microscopic fields from the lung of each mouse were taken at random, eight blinded observers scored the grade of fibrosis, and the mean fibrosis score for each mouse lung was calculated[71,89]. For evaluating the Masson trichrome stained area, an investigator unaware of the treatment group took microphotographs of five microscopic fields from the lung of each mouse at random and measured the trichrome stained area using the WinRoof image processing software (Mitani Corp., Fukui, Japan).

**Treatment of lipopolysaccharide (LPS)-induced acute lung injury with anticorisin mAtb.** We allocated WT 8-week male C57BL6 mice (26–30 g of weight) into two weight-matched groups. We then treated one group with anticorisin mAtb and another group with control IgG at a dose of 20 mg/kg by intraperitoneal route once a day every other day for a week. Chest CT was performed 1 day before and after the intratracheal instillation of 150 µg of LPS in a volume of 75 µl. We performed euthanasia of mice on the second day after intratracheal instillation of LPS to sample BALF, blood, and both lungs for biochemical and histopathological studies.

**Induction of acute lung injury by intratracheal delivery of BLM.** WT 8-week male C57BL6 mice (25 of weight) were separated into two groups. One group was treated with anticorisin mAtb and another group with control IgG at a dose of 20 mg/kg by intraperitoneal route and received intratracheal instillation of BLM (1.5 mg/kg mouse weight) dissolved in 75 µL of saline. Untreated mice receiving intratracheal instillation of saline (75 µL) were the control group. Mice of each

group were sacrificed on the seventh day to remove the lungs. Apoptosis of lung cells was evaluated by TUNEL assay performed at MorphoTechnology Corporation (Sapporo, Hokkaido, Japan).

**Intratracheal instillation of *Staphylococcus haemolyticus* and *Staphylococcus epidermidis* in germ-free TGFβ1 TG mice.** Oral gavage (200 µl) of a solution containing a cocktail of antibiotics including vancomycin (0.5 mg/ml), neomycin (1 mg/ml), ampicillin (1 mg/ml), metronidazole (1 mg/ml), and gentamycin (1 mg/ml) was administered once a day for 4 days to three groups of TGFβ1 TG mice. The CT scores of lung fibrosis of all mice were matched. On the 5th day, one group of mice received intratracheal instillation of $1 \times 10^8$ colony forming units (75 µl) of *Staphylococcus haemolyticus* strain 12 or *Staphylococcus epidermidis* ATCC14990 and was sacrificed after 2 days. Germ-free TGFβ1 TG mice treated with 0.9% NaCl solution were the controls.

**Biochemical analysis.** The total protein level was measured by a dye-binding assay using a commercial kit from ThermoFisher Scientific Incorporation (Waltham, MA). The levels of TGFβ1 (R&D Systems, Minneapolis, MN), TNFα (BD Bioscience, BD opt-EIA kits, San Diego, CA), and SP-D (Sino Biologicals, Beijing, China), SP-C (Abcam, Cambridge, UK) were measured using commercial enzyme immunoassay kits following the manufacturer's instructions. Collagen type I was measured by enzyme immunoassay using anti-collagen type I antibody and anti-collagen type I biotin-conjugated antibody from Rockland Immunochemicals Incorporation (Limerick, PA), and serum amyloid P component, MMP-1, MUC-1, CCL2, CCL3, CXCL1, CXCL2, MUC5B, and LDHA were measured by commercially available enzyme immunoassay kits following the manufacturer's instructions. The lungs' hydroxyproline content was measured by a colorimetric method using a commercial kit (Hydroxyproline colorimetric assay kit, BioVision, San Francisco, CA) following the instructions of the manufacturer.

**Measurement of corisin.** We developed an enzyme-linked immunoassay using antitransglycosylase 351 polyclonal antibody as a capture antibody and biotinylated anticorisin 9A mAtb as a detection antibody. Briefly, the capture antibody was coated on a 96-well plate at a final concentration of 2 µg/ml in phosphate-buffered saline at 4 °C overnight. After blocking with 1% bovine serum albumin in phosphate-buffered saline and appropriate washing with phosphate-buffered saline in Tween, varying concentrations of corisin as standards and plasma samples were added to the wells and incubated at 4 °C overnight. The wells were then washed before adding horseradish peroxidase-conjugated streptavidin (R&D System) in a phosphate-buffered saline solution. After appropriate washing and incubation, substrate solution was added for color development, and absorbance was read at 450 nm. Values were extrapolated from a standard curve drawn using several concentrations of corisin.

**Measurement of the half-life of anticorisin neutralizing antibody in plasma.** Female TGFβ1 TG mice without fibrosis with the same genetic background (C57BL/6 J) as the wild-type mice were used in the experiment. Mice received an intraperitoneal injection of (20 mg/kg) anticorisin mAtb (clone 21A), and blood was sampled after 3, 6, 24, 48 168, 336, and 504 h. We used a corisin-based immunoassay to monitor the plasma concentrations of the anticorisin neutralizing antibody clone 21A. Briefly, we coated 96-well plates with 2 µg/ml corisin in phosphate-buffered saline (PBS) by overnight incubation. After blocking with 1% bovine serum albumin (BSA) in PBS and appropriate washing in PBS, we added diluted (1:1000) plasma samples and standard of the anticorisin mAtb (varying concentrations between 100 and 0.8 ng/ml) to the plates and incubated overnight at 4 °C. The plate wells were then washed before incubation in the presence of horse peroxidase-labeled anti-rat IgG. After appropriate washing with PBS, color development was performed using 3,3′,5,5′-tetramethylbenzidine. The plasma concentration was extrapolated from the curve drawn using the standard antibody levels. We expressed the plasma concentration of the anticorisin neutralizing antibody as a percent remaining in the circulation at different intervals.

**Gene expression analysis.** Total RNA from cells or lung tissue were extracted using Sepasol RNA-I Super G reagent (Nacalai Tesque Inc., Kyoto, Japan), synthesized cDNA from 2 µg of total RNA with oligo-dT primer and ReverTra Ace Reverse Transcriptase (Toyobo Life Science Department, Osaka, Japan) and then performed standard PCR using primers (Supplementary Table 2). PCR was performed with 26 to 35 cycles depending on the gene, denaturation at 94 °C for 30 s, annealing at 65 °C for 30 s, elongation at 72 °C for 1 min followed by a further extension at 72 °C for 5 min. The expression of mRNA was normalized against the glyceraldehyde 3-phosphate dehydrogenase (GAPDH) mRNA expression.

**Statistical analysis.** We expressed the data as the mean ± standard deviation of the means (S.D.). The statistical difference between the two variables with normal distribution was assessed by the two-tailed unpaired Student's $t$ test and the difference between three or more variables with normal distribution by one-way analysis of variance (ANOVA) using the Newman–Keuls' test for post hoc analysis. Variables with skewed distribution were evaluated by the two-tailed

Mann–Whitney *U* test or the Kruskal–Wallis ANOVA with Dunn's test. *P* value < 0.05 was considered statistically significant. We used GraphPad Prism vs. 7 (GraphPad Software, Inc., San Diego, CA) to perform the statistical analysis.

**Reporting summary**. Further information on research design is available in the Nature Research Reporting Summary linked to this article.

## Data availability

The authors declare that all data supporting this study's findings are available within this paper and its Supplementary Files. The complete genome sequences of the bacteria designated *S. haemolyticus* strain 1 (accession No: CP071512-CP071515), *S. haemolyticus* strain 7 (accession No: CP071508-CP071511) and *S. haemolyticus* strain 12 (accession number: CP071505-CP071507) have been deposited at the Genbank database (https://www.ncbi.nlm.nih.gov/genbank/). All raw data and sequences are also available at the open-access repository Zenodo (https://zenodo.org/record/5803063#.YcWY-WDP2ck). All datasets generated during and/or analyzed during the current study are also available from the corresponding authors on reasonable request. Source data are provided with this paper.

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

## Acknowledgements

This research was supported in part by the Japan Society for the Promotion of Science (Kakenhi No 17K08442 and 18K08175), in part by a grant from Takeda Science Foundation (2019), Kowa Life Science Foundation (2021), Japan Intractable Diseases (Nanbyo) Research Foundation (2021), GSK Japan Research Grant 2021, and in part by funding support for the Microbiome Metabolic Engineering Theme (Carl R. Woese Institute for Genomic Biology), by the College of Agricultural, Consumer and Environmental Sciences Office of International Programs, the University of Illinois at Urbana-Champaign, and a gift from the Charles and Margaret Levin Family Foundation. The funders had no role in study design, data analysis, decision to publish, or paper preparation.

## Author contributions

C.N.D.G.: preparation of disease model, preparation of the first paper draft. A.M.: evaluation by transmission electron microscope. A.Ta., K.N., Y.Y.: analysis of signal pathways and protein expression. I.C., R.I.M., A.M.A-H., M.S.: preparation of recombinant proteins, bacterial culture, genome analysis, chromatography. T.K., H.F., O.H., T.O., A.To., H.N., K.K., Y.K., M.H., T.A., Y.I.,: clinical evaluation of subjects, provision of clinical data, and samples. M.T., T.Y., V.F.D., Y.O., H.S.: flow cytometry analyses, gene, and protein expression analysis, computed tomography analysis. R.O., J.O., M.F., T.N.: biological activity of peptides. X.M., D.S.: X.M., D.S.: interpretation of the data and intellectual contribution. I.C., E.C.G.: study design, analysis, and interpretation of the data, preparation, and correction of the paper.

## Competing interests

E.C.G. and T.K. have a patent on the TGFβ1 TG mice used in the present study. In addition, there is an invention disclosure by C.N.D.G., E.C.G., and I.C. on the apoptotic peptides identified in this study and anticorisin mAtbs developed for the treatment of pulmonary fibrosis described in this study. None of the other authors declared any competing interests regarding the present work.
