## [Peer Review File · Nature Communications]

REVIEWER COMMENTS

Reviewer #1 (Remarks to the Author):

The role of epithelial pathways in the initiation of IPF is not questioned. The authors extrapolate from this to argue that epithelial pathways may be central to IPF progression. However, whilst it may be correct that initiation and progression pathways are synonymous, this has yet to be established in IPF. In malignancy, for example, many cancers arise in the epithelium but progressiveness is generally governed by stromal pathways. Whether progression in IPF is primarily driven by the epithelium and related factors or by fibroblast or connective tissue matrix pathways is uncertain. Therefore, some caution would seem to be indicated in the statement of the hypothesis underpinning the current work.

If one accepts the “big picture” pathogenetic statement made by the authors referred to above, an exploration of microbiome pathogenesis appears logical. Whether microbiome content is a primary driver or is a secondary phenomenon has not been teased out but that “chicken/egg” caveat is ubiquitous in the exploration of pathogenetic pathways in IPF. The demonstration of linkage between a pro-apoptotic peptide expressed in the IPF microbiome and progressive pulmonary fibrogenesis is, thus, an important piece of ancillary information.

The authors, have a) identified a pathway for activation of the pro-apoptotic peptide; b) established that coriasin activates e.g. apoptosis; c) identified monoclonal antibodies inhibiting coriasin and associated peptides; d) established in two pulmonary fibrosis models that antibodies inhibit AE and thus ameliorate [ulmonary fibrosis; and e) show this also in a general model of lung injury.

The narrative is detailed and impressive. Once again, there is a note of caution: this work applies specifically to acute exacerbations and this is highlighted by the applicability to a general lung injury model. This extension of the current work was extremely interesting. It should be stressed, though, that acute lung injury is, in general (and in IPF), very likely to be driven by epithelial pathways – this current work has not been applied to chronic progression. Agents shown to attenuate chronic progression in IPF and other progressive fibrotic ILDs (pirfenidone and nintedanib) have not been shown, as yet, to suppress acute exacerbations. This again suggests that it may not be valid to start by discussing disease progression in general in IPF, but to centre the hypothesis on acute exacerbations.

On the first step, the authors show that their pathway is applicable to a broad group of pathogens. The argument would be more complete if they had shown that similar pathways do not exist in microbiota within control subjects and are, therefore, specific to the IPF microbiome. Ideally, one would like to see a coriasin activity profile related to the IPF microbiome that is not expressed in the normal microbiome. It might be useful for the authors to advance arguments rebutting this reservation.

Activation of apoptotic pathways appears broadly convincing.

The process of identifying antibodies is validated by their efficacy in animal models.

It should again be stressed that the models were AE/lung injury models and that “immature fibrosis” is a feature of various forms of lung injury, remodelling with time in humans. A very topical example is “ILD” post-COVID. It is clear that “fibrotic-like” abnormalities on CT post COVID (including traction bronchiectasis) regress with time in a substantial majority of patients. Therefore, the inhibition of pulmonary fibrosis in AE and acute lung injury models is not necessarily relevant to chronic progression in IPF and other fibrotic lung diseases. The effects the authors have quantified are not necessarily synonymous with putative effects in chronic progression.

In summary, this reviewer is intrigued by the potential applicability of this narrative to acute lung injury in IPF and other settings. However, I am not persuaded that there is the same force of argument with reference to chronic progression. It would seem advisable to reframe the hypothesis along these lines: AE-IPF, and lung injury in general, are in great need of novel treatment approaches and these data are amply justified by this goal.

Reviewer #3 (Remarks to the Author):

The manuscript entitled “Inhibition of a lung microbiota-derived peptide ameliorates acute exacerbation of pulmonary fibrosis.” by D’Alessandro-Gabazza et al. uses two recognized models of murine pulmonary fibrosis, i.e., TGFb overexpression and bleomycin induced fibrosis as well as a model of LPS induced acute lung injury to study the effects of a microbiology derived peptide ‘Corisin’ in lung disease. The authors provide evidence that Corisin is produced by proteolytic degradation of bacterial transglycosylase produced in several commensal staphylococcus species and causes epithelial cell apoptosis through activation of the intrinsic caspase-9 mediated pathway. Further, evidence is provided that Corisin treatment results in exacerbation of pulmonary fibrosis and lung injury. The manuscript details the production of a neutralizing anti-Corisin monoclonal antibody that blocks the deleterious effects of Corisin and protects from acute exacerbation. Overall, the article is well written, although somewhat lengthy, and although Corisin was characterized previously by this same group, the detailing of the anti-Corisin antibody and its biologic effects are assuredly novel. I list my comments and concerns about the manuscript below.

My major concern is with the discrepancy between the two fibrotic models used and the timing of the Corisin / anti-corisin treatments as well as the inclusion of an acute lung injury model. I feel that, while the inclusion of all of these models was interesting, the manuscript suffered from a lack of mechanistic insight into any particular model. Further, the models all used different treatment regimens and timings for the delivery of Corisin / anti-Corisin. For instance, in the TGBb model, mice with existing fibrosis were either given Corisin alone to induce what the authors describe as an acute exacerbation of fibrosis or Corisin and the anti-Corisin antibody together which was shown to reduce the exacerbation. However, in the BLM model, the authors used two separate experiments, one where Corisin was given alone and one where the anti-Corisin antibody was given very early during BLM administration before fibrosis is established. Further, the LPS injury model used an entirely different dosing scheme where mice were pretreated with anti-Corisin antibody and then treated

with LPS. Unfortunately, this makes it difficult to deduce the mechanism for how Corisin maybe be inhibiting lung injury / fibrosis. The authors argue that Corisin causes / exacerbates epithelial cell apoptosis and that the anti-corisin antibody neutralizes this effect. However, the effects of early Corisin treatment also greatly diminish the infiltration of inflammatory cells into the lungs and thus effects on inflammatory pathways, such as TLR signaling, are suggested.

1) I feel the authors should examine the effects of Corisin on inflammatory TLR signaling, especially TLR4, as LPS instillation was used in the manuscript. This can be done in epithelial cells and/or cultured macrophages looking at inflammatory cytokines / chemokine output as an endpoint. Because neutrophils were shown to be reduced especially in the LPS model I would be interested to see the effect of Corisin treatment on the induction of neutrophil specific chemokines such as CXCL1 and CXCL2. Other genes of interest would be CCL2 and CCL12 which have both shown to be important for the pathogenesis of BLM induced fibrosis.

2) All of the in vitro assays showed a mild but statistically significant increase in epithelial cell apoptosis when using Corisin alone, however, Corisin treatment alone is insufficient to induce any pathology when administered to mice. Do the authors have an explanation for this seemingly incongruent observation? Would epithelial cell apoptosis in response to Corisin be augmented if the cells were pre-treated with Bleomycin or another apoptosis inducing agent in vitro?

3) Lung tissue associated Corisin was quantified in Fig. S29C. What time point was this measurement taken at? If it was 22days after BLM like the rest of figure s25 then it would appear that early neutralization can inhibit the production of Corisin throughout the BLM timecourse but only results in a modest decrease in the amount of cleaved caspase3 while showing a much larger reduction in hydroxyproline.

a. Does neutralization of Corisin have a greater effect on cell apoptosis early after BLM instillation (7 – 14 days)? Similarly, did the authors explore early delivery of exogenous Corisin (0 – 14 days) and its effect on late fibrosis?

b. I would suggest using the Corisin ELISA throughout a time course of BLM instillation to determine when Corisin concentrations are highest. It seems the Authors are suggesting that the effects are most apparent in the late stages of fibrosis, however, neutralization of Corisin immediately after BLM instillation rescued fibrosis suggesting that Corisin is most important during the early stage, i.e. inflammatory phase, of fibrosis.

c. Why does treatment with anti-Corisin antibody show a decrease in the amount of detectable Corisin only after BLM instillation (Fig. S29C) when there is a demonstrable amount of Corisin present in the lungs even in saline treated mice?

d. Even 300ug of purified Corisin resulted in only modest increases in cell apoptosis and fibrosis area in the lungs of BLM treated mice. Is the amount of exogenous Corisin given in figures 4 and 5 physiologically relevant given that there is likely sub-microgram levels of Corisin present in an entire mouse lung?

4) I am curious as to why the Authors chose a multiple comparison statistical method for the comparisons in figure 4 (ANOVA, 3 groups compared) but chose to do a single comparison method in figure 5 (T-test, 4 groups split into two comparisons). Figure 5 makes it very difficult to determine if the group of mice treated with BLM are significantly different in comparison to the saline controls. I would strongly suggest using a multiple comparison method such as ANOVA on this figure.

5) Figure 5g and 6j show relatively small increases / decreases in the area of fibrosis between treated and control groups, however, histology images in 5g and 6i seem to show massive changes in lung architecture. Are these images representative? I would suggest showing lower power magnifications or choosing images that accurately reflect the quantified data. Furthermore, please add the magnification and scale bar information for the histology figures to either the materials and methods or the figure legend.

6) The amount of data presented, especially, supplementary data is excessive and needs to be trimmed substantially. For instance, figures S19 through S25 are largely redundant with figure 5 and 6 in the manuscript. If the Authors could pick just the most important data from these and create a new supplemental figure it would substantially improve manuscript readability.

7) Figure 9, while a nice graphical depiction of how lung fibrosis occurs does not really illustrate what Corisin might be doing to alleviate acute exacerbations. Many of the things displayed in this figure are not explicitly covered in the manuscript while the topics of the manuscript, such as the proteolytic degradation of bacterial proteins, are not. This figure would be well suited for a review on fibrosis / acute exacerbations, but I would like to see a more mechanistic depiction of how Corisin is exacerbating fibrosis here.

The manuscript by D'Alessandro-Gabazza et al. reported on follow-up studies on corisin, a microbiota peptide that induces apoptosis of alveolar epithelial cells and triggers acute exacerbation of pulmonary fibrosis, that the same group published in NatComm 2020. They provided additional details on how corisin is processed and released by *Staphylococcus* spp. and the protective effect of anti-corisin monoclonal antibodies using mouse models of acute exacerbation of pulmonary fibrosis.

Specific Comments for the Author:

1. The abstract contains a long, speculative statement about COVID-19 and SARS even though the authors did not perform any such studies in their submitted manuscript. This should be deleted.
2. The link between corisin and AE-IPF was established principally using synthetic corisin. The 300 microgram/mouse of corisin used to induce AE-IPF seems like an enormous amount, considering actual amounts measured in mouse lungs in Sup Fig 29c. Accordingly, it is not clear whether the amounts of corisin used in the mouse studies and in the in vitro studies (e.g. with A549) are biologically relevant, or whether enormous amounts of corisin—which the microbiota cannot possibly produce—was used. The authors should quantify amounts of corisin—using ELISA that they described in the present manuscript—in patient BALF samples to determine a biologically relevant concentration of corisin that could be used in mouse and in vitro studies.
3. Notwithstanding the preceding comments about biologically relevant amounts of synthetic corisin, the authors should provide more definitive mouse and in vitro experiments about the roles of corisin in AE-IPF using isogenic bacteria:
 - i. *S. haemolyticus* wild-type
 - ii. isogenic mutant deficient in production/release of corisin
 - iii. complemented strain
4. The authors speculated on a putative serine protease that was needed to cleave the transglycosylase to release corisin. Because the authors sequenced the genomes of several *S. haemolyticus* strains, why not make use of this information to generate deletions of genes encoding putative serine proteases to identify the specific serine protease involved in this crucial step. Half of the manuscript seems to be about understanding better the mechanisms of corisin processing and release, so it is unsatisfying that the protease involved in this process was ultimately not identified.
5. Corisin concentration was quantified in mouse plasma and lung samples (Supplemental Figure 29) but this was not done for human plasma and BALF samples. To further support the authors' hypothesis that "plasma level of corisin is a potential biomarker of pulmonary fibrosis" (line 472), the authors should use their newly developed immunoassay to measure concentrations of corisin in patient plasma and BALF samples.
6. Severity of pulmonary fibrosis was evaluated by CT scan using apparently five different methods:

- i. CT opacity area (%)—e.g., Sup Fig 27b
- ii. CT area of fibrosis (%)—e.g., Fig 4
- iii. Ashcroft score—e.g., Fig 6h
- iv. CT score—e.g., Sup Fig 20
- v. CT fibrosis score—e.g. Sup Fig 25

It is not clear whether (i) opacity area and (ii) area of fibrosis are one and the same but were labeled differently, or they were indeed different scoring methods. More importantly, the five different methods of scoring the CT scans were not applied consistently for all studies (e.g. only Ashcroft score and fibrosis area (%) was presented in Sup Fig 25). Please clarify in the methods section why four different scoring methods are needed/used and why were they not used consistently in all studies. Did the authors selectively show only results with a particular scoring method because it produced statistical significant results? It would be important to fully disclose the data using the other scoring methods even when the other methods did not yield statistical significance.

7. Related to the above comment regarding five different methods for scoring CT scan, in the mouse model of bleomycin-induced pulmonary fibrosis (Sup Fig 23), CT scan was performed on day 9 and day 21. CT opacity area was provided for day 9 but not day 21 (Sup Fig 23), whereas CT fibrosis score was provided for day 21 but not day 9 (Sup Fig 24). Also, regarding Sup Fig 24, it seems that day 21 CT scan was mislabeled as day 22 CT scan (the mislabel would otherwise be in the timeline provided in Sup Fig 23a showing day 21 CT scan). Another example of inconsistent scoring would be that Fig 4 shows for the primary outcome only area of fibrosis whereas Fig 6 shows both area of fibrosis and Ashcroft score.
8. WT mice and TGFβ1 TG mice were treated with anticorisin neutralizing mAb 21A three times a week, and this was based on the half-life of the anticorisin mAb in plasma from TGFβ1 TG mice was shown in Sup Fig 16. Would the authors expect the half-life anticorisin mAb in plasma in TGFβ1 TG mice different from in WT mice?
9. In their pivotal studies involving treatment with anti-corisin IgG vs. isotype (Fig. 4e), differences between the two groups appeared to be small and not statistically significant. Using raw data provided by the authors in the supplementary excel file, the Reviewer performed the statistical analysis shown below. With a small sample size of 6 animals per experimental group and a small effect size of only 2.54% difference in area of fibrosis, the difference was not significant.

```
. ttest var1, by(var2)
Two-sample t test with equal variances
```

Group	obs	Mean	Std. Err.	Std. Dev.	[95% Conf. Interval]	
0	6	6.189167	1.051457	2.575533	3.486311	8.892022
1	6	8.731333	.9551523	2.339636	6.276036	11.18663
combined	12	7.46025	.7781298	2.695521	5.747598	9.172902
diff		-2.542167	1.42052		-5.707283	.6229494

```
diff = mean(0) - mean(1)          t = -1.7896
Ho: diff = 0                      degrees of Freedom = 10
Ha: diff < 0                      Ha: diff != 0                      Ha: diff > 0
Pr(T < t) = 0.0519                 Pr(|T| > |t|) = 0.1038                 Pr(T > t) = 0.9481
```

The Reviewer strongly suggests that the authors should work with a statistician to ensure that all statistical analyses were performed rigorously, and more importantly, that

the appropriate sample size (e.g. number of mice per experimental group) was pre-determined and appropriate to detect small effect size. The Reviewer also suggests that the Editor consults with a statistician to review these data. More importantly, the authors should discuss the biological significance of a 2.5% difference in area of fibrosis and comment on whether this is clinically meaningful.

Other minor comments:

1. Line 448: Supplementary fig. 37 ???→ 27
2. Supplementary Fig. 22 legend “sacrificed on day **21**”
3. Line 396-397: We performed a chest CT study before sacrifice on day **22** after BLM or SAL infusion: mice was sacrificed on day **21** but panel 22a shows day **22**.
4. Tissue concentrations of proteins (e.g. corisin) should be expressed as ng/mg of tissue (or ng/lung) and not as ng/mL (e.g. Supplemental Figure 29b).

Response to Queries of Reviewer #1

We thank the reviewer for her/his time and appreciate the comments which were very constructive and have helped to improve the interpretation and the quality of our manuscript.

Comment 1:

The role of epithelial pathways in the initiation of IPF is not questioned. The authors extrapolate from this to argue that epithelial pathways may be central to IPF progression. However, whilst it may be correct that initiation and progression pathways are synonymous, this has yet to be established in IPF. In malignancy, for example, many cancers arise in the epithelium but progressiveness is generally governed by stromal pathways. Whether progression in IPF is primarily driven by the epithelium and related factors or by fibroblast or connective tissue matrix pathways is uncertain. Therefore, some caution would seem to be indicated in the statement of the hypothesis underpinning the current work.

If one accepts the “big picture” pathogenetic statement made by the authors referred to above, an exploration of microbiome pathogenesis appears logical. Whether microbiome content is a primary driver or is a secondary phenomenon has not been teased out but that “chicken/egg” caveat is ubiquitous in the exploration of pathogenetic pathways in IPF. The demonstration of linkage between a pro-apoptotic peptide expressed in the IPF microbiome and progressive pulmonary fibrogenesis is, thus, an important piece of ancillary information.

The authors, have a) identified a pathway for activation of the pro-apoptotic peptide; b) established that corisin activates e.c. apoptosis; c) identified monoclonal antibodies inhibiting corisin and associated peptides; d) established in two pulmonary fibrosis models that antibodies inhibit AE and thus ameliorate pulmonary fibrosis; and e) show this also in a general model of lung injury.

The narrative is detailed and impressive. Once again, there is a note of caution: this work applies specifically to acute exacerbations and this is highlighted by the applicability to a general lung injury model. This extension of the current work was

extremely interesting. It should be stressed, though, that acute lung injury is, in general (and in IPF), very likely to be driven by epithelial pathways – this current work has not been applied to chronic progression. Agents shown to attenuate chronic progression in IPF and other progressive fibrotic ILDs (pirfenidone and nintedanib) have not been shown, as yet, to suppress acute exacerbations. This again suggests that it may not be valid to start by discussing disease progression in general in IPF, but to centre the hypothesis on acute exacerbations.

Response:

Following the suggestions of Reviewer #1 we have made changes and added statements in the revised manuscript to show that our present study is focused on the acute exacerbation phase rather than the progressive phase of IPF.

**Please see (pdf file),
page 4, lines 77 to 79:**

“..... . Acute exacerbation of idiopathic pulmonary fibrosis is associated with high mortality. Excessive apoptosis of lung epithelial cells occurs in pulmonary fibrosis acute exacerbation.....”

page 4, lines 86 to 89:

“.....These results underscore the role of corisin in the pathogenesis of acute exacerbation of pulmonary fibrosis and acute lung injury and provide a novel approach to treating this incurable disease.....”

page 6, lines 133 to 135:

“.....Therefore, we hypothesized that treatment with a monoclonal antibody against corisin would directly inhibit or ameliorate the AE of pulmonary fibrosis (AE-IPF) by blocking the apoptotic activity of corisin.....”

pages 7, lines 150 to 152:

“.....These results underscore the role of corisin and corisin-like peptides in the pathogenesis of AE in pulmonary fibrosis and perhaps other acute lung injury-

associated diseases and provide a novel approach to treating this incurable disease.....”

page 30, lines 736 to 738:

“..... Overall, these beneficial effects of the neutralizing anticorisin mAtb in vivo indicate that corisin plays a causative role in AE, thereby pointing to corisin as a potential molecular target for the therapy of this devastating complication of IPF..”

page 32, lines 792 to 794:

“.....As corisin plays an active role in AE of pulmonary fibrosis, we posit that the level of corisin would be a marker of fibrosis.....”

Comment 2:

On the first step, the authors show that their pathway is applicable to a broad group of pathogens. The argument would be more complete if they had shown that similar pathways do not exist in microbiota within control subjects and are, therefore, specific to the IPF microbiome. Ideally, one would like to see a coriasin activity profile related to the IPF microbiome that is not expressed in the normal microbiome. It might be useful for the authors to advance arguments rebutting this reservation.

Response:

We very much appreciate the insightful suggestions of the reviewer. We also thought that the corisin activity profile would be specific to IPF (absence in control subjects) but to our surprise we found detectable levels of corisin in body fluid (BALF) from healthy subjects as well. From this observation in healthy subjects, we got the hint to evaluate the potential role of corisin in other tissue injury-associated lung diseases such as acute lung injury or ARDS. It is very possible that the microbial community producing corisin in the IPF lung (or the sequence of the peptides) differs from what we observed in healthy subjects. Although we currently have no data to support this assumption, in our earlier manuscript, we present alignment of transglycosylases containing corisin-like peptides. The alignment shows a large diversity of the peptides at the corisin position, and our analysis showed variation of apoptotic activity on alveolar epithelial cells, although

based on the amino acid sequences they may cross-react with antibodies developed against corisin. The reviewer's insightful question is deserving of further analyses, and we look forward to clearly addressing this in future studies by determining the peptide sequences present in healthy patients and their apoptotic activities compared with peptides enriched in IPF patient.

We acknowledged in the revised manuscript the lack of evidence on whether the corisin-producing microbiota is similar or different in IPF and normal subjects, and the way we got the hint to explore the role of corisin in other tissue injury-associated lung diseases.

Please see page 30, lines 741 to 747:

“.....Whether the corisin-releasing lung microbiota of IPF patients or their peptides are similar to microbiota of normal subjects is currently unknown. However, the detectable levels of corisin in BALF from healthy subjects and in tissue from mice with no fibrosis, albeit at lower levels, suggest that the microbiota releases corisin (or corisin-like peptides) even under physiological conditions¹⁷. Therefore, the involvement of corisin in other diseases associated with lung injury, such as acute respiratory distress (ARDS) and other diseases is predictable^{48,49}.....”

Comment 3:

Activation of apoptotic pathways appears broadly convincing.

The process of identifying antibodies is validated by their efficacy in animal models.

It should again be stressed that the models were AE/lung injury models and that “immature fibrosis” is a feature of various forms of lung injury, remodelling with time in humans. A very topical example is “ILD” post-COVID. It is clear that “fibrotic-like” abnormalities on CT post COVID (including traction bronchiectasis) regress with time in a substantial majority of patients. Therefore, the inhibition of pulmonary fibrosis in AE and acute lung injury models is not necessarily relevant to chronic progression in IPF and other fibrotic lung diseases. The effects the authors have quantified are not necessarily synonymous with putative effects in chronic progression.

In summary, this reviewer is intrigued by the potential applicability of this narrative to acute lung injury in IPF and other settings. However, I am not persuaded that there is

the same force of argument with reference to chronic progression. It would seem advisable to reframe the hypothesis along these lines: AE-IPF, and lung injury in general, are in great need of novel treatment approaches and these data are amply justified by this goal.

Response:

We thank again the stimulating comment of the Reviewer. As we described above under “Response” Comment 1, we avoided implicating corisin in chronic progressive stages of IPF and focused on the potential role of corisin in acute exacerbation of pulmonary fibrosis.

Please see again the description above under “Response” to Comment 1 of Reviewer #1.

Response to Queries of Reviewer #2

Comment 1:

The manuscript by D’Alessandro-Gabazza et al. reported on follow-up studies on corisin, a microbiota peptide that induces apoptosis of alveolar epithelial cells and triggers acute exacerbation of pulmonary fibrosis, that the same group published in NatComm 2020. They provided additional details on how corisin is processed and released by Staphylococcus spp. and the protective effect of anti-corisin monoclonal antibodies using mouse models of acute exacerbation of pulmonary fibrosis.

Response

We very much appreciate the insightful and constructive comments of the Reviewer that have helped to improve the quality of the revised version of the manuscript.

Comment 2:

The abstract contains a long, speculative statement about COVID-19 and SARS even though the authors did not perform any such studies in their submitted manuscript. This should be deleted.

Response:

As suggested, we have deleted the sentences pointed out by the Reviewer, and instead replaced them by the conclusion of the study.

Please see page 4 (pdf file), lines 86 to 89 in the revised manuscript.

Comment 3:

The link between corisin and AE-IPF was established principally using synthetic corisin. The 300 microgram/mouse of corisin used to induced AE-IPF seems like a enormous amount, considering actual amounts measured in mouse lungs in Sup Fig 29c. Accordingly, it is not clear whether the amounts of corisin used in the mouse studies and in the in vitro studies (e.g. with A549) are biologically relevant, or whether enormous amounts of corisin—which the microbiota cannot possibly produced—was used. The authors should quantify amounts of corisin—using ELISA that they described in the present manuscript—in patient BALF samples to determine a biologically relevant concentration of corisin that could be used in mouse and in vitro studies.

Response:

We performed additional experiments to demonstrate the biologically active concentration of native corisin. We collected the bacterial supernatant of *S. nepalensis*, added the bacterial supernatant to the medium (DMEM) of A549 cells at a 1:10 dilution, and after 48h culture evaluated apoptosis by flow cytometry. The bacterial supernatant showed strong apoptotic activity on A549 cells, and this apoptotic activity was almost completely inhibited by the anticorisin neutralizing antibody. We then measured the concentration of native corisin by ELISA in the bacterial supernatant and found that the average concentration of native corisin in the undiluted bacterial supernatant from *S. nepalensis* was 1042 ± 174.6 pg/ml. As we observed that the bacterial supernatant at 1/10 dilution induces apoptosis of cultured A549 cells, the active biological concentration of native corisin is approximately 100 pg/ml. Therefore, the biological activity of corisin in its natural or native form is significantly (100 pg/ml versus 10 µg/ml: → 10^5 times) stronger than the synthetic peptide.

We also showed that the anticorisin neutralizing antibody significantly inhibits the proapoptotic activity of diluted (1:2 dilution) bronchoalveolar lavage fluid (BALF) from patients with idiopathic pulmonary fibrosis with acute exacerbation on A549 alveolar

epithelial cells. The immunoassay showed that the average concentration of native corisin in undiluted BALF from the IPF patients is 650.9 ± 218.3 pg/ml. As we observed that the BALF from IPF patients at 1/2 dilution induces apoptosis of cultured A549 cells, the active biological concentration of native corisin is approximately 325 pg/ml. Therefore, the biological activity of corisin in its natural or native form in BALF is also significantly (325 pg/ml versus 10 μ g/ml: $\rightarrow 3 \times 10^4$ times) stronger than the synthetic peptide.

We have also measured the half-life of (synthetic corisin) in the A459 cell culture system and found that the in vitro half-life of corisin is less than 1 hour. Rapid penetration of corisin into cells, its rapid metabolism or rapid binding to other proteins in solution may explain this result. Based on this finding, we speculate that the concentration of corisin under in vivo conditions may vary depending on the timing of sample collection.

Please see Figure 4 and the result description on pages 14 and 15 (pdf file), lines 339 to 369 in the revised manuscript:

“...Significant biological activity of native corisin in bacterial culture supernatant and BALF from IPF patients. We then evaluated the concentration of native corisin that is biologically active in the bacterial supernatant with an ELISA system prepared using rabbit antitransglycosylase polyclonal antibody and biotinylated anticorisin mAtb clone 9A. We first confirmed the apoptotic activity of the bacterial supernatant. Bacterial supernatant from *S. nepalensis* was added to the medium of A549 alveolar epithelial cells at 1:10 dilution in the presence or absence of the neutralizing anticorisin A21 mAtb (20 μ g/ml). A549 cells were cultured for 48h, and then apoptosis was evaluated by flow cytometry. Apoptosis induced by the diluted bacterial supernatant was significantly and almost completely inhibited by anticorisin mAtb (**Figure 4a,b; Supplementary Fig. 15d**). Immunoassay of corisin showed that the mean concentration of native corisin in undiluted bacterial supernatants from *S. nepalensis* was 1042 ± 174.6 pg/ml and from *S. haemolyticus* was 802.4 ± 57.7 pg/ml (**Figure 4c**). As the bacterial supernatant was used at 1:10 dilution in the apoptosis assay, we estimate that native corisin has strong biological activity at a concentration of ≥ 100 pg/ml.

We also conducted a similar experiment to evaluate the proapoptotic activity of native corisin present in BALF from IPF patients (**Supplementary Table 1**). The BALF from IPF patients has been shown to cause apoptosis of lung epithelial cells²⁷, and we have also demonstrated that corisin levels are increased in BALF from IPF patients with AE compared to patients with stable disease¹⁷. BALF collected from IPF patients with AE (n=14), and healthy subjects (n=5) was added to the medium of A549 alveolar epithelial cells at 1:2 dilution in the presence of anticorisin mAb clone A21 or control IgG (20 µg/ml), and apoptosis was assessed after 48h. Apoptosis of A549 cells in the presence of BALF from IPF patients was significantly inhibited by the anticorisin mAb (**Figure 4d,e**). The mean concentration of native corisin in undiluted BALF from IPF patients (n=14) was 650.9 ± 218.3 pg/ml and from healthy subjects (n=8) was 458.4 ± 60.7 pg/ml, suggesting that native corisin also has strong biological activity at low concentrations in BALF from IPF patients with AE (**Figure 4f**). However, it is worth noting that the half-life of (synthetic) corisin in solution is less than 1h in a human alveolar cell culture system (**Figure 4g**). Therefore, the concentration of native corisin in vivo during an acute exacerbation event may differ depending on the sample collection time.....”

Native corisin has potent proapoptotic activity probably because its folded structure is conserved after being cleaved from the transglycosylase. However, high concentration of synthetic corisin is needed most likely because a significant number of the synthetic corisin peptides are not in the native state achieved by the natural peptide cleaved from the parental transglycosylase polypeptide. We have investigated this hypothesis by collaborating with an expert in the field of molecular dynamics simulations and confirmed that the probability of the synthetic peptide adopting the native structure is very low (5.8×10^{-4}).

The results have been added in the supplementary data. Please see Supplementary Fig. 18 and the description of the results on page 15, 16 and 17 (pdf file), lines 370 to 407 in the revised manuscript.

“..... **Molecular Dynamics simulations of synthetic corisin.** When the peptide is processed by the bacterium it is active at 100 pg/ml, however, the synthetic peptide

requires 10 $\mu\text{g/ml}$ to observe similar level of activity. To explain the large difference between the activity of the synthetic peptide and the native peptide from the bacterial cell supernatant, we hypothesize that the synthetic peptide is not well-folded in solution which leads to a loss of activity whereas the cleaved peptide from the native polypeptide maintains a folded state. Due to loss of potential stabilizing interactions in the solution, the synthetic peptide conformational ensemble is likely a heterogeneous mixture of peptides in the folded and unfolded states, with a lower population in the native state (folded structure) required to induce apoptosis.

To test the proposed hypothesis, we performed Molecular Dynamics (MD) simulations of synthetic peptide to observe its conformational ensemble in the solution. In order to obtain the conformational ensemble of the peptide effectively, we run simulations from two starting points: the native folded state and the unfolded state. We constructed Markov State Model (MSM)²⁸⁻³⁰ to capture the thermodynamics and kinetics of the distinctly folded states of the synthetic peptide. MSMs are kinetic network models that represent the conformational ensembles of proteins in the form of different conformational states and the rates of transition between them.

The free energy landscape that describes energy barriers among the different conformational states of peptide is shown in **Supplementary Fig. 18a**. In the native structure of peptide, the only one secondary structure is α -helix locating between Val 2 and Ser 6, so the distance between these two residues was chosen as the metric to characterize the peptide folding process. We also found that the formation of the α -helix is the slowest process in the folding of the synthetic peptide. Another important metric used for representing the folding process of the peptide is the root-mean-square deviation (RMSD) of the peptide from its native structure with low RMSD values representing the folded state. Therefore, these two metrics were used for describing the free energy landscape of the peptide folding process. In the native structure of the peptide, the actual distance between Val 2 and Ser 6 is ~ 3.2 Å. Based on the free energy landscape, the predicted free energy of the synthetic peptide adopting the native structure (i.e., an RMSD of < 2 Å and ~ 3 Å distance between Val2 & Ser6) is ~ 5 Kcal/mol, and this is very high to attain (> 5 kcal/mol) (**Supplementary Fig. 18a**). The structure of the native state is shown in **Supplementary Fig. 18c**. In addition, we

observe three different stable unfolded conformational states in the free energy landscape (**Supplementary Fig. 18a, d-f**). These landscapes reveal the synthetic peptide conformational ensemble is a mixture of different conformations of the peptide, and the native conformation only has a small population. To quantify the population of native conformation in the synthetic peptide, we calculate the probability density of each RMSD value (**Supplementary Fig. 18b**). The probability of the synthetic peptide adopting the native structure is 5.8×10^{-4} , and thus validates our proposed hypothesis....."

We also described in the new version of the manuscript how we calculated the dose of synthetic corisin for inducing acute exacerbation in mouse models of pulmonary fibrosis. We based our calculation on the biologically active concentration of synthetic corisin (10 µg/ml) in the in vitro cell experiments.

Synthetic corisin showed potent proapoptotic activity on A549 alveolar epithelial cells at concentration of 10 µg/ml (5 µM). Based on this concentration we calculated the dose of synthetic corisin that would be biologically active in the mouse models of fibrosis using the molarity equation and a software.

Please see page 17 (pdf file), lines 408 to 420 in the revised manuscript:

".....Dose of synthetic corisin for in vivo experiments. Unlike native corisin, a high concentration of synthetic corisin (10 µg/ml or 5 µM) was required to induce significant apoptosis in A549 cells (current study **Supplementary Fig. 14b**)¹⁷. As suggested by the Molecular Dynamics simulations, a significant number of synthetic corisin peptides fail to attain the native state of the natural corisin, and therefore high concentrations of synthetic corisin are required to induce biological activity³¹. We used the biologically active in vitro concentration (5 µM) of the synthetic peptide as a reference to estimate the approximate dose of synthetic corisin required to induce AE in our disease mouse models using a molarity equation and a molarity calculator software as described under Methods. The estimated dose was about 240 µg, and in a preliminary study using varying intranasal doses of synthetic corisin (100, 150, and 300 µg), we found that a dose of 300 µg induces significant lung radiological changes. Therefore, we used 300 µg of synthetic corisin to induce AE in our subsequent in vivo experiments....."

Comment 4:

Notwithstanding the preceding comments about biologically relevant amounts of synthetic corisin, the authors should provide more definitive mouse and in vitro experiments about the roles of corisin in AE-IPF using isogenic bacteria:

- i. *S. haemolyticus* wild-type
- ii. isogenic mutant deficient in production/release of corisin
- iii. complemented strain

Response:

Genetic systems for *Staphylococcus haemolyticus* is not currently available. Therefore, to provide more evidence on the role of corisin in AE-IPF we evaluated whether the intratracheal instillation of *S. haemolyticus*, which carries corisin, and *Staphylococcus epidermidis*, which lacks corisin, induces acute exacerbation of pulmonary fibrosis in the TGF β 1 TG mice with lung established lung fibrosis. We found that the instillation of *S. haemolyticus* induced significant exacerbation of lung fibrosis as compared to *S. epidermidis*.

Please see Supplementary Fig. 3, and the description of the results on page 8, lines 194 to 198 in the revised manuscript.

“.....In addition, intratracheal instillation of *Staphylococcus haemolyticus* strain 12 resulted in a significant increase of lung neutrophil infiltration, pulmonary fibrosis score and lung area of epithelial cell apoptosis in TGF β 1 TG mice compared to intratracheal instillation of *Staphylococcus epidermidis*, which contains transglycosylases lacking the corisin sequence (Supplementary Fig. 3a,b,c,e,f,h,i).....”

Comment 5:

The authors speculated on a putative serine protease that was needed to cleave the transglycosylase to release corisin. Because the authors sequenced the genomes of several *S. haemolyticus* strains, why not make use of this information to generate deletions of genes encoding putative serine proteases to identify the specific serine protease involved in this crucial step. Half of the manuscript seems to be about

understanding better the mechanisms of corisin processing and release, so it is unsatisfying that the protease involved in this process was ultimately not identified.

Response:

We are currently performing experiments to identify and isolate the serine protease from several bacteria that release corisin. As soon as we get definite and repeatable results we are planning to publish in as separate manuscript. Also, please, note that as we stated above, there is currently no genetic manipulation system for our isolated *S. haemolyticus* strains.

Comment 6:

Corisin concentration was quantified in mouse plasma and lung samples (Supplemental Figure 29) but this was not done for human plasma and BALF samples. To further support the authors' hypothesis that "plasma level of corisin is a potential biomarker of pulmonary fibrosis" (line 472), the authors should use their newly developed immunoassay to measure concentrations of corisin in patient plasma and BALF samples.

Response:

As suggested, we have measured the concentration of corisin in serum samples from IPF patients with and without acute exacerbation and healthy subjects and presented the results in the revised version of the manuscript. We found a significant increase in the serum levels of corisin in IPF patients compared to healthy subjects and a significant increased serum levels of corisin in IPF patients with acute exacerbation compared to patients without acute exacerbation.

Please see Figure 9, and the results of the measurements on page 26 (pdf file), lines 631 to 636 in the revised manuscript.

"...High circulating levels of corisin in IPF patients with AE. Based on the above results, we compared the serum levels of corisin among healthy subjects, IPF patients with stable disease, and acute exacerbation. The serum concentration of corisin was significantly increased in IPF patients with the stable disease compared to healthy subjects and in IPF patients with acute exacerbation compared to IPF patients with

stable disease, further supporting the potential usefulness of corisin as a biomarker in IPF patients (Figure 9a, b).....”

Comment 7:

Severity of pulmonary fibrosis was evaluated by CT scan using apparently five differently methods:

- i. CT opacity area (%)—e.g., Sup Fig 27b
- ii. CT area of fibrosis (%)—e.g., Fig 4
- iii. Ashcroft score—e.g., Fig 6h
- iv. CT score—e.g., Sup Fig 20
- v. CT fibrosis score—e.g. Sup Fig 25

It is not clear whether (i) opacity area and (ii) area of fibrosis are one and the same but were labeled differently, or they were indeed different scoring methods. More importantly, the five different methods of scoring the CT scans were not applied consistently for all studies (e.g. only Ashcroft score and fibrosis area (%) was presented in Sup Fig 25). Please clarify in the methods section why four different scoring methods are needed/used and why were they not used consistently in all studies. Did the authors selectively show only results with a particular scoring method because it produced statistical significant results? It would be important to fully disclose the data using the other scoring methods even when the other methods did not yield statistical significance.

Response

We apologized for the lack of clarity regarding parameters used to assess the grade of pulmonary fibrosis and the purpose for using computed tomography (CT) in our previous original manuscript.

We would like to clarify point-by-point as follows:

- i. CT opacity area (%)—e.g., Sup Fig 27b (**Figure 8a in the revised manuscript**): We used CT opacity area to evaluate only LPS-acute lung injury in the revised version of the manuscript

- ii. CT area of fibrosis (%)—e.g., Fig 4 (**Supplementary Fig. 20** and **Supplementary Fig. 21** in the revised manuscript): This was a typo. This should be “area of fibrosis”, which is evaluated in histopathological specimens stained with Masson trichrome using the WinRoof image processing software. Positively stained areas represent collagen deposition area. In the revised manuscript we used the expression “trichrome stain (+) area” for clarity. CT was not used for this evaluation.
- iii. Ashcroft score—e.g., Fig 6h (**Figure 7c,d** in the revised manuscript): For the Ashcroft fibrosis score, lung histopathological specimens stained with hematoxylin & eosin are used. CT is not used for this evaluation.
- iv. CT score—e.g., Sup Fig 20 (**Supplementary Fig. 22b in the revised manuscript**): This is the same as CT fibrosis score. In the revised manuscript we used only the expression “CT fibrosis score” for clarity
- v. CT fibrosis score—e.g. Sup Fig 25: this is the same as “CT score”. As clarified above, in the revised manuscript we used the expression “CT fibrosis score”.

In summary, we used the CT radiological findings only for two purposes:

1) To evaluate lung opacity in LPS-induced acute lung injury:

For clarity, we did not measure lung opacity in the bleomycin-induced lung fibrosis in the revised version of the manuscript.

2) To evaluate CT fibrosis score: We used CT to evaluate the grade of fibrosis in the TGF β 1 TG- and bleomycin-induced lung fibrosis models using a published CT fibrosis scoring system.

It is important to clarify that the expression “CT score” used in the previous manuscript is the same as “CT fibrosis score”. In the revised manuscript we used only the expression “CT fibrosis score”. Also, it is important to clarify that the Ashcroft score is not evaluated by CT findings. As described above, lung specimens stained with hematoxylin & eosin are used for grading lung fibrosis by the Ashcroft fibrosis score.

The use of CT was clarified in the revised version of the manuscript.

Please see page 35 (pdf file), lines 855 to 864:

“..... The radiological findings of lung fibrosis in both TGF β 1-induced (TGF β 1 TG) and BLM-induced lung fibrosis models were evaluated using a CT fibrosis score as follows: score 1, normal radiological lung findings; 2, intermediate findings; 3, slight lung fibrosis; 4, intermediate findings; 5, moderate lung fibrosis; 6, intermediate findings; and 7, advanced lung fibrosis¹⁷. The radiological findings of LPS-acute lung injury were evaluated by measuring lung opacity on axial CT images using the WinRoof Image Processing Software. The average percentage of the total lung opacity area of all axial (transverse) CT images (56 slides) from each individual mouse was calculated, and the mean percentage of lung opacity area was compared among all groups.....”

Finally, we would like to clarify that lung fibrosis was evaluated using 4 methods in the TGF β 1 TG and bleomycin pulmonary fibrosis models:

1. CT fibrosis score: This is the score of radiological findings of lung fibrosis using a score system we have previously published (Nat Commun, 2020). “CT score” is the same as “CT fibrosis score”. However, in the revised version of the manuscript we used only the expression “CT fibrosis score”.
2. Ashcroft fibrosis score: This is the score of histopathological findings of lung fibrosis using lung tissue stained with hematoxylin & eosin. This score was originally published by Ashcroft et al (J Clin Pathol. 1988; 41:467-70. PMID: 3366935). In the revised manuscript we used the expression “Ashcroft fibrosis score” for clarity.
3. Trichrome stained area: This is the Masson trichrome or collagen positive area in lung tissue assessed using the WinRoof image processing software. In the revised manuscript we used for clarity the expression “trichrome stain (+) area” instead of “fibrosis area”.
4. Lung content of hydroxyproline: measured using a commercially available colorimetric assay kit. This is the most objective and most used method to measure the degree of tissue collagen deposition in any organ.

We clarified the methods for assessing pulmonary fibrosis in the revised version of the manuscript.

Please see pages 48 and 49 (pdf file), lines 1200 to 1210:

“..... The grade of lung fibrosis was assessed using the following parameters: 1) CT fibrosis score using a score system¹⁷; 2) Ashcroft fibrosis score using lung tissue stained with hematoxylin & eosin^{73, 106}; 3) Masson trichrome (or collagen) stained area expressed in percentage using the WinRoof image processing software⁷³, and 4) lung hydroxyproline content measured using a colorimetric assay⁷³. For the Ashcroft fibrosis score, microphotographs of five microscopic fields from the lung of each mouse were taken at random, eight blinded observers scored the grade of fibrosis, and the mean fibrosis score for each mouse lung was calculated^{73,106}. For evaluating the Masson trichrome stained area, an investigator unaware of the treatment group took microphotographs of five microscopic fields from the lung of each mouse at random and measured the trichrome stained area using the WinRoof image processing software (Mitani Corp., Fukui, Japan).....”

As suggested, we used consistently these 4 methods to evaluate lung fibrosis in the TGFβ1 TG mouse and bleomycin model of lung fibrosis in the revised version of the manuscript. However, in some experiments the conditions of the mice were very severe, and CT study was not possible. This is because CT is performed under anesthesia, and during anesthesia the mortality of mice with severe disease states is very high.

Please see Figure 7, Supplementary Fig. 18, Supplementary Fig. 24, Supplementary Fig. 25, Supplementary Fig 29, Supplementary Fig 35 in the revised version of the manuscript.

It is important to clarify here that we did not evaluate Ashcroft fibrosis score, trichrome stained area or hydroxyproline in the LPS model because this is a model of acute lung injury (no fibrosis).

Comment 8:

Related to the above comment regarding five different methods for scoring CT scan, in the mouse model of bleomycin-induced pulmonary fibrosis (Sup Fig 23), CT scan was performed on day 9 and day 21. CT opacity area was provided for day 9 but not day 21

(Sup Fig 23), whereas CT fibrosis score was provided for day 21 but not day 9 (Sup Fig 24).

Response:

In the original Supplementary Fig 23, we evaluated lung opacity by CT on day 9 after bleomycin administration because in our model (infusion of bleomycin through osmotic minipump) at early stages (day 9) there is almost no fibrosis but acute lung injury. However, Reviewer #3 considered this figure redundant and suggested to delete this original Supplementary Fig 23. Therefore, we deleted this Supplementary Fig 23 from the revised version of the manuscript.

The CT fibrosis score was used to evaluate lung fibrosis on day 21 because there is more fibrosis than lung inflammation at late stages of bleomycin-induced lung fibrosis.

Please see Figure 7a, and Supplementary Fig. 22 in the revised version of the manuscript.

Comment 9:

Also, regarding Sup Fig 24, it seems that day 21 CT scan was mislabeled as day 22 CT scan (the mislabel would otherwise be in the timeline provided in Sup Fig 23a showing day 21 CT scan). Another example of inconsistent scoring would be that Fig 4 shows for the primary outcome only area of fibrosis whereas Fig 6 shows both area of fibrosis and Ashcroft score.

Response:

We have corrected the mislabeling in the revised version of the manuscript.

We have added new panels to the figures showing evaluation of lung fibrosis consistently using the 4 following methods:

- 1) CT fibrosis score
- 2) Ashcroft fibrosis score
- 3) Trichrome stained area
- 4) Lung content of hydroxyproline

Please see Figure 7, Supplementary Fig. 20, Supplementary Fig. 21, Supplementary Fig. 21, Supplementary Fig. 22, Supplementary Fig. 23,

Supplementary Fig. 25, Supplementary Fig. 29 in the revised version of the manuscript.

Comment 10:

WT mice and TGF β 1 TG mice were treated with anticorisin neutralizing mAtb 21A three times a week, and this was based on the half-life of the anticorisin mAtb in plasma from TGF β 1 TG mice was shown in Sup Fig 16. Would the authors expect the half-life anticorisin mAtb in plasma in TGF β 1 TG mice different from in WT mice?

Response:

We clarified in the revised version of the manuscript that we used female TGF β 1 TG without fibrosis with the same genetic background (C57BL/6J) as the wild type mice used in the experiment to measure the half-life of the anti-corisin mAtb in plasma. Therefore, we believe the results are also applicable to the wild type mice.

Please see,

page 51 (pdf file), lines 1261 to 1265:

“.....Measurement of the half-life of anticorisin neutralizing antibody in plasma. Female TGF β 1 TG mice without fibrosis with the same genetic background (C57BL/6J) as the wild-type mice were used in the experiment. Mice received an intraperitoneal injection of (20 mg/kg) anticorisin mAtb (clone 21A), and blood was sampled after 3, 6, 24, 48 168, 336, and 504 hours.....”

page 33 (pdf file), lines 830 to 836:

“..... Animals. We purchased female WT C57BL/6J mice from Nihon SLC (Hamamatsu, Japan) to induce lung injury, pulmonary fibrosis, and acute exacerbation. WT mice weighing 20~22 g and 8~9 weeks of age were used for the BLM-induced lung fibrosis experiment. We also used male and female transforming growth factor β 1 (TGF β 1) TG mice in a C57BL/6J background that develop spontaneously progressive and fatal pulmonary fibrosis and their WT littermates in the experiments^{17,35,73}. The TG mice and WT littermates weighed 23-26 g and aged 8-10 weeks.....”

Figure legend of Supplementary Fig. 19:

“..... Female TGFβ1 transgenic mice without fibrosis with the same genetic background (C57BL/6J) as the wild-type mice were used in the experiment. Mice received an intraperitoneal injection of (20 mg/kg) anticorisin mAtb (clone 21A), and blood was sampled after 3, 6, 24, 48 168, 336, and 504 hours....”

Comment 11:

In their pivotal studies involving treatment with anti-corisin IgG vs. isotype (Fig. 4e), differences between the two groups appeared to be small and not statistically significant. Using raw data provided by the authors in the supplementary excel file, the Reviewer performed the statistical analysis shown below. With a small sample size of 6 animals per experimental group and a small effect size of only 2.54% difference in area of fibrosis, the difference was not significant.

The Reviewer strongly suggests that the authors should work with a statistician to ensure that all statistical analyses were performed rigorously, and more importantly, that the appropriate sample size (e.g. number of mice per experimental group) was pre-determined and appropriate to detect small effect size. The Reviewer also suggests that the Editor consults with a statistician to review these data. More importantly, the authors should discuss the biological significance of a 2.5% difference in area of fibrosis and comment on whether this is clinically meaningful.

Response

As suggested by the Reviewer, we have consulted a statistician and re-done all the statistical analysis for all figures. The statistical analysis and results are detailed in a separate file (**Statistical analysis**).

For the data described in the previous manuscript Figure 4e (**Supplementary Fig 21c,d in the revised manuscript**) we used a wrong statistical approach (one-tailed). In the revised manuscript, statistical analysis was performed using analysis of variance (ANOVA) with post hoc test. Using this method, the percentage of trichrome stain (+) (fibrosis area) area was not significant ($p=0.1$), although mice treated with anticorisin mAtb showed reduced area of fibrosis (**Supplementary Fig 21c,d in the revised manuscript**).

We also performed the Ashcroft fibrosis analysis using the hematoxylin-eosin stained lung tissue and observed the same tendency. However, the lung hydroxyproline content was significantly reduced in mice treated with anticorisin mAtb compared to mice treated with the scrambled peptide. In general, measurement of lung hydroxyproline is the most objective and generally used marker of pulmonary fibrosis (or organ fibrosis in general) because the amount of hydroxyproline represents a direct marker of collagen deposition in the lungs.

Please see Supplementary Fig 21, and the description of the results on page 18 (pdf file), lines 448 to 454:

“..... The Ashcroft fibrosis score and the collagen (trichrome) stained area were decreased in mice treated with anticorisin mAtb compared to the control group, although the decrease was not statistically significant. The lung hydroxyproline content, a generally used marker of collagen tissue deposition, was significantly lower in mice treated with anticorisin mAtb than in mice treated with control IgG. The lung hydroxyproline content was significantly associated with the number of lung inflammatory cells (**Supplementary Fig. 21a,b,c,d,e,f**). These observations point to corisin as a potential therapeutic target for AE in pulmonary fibrosis.....”

Comment 12:

Line 448: Supplementary fig. 37 ???□ 27

Supplementary Fig. 22 legend “sacrificed on day **21**”

Line 396-397: We performed a chest CT study before sacrifice on day **22** after BLM or SAL infusion: mice was sacrificed on day **21** but panel 22a shows day **22**.

Tissue concentrations of proteins (e.g. corisin) should be expressed as ng/mg of tissue (or ng/lung) and not as ng/mL (e.g. Supplemental Figure 29b).

Response

We have corrected the error in the previous manuscript line 448: Supplementary Fig 27, which is **Figure 8bc** in the revised version of the manuscript.

The error in the legend of Supplementary Fig 22 (**Supplementary Fig 27** in the revised version of the manuscript) was also corrected.

The error in the previous line 396-397 was also corrected. **Please see page 21, lines 516 to 517 in the revised manuscript.**

The tissue concentration of corisin was expressed as **pg/mg protein** as suggested. Please see **Supplementary Fig. 35c.**

Response to Queries of Reviewer #3

Comment 1:

The manuscript entitled “Inhibition of a lung microbiota-derived peptide ameliorates acute exacerbation of pulmonary fibrosis.” by D’Alessandro-Gabazza et al. uses two recognized models of murine pulmonary fibrosis, i.e., TGFb overexpression and bleomycin induced fibrosis as well as a model of LPS induced acute lung injury to study the effects of a microbiology derived peptide ‘Corisin’ in lung disease. The authors provide evidence that Corisin is produced by proteolytic degradation of bacterial transglycosylase produced in several commensal staphylococcus species and causes epithelial cell apoptosis through activation of the intrinsic caspase-9 mediated pathway. Further, evidence is provided that Corisin treatment results in exacerbation of pulmonary fibrosis and lung injury. The manuscript details the production of a neutralizing anti-Corisin monoclonal antibody that blocks the deleterious effects of Corisin and protects from acute exacerbation. Overall, the article is well written, although somewhat lengthy, and although Corisin was characterized previously by this same group, the detailing of the anti-Corisin antibody and its biologic effects are assuredly novel. I list my comments and concerns about the manuscript below.

Response

We thank very much the constructive comments of the Reviewer that have substantially improved the quality of the revised version of the manuscript.

Comment 2:

My major concern is with the discrepancy between the two fibrotic models used and the timing of the Corisin / anti-corisin treatments as well as the inclusion of an acute lung injury model. I feel that, while the inclusion of all of these models was interesting, the manuscript suffered from a lack of mechanistic insight into any particular model. Further, the models all used different treatment regimens and timings for the delivery of Corisin / anti-Corisin. For instance, in the TGBb model, mice with existing fibrosis were either given Corisin alone to induce what the authors describe as an acute exacerbation of fibrosis or Corisin and the anti-Corisin antibody together which was shown to reduce the exacerbation. However, in the BLM model, the authors used two separate experiments, one where Corisin was given alone and one where the anti-Corisin antibody was given very early during BLM administration before fibrosis is established. Further, the LPS injury model used an entirely different dosing scheme where mice were pretreated with anti-Corisin antibody and then treated with LPS.

Response

Our hypothesis is that corisin is involved in the acute phase of lung injury (acute exacerbation in IPF and acute lung injury in a general model). Therefore, we pretreated the TGF β 1 TG mice with lung fibrosis with the anticorisin mAtb before inducing acute exacerbation by intratracheal corisin, and in similar schedule, we pretreated normal wild type mice with anticorisin antibody before inducing acute lung injury by intratracheal LPS. We hypothesized that during LPS-induced lung injury, the lung microbiota reacts releasing corisin that further worsens lung injury induced by LPS, and that the lung injury worsened by corisin is inhibited by anticorisin mAtb.

The regimens for both models are depicted below:

Regimen 1: Pre-treatment with anticorisin mAtb and induction of acute exacerbation in TGF β 1 TG mice with pulmonary fibrosis (please see Supplementary Fig 20a, revised manuscript)

Regimen 2: Pre-treatment with anticorisin mAtb or control IgG and induction of acute lung injury with LPS in normal wild type mice (please see Figure 8a, revised manuscript)

As described above, **regimen 1** and **regimen 2** for corisin/anticorisin treatment in both TGF β 1-induced lung fibrosis and LPS-induced lung injury models were similar. The only difference was the interval of pre-treatment with the anticorisin mAtb or control IgG. TGF β 1 TG with pulmonary fibrosis were pre-treated with anticorisin mAtb for 2 weeks (5 injections), while wild type mice for the LPS model were pre-treated for only 1 week (three injections).

An important reported limitation (PMID: 2404582; PMID: 17504872) of using monoclonal antibodies is their poor penetration into abnormal tissues (e.g., fibrotic tissue, tumors) due to their relatively large size. **This was the reason why we pre-treated for longer time (two additional injections) TGF β 1 TG mice with pulmonary fibrosis than wild type mice (for the LPS model).**

The rationale for performing the LPS model was to demonstrate that corisin may also be involved in other forms of acute lung injury (i.e., with similarity to acute exacerbation of IPF) such as that caused by bacterial or viral infection.

Regarding the bleomycin-induced lung fibrosis, the regimens for treating with anticorisin mAtb or control IgG was different from the above-described two models (TGF β 1 TG and LPS models). The following three treatment regimens were used for different purposes:

Regimen 3: Treatment with anticorisin mAtb or control IgG during the acute phase and chronic (fibrotic) phase of lung fibrosis induced by bleomycin delivered through subcutaneous osmotic minipumps (please see Supplementary Fig 27a, revised manuscript). The osmotic minipumps deliver constantly their contents of bleomycin (100 mg/kg bleomycin diluted in 200 μ L of saline) at 0.5 μ L/hour for 7 days.

Regimen 4: Treatment with anticorisin mAtb or control IgG only during the acute phase of lung injury/fibrosis induced by bleomycin delivered through subcutaneous osmotic minipumps (please see Supplementary Fig 28a, revised manuscript).

We used this **regimen 4**, to clarify whether the beneficial effect of the anticorisin mAtb observed with **regimen 3** was because the anticorisin mAtb inhibited the acute phase of bleomycin-induced lung injury/fibrosis. Acute lung injury occurs at early stages whereas lung fibrosis occurs at chronic or late stages in this model of bleomycin-induced lung injury (PMID: 17993587; PMID: 20167853; PMID: 28804709). It is known that “**the Intensity of chronic lung inflammation and fibrosis after bleomycin is directly related to the severity of acute injury**” (PMID: 2449833; PMID: 20167853; PMID:

28804709). Therefore, we hypothesized that the anticorisin mAb is effective because it inhibits the acute phase of the bleomycin-induced lung injury/fibrosis. To interrogate our hypothesis we treated the mice (with this **regimen 4**) only during the acute phase of the disease.

Regimen 5: Treatment with anticorisin mAb or control IgG during the acute phase of lung injury induced by intratracheal instillation of bleomycin (please see Supplementary Fig 30a, revised manuscript). Bleomycin was administered once by intratracheal route (1.5 mg/kg mouse weight) in a total volume of 75 μ L.

We used this **regimen 5** to evaluate the grade of inhibition of apoptosis by the anticorisin mAb in the acute phase of the disease. There is no clear report on the time course of lung epithelial cell apoptosis in the model of lung fibrosis induced by bleomycin delivered using osmotic minipumps. However, there are reports showing prominent apoptosis of lung epithelial cells 7 days after intratracheal instillation of bleomycin (PMID 22394287). Therefore, we used this regimen 5 for this evaluation.

We have clarified these points in the revised manuscript.

Please see,

pages 21 (pdf file), lines 505 to 509:

“..... we infused BLM in mice through subcutaneous osmotic mini-pumps for 7 days and treated them with the anticorisin neutralizing mAb or control IgG by intraperitoneal route

three times a week for three weeks during the acute and chronic phase of BLM-induced lung injury before sacrifice on day 22 (**Supplementary Fig. 27a**).....”

page 22 (pdf file), lines 527 to 533:

“..... In general, lung injury induced by BLM administered subcutaneously through osmotic mini-pumps is characterized by an acute phase of lung inflammation that peaks on day 12, followed by lung fibrosis in the chronic phase³¹. To determine whether the anticorisin mAtb ameliorates lung fibrosis by inhibiting lung injury induced by BLM in the early acute phase, we treated mice with anticorisin neutralizing mAtb or control IgG in the acute phase (days 2, 4, 6, 9, and 11) after BLM mini-pump infusion. We then stopped the treatment until sacrifice on day 22 (**Supplementary Fig. 28a**).....”

Pages 22 and 23 (pdf file), lines 549 to 554:

“..... **Anticorisin mAtb inhibits apoptosis during the acute phase of BLM-induced lung injury.** We administered anticorisin mAtb or control IgG by intraperitoneal route to mice with acute lung injury induced by BLM delivered once by intratracheal instillation and compared the grade of apoptosis. Mice treated with the anticorisin mAtb showed a significant reduction in apoptosis of lung epithelial cells compared to mice treated with control IgG (**Supplementary Fig. 30a,b**).....”

Comment 3:

Unfortunately, this makes it difficult to deduce the mechanism for how Corisin maybe be inhibiting lung injury / fibrosis. The authors argue that Corisin causes / exacerbates epithelial cell apoptosis and that the anti-corisin antibody neutralizes this effect. However, the effects of early Corisin treatment also greatly diminish the infiltration of inflammatory cells into the lungs and thus effects on inflammatory pathways, such as TLR signaling, are suggested.

1)I feel the authors should examine the effects of Corisin on inflammatory TLR signaling, especially TLR4, as LPS instillation was used in the manuscript. This can be done in epithelial cells and/or cultured macrophages looking at inflammatory cytokines / chemokine output as an endpoint. Because neutrophils were shown to be reduced

especially in the LPS model I would be interested to see the effect of Corisin treatment on the induction of neutrophil specific chemokines such as CXCL1 and CXCL2. Other genes of interest would be CCL2 and CCL12 which have both shown to be important for the pathogenesis of BLM induced fibrosis.

Response

As suggested by the reviewer we have examined the TLR4 signaling after treating alveolar epithelial cells with corisin. As described in **Supplementary Fig. 33**, corisin induces cytokines and chemokines in alveolar epithelial cells.

Please see Supplementary Fig 33, and pages 24 (pdf file), lines 588 to 598 in the revised manuscript:

“...Corisin increases the expression of pro-inflammatory factors and enhances the pro-apoptotic activity of BLM and LPS in alveolar epithelial cells. The significant amelioration of lung inflammation in mice with BLM-induced pulmonary fibrosis and LPS-induced acute lung injury treated with anticorisin mAb suggests the participation of corisin in the mechanism of both disease models. To demonstrate this, we cultured A549 cells for 24h in the presence of corisin and assessed the expression of cytokines and chemokines. Corisin significantly increased the mRNA expression of CCL2, CXCL1, CXCL2, IL-8, the secretion of CCL2, CCL3, CXCL1, IL-8, and the activation of the NF κ B pathway in A549 alveolar epithelial cells compared to controls (**Supplementary Fig. 33a,b**). Activation of the NF κ B pathway is probably secondary to increased levels of chemokines in the culture supernatant of cells treated with corisin..”

We have also performed additional studies to further demonstrate the involvement of the mitochondrion in the mechanism of apoptosis induced by corisin in alveolar epithelial cells.

Please see Supplementary Fig 6, and pages 10 (pdf file), lines 233 to 236 in the revised manuscript:

“..... Consistent with findings in other types of cells, we also found a significantly increased reactive oxygen species accumulation in alveolar epithelial cells treated with corisin compared to scrambled peptide-treated cells, and loss of mitochondrial

membrane integrity during apoptosis induced by corisin (**Supplementary Fig. 6a,b,c,d**)....”

Comment 4:

2) All of the in vitro assays showed a mild but statistically significant increase in epithelial cell apoptosis when using Corisin alone, however, Corisin treatment alone is insufficient to induce any pathology when administered to mice. Do the authors have an explanation for this seemingly incongruent observation? Would epithelial cell apoptosis in response to Corisin be augmented if the cells were pre-treated with Bleomycin or another apoptosis inducing agent in vitro?

Response

As suggested by the Reviewer, we have evaluated the effect of corisin in cells pre-treated with bleomycin.

Corisin increased the grade of apoptosis induced by bleomycin or LPS in A549 cells.

Therefore, although corisin is not pathological under physiological conditions, under pathological conditions we believe that corisin acts in the second “hit” of tissue injury, as we discussed in the revised manuscript.

Please see Supplementary Fig. 34a,b,c and the result description on pages 24 and 25 (pdf file), lines 598 to 604 in the revised manuscript.

“.....In a separate experiment, we cultured A549 cells in the presence of corisin alone or in the presence of a combination of corisin and BLM or LPS and assessed apoptosis by flow cytometry. Corisin in combination with BLM or LPS significantly increased the percentage of apoptotic cells compared to BLM or LPS alone (**Supplementary Fig. 34a,b,c**). These observations suggest that corisin per se stimulates the secretion of cytokines and chemokines and enhances apoptosis of BLM or LPS in alveolar epithelial cells.....”

Please also see the discussion section on pages 31 and 32 (pdf file), lines 769 to 779 in the revised manuscript.

“.....In a cellular milieu, where the alveolar epithelial cells have sustained acute injury and therefore predisposition to the mitochondrial-targeting and damaging corisin, the enhanced secretion of the proapoptotic peptide (due to the enriched corisin-producing bacteria) then leads to a concomitant release of potent activators that exacerbate the apoptotic activity on the lung alveolar epithelial cells. Enhancement of LPS- and BLM-induced apoptosis and expression of cytokines and chemokines in alveolar epithelial cells observed in the present study supports this assumption. It is thus reasonable to propose that the “first hit” or lung injury increases the vulnerability of the lung to the proapoptotic activity of corisin that probably constitutes the “second hit” or “final hit” for triggering AE-IPF or the most severe form of acute lung injury termed ARDS.....”

Comment 5:

3) Lung tissue associated Corisin was quantified in Fig. S29C. What time point was this measurement taken at? If it was 22days after BLM like the rest of figure s25 then it would appear that early neutralization can inhibit the production of Corisin throughout the BLM timecourse but only results in a modest decrease in the amount of cleaved caspase3 while showing a much larger reduction in hydroxyproline.

Response

The reviewer is correct. The tissue samples for corisin measurement were collected on day 22 in the experiment described in Fig. S29C (**Supplementary Fig 35c in the revised manuscript**).

We have clarified the timing of sample collection in the revised manuscript.

Please see page 25, lines 613 to 616:

“..... We also measured the lung tissue levels of corisin in mice with BLM-induced lung fibrosis and control mice (described in **Figures 6 and 7**) collected on day 22 after starting bleomycin infusion and evaluated correlation with fibrosis markers.....”

As described above, the Intensity of chronic lung inflammation and fibrosis after bleomycin is directly related to the severity of acute injury (PMID: 2449833). Therefore, we believe that corisin plays a critical role in the acute phase of bleomycin-induced lung

injury, and that early inhibition of corisin by the anticorisin mAtb results in less lung fibrosis at late stages.

To demonstrate that corisin's detrimental effect during the acute phase of bleomycin-induced lung injury is a determinant factor of the severity of lung fibrosis at late stages, we have performed the experiment in which we treated the mice only during the acute phase of the disease (and left untreated during the chronic phase) as described in **Supplementary Fig. 28** and **Supplementary Fig. 29**.

Please see page 22, lines 526 to 546 in the revised manuscript:

“... Treating the acute phase of BLM-induced lung injury with the anticorisin mAtb ameliorates pulmonary fibrosis. In general, lung injury induced by BLM administered subcutaneously through osmotic mini-pumps is characterized by an acute phase of lung inflammation that peaks on day 12, followed by lung fibrosis in the chronic phase²⁹. To determine whether the anticorisin mAtb ameliorates lung fibrosis by inhibiting lung injury induced by BLM in the early acute phase, we treated mice with anticorisin neutralizing mAtb or control IgG on the acute phase (days 2, 4, 6, 9, and 11) after BLM mini-pump infusion. We then stopped the treatment until sacrifice on day 22 (**Supplementary Fig. 26a**). Mice treated with anticorisin mAtb revealed a significant reduction in the BALF number of lymphocytes during the chronic phase of the disease (day 22 after starting BLM infusion) compared to mice receiving control IgG. The plasma levels of SP-C, SP-D, periostin, osteopontin, the BALF levels of MUC-1, total TGF β 1 and the cleavage of caspase-3 on day 22 were significantly lower in mice treated with anticorisin mAtb than in mice treated with control IgG (**Supplementary Fig. 28b,c,d,e,f**). The plasma levels of MUC-1 and collagen I also decreased in mice treated with anticorisin mAtb compared to mice treated with control IgG, although the reduction was not statistically significant

In addition, mice treated with anticorisin mAtb showed significant amelioration of the CT fibrosis score and significantly reduced Ashcroft fibrosis score and hydroxyproline content in the lungs compared to mice treated with control IgG. The collagen (trichrome) stained area was also reduced in mice treated with anticorisin mAtb, but the reduction was not statistically significant (**Supplementary Fig. 29a,b,c,d,e,f,g**..”

page 30, lines 730 to 733:

“...Moreover, additional experiments showed that treating mice with anticorisin mAtb during the acute phase of BLM-induced injury is sufficient to suppress lung fibrosis development in the chronic phase, further supporting the pathogenic role of corisin in acute inflammatory responses...”

Comment 6:

a. Does neutralization of Corisin have a greater effect on cell apoptosis early after BLM instillation (7 – 14 days)? Similarly, did the authors explore early delivery of exogenous Corisin (0 – 14 days) and its effect on late fibrosis?

Response

We performed additional experiments and demonstrated that neutralization with the anticorisin mAtb at early stages also has an inhibitory effect on apoptosis.

Please see Supplementary Fig. 30 and the result description on pages 22 and 23 (pdf file), lines 549 to 554 in the revised version of the manuscript.

“...**Anticorisin mAtb inhibits apoptosis during the acute phase of BLM-induced lung injury.** We administered anticorisin mAtb or control IgG by intraperitoneal route to mice with acute lung injury induced by BLM delivered once by intratracheal instillation and compared the grade of apoptosis. Mice treated with the anticorisin mAtb showed a significant reduction in apoptosis of lung epithelial cells compared to mice treated with control IgG (**Supplementary Fig. 30a,b**).....”

We also performed additional experiments and demonstrated that early delivery of exogenous corisin worsens lung fibrosis.

Please see Supplementary Fig. 25 and the result description on page 20 (pdf file), lines 486 to 495 in the revised version of the manuscript.

“...**Mice receiving intranasal corisin at early stages of BLM-induced lung injury develop advanced pulmonary fibrosis.** Mice received BLM infusion through osmotic mini-pump for 7 days to induce lung injury/fibrosis, and treated with intranasal corisin (100 µg) or scrambled peptide (100 µg) on days 3, 4, 5, 7, 9, 10, and 11 and euthanized

on day 22 after starting BLM infusion. Mice with lung fibrosis receiving intratracheal corisin showed significantly increased Ashcroft score, collagen (trichrome) stained area, and lung hydroxyproline content compared to mice receiving scrambled peptide (**Supplementary Fig. 25a,b,c,d**). No difference was observed in Ashcroft score, trichrome stained area, or hydroxyproline content between mice without lung fibrosis receiving corisin or scrambled peptide.....”

Comment 7:

b. I would suggest using the Corisin ELISA throughout a time course of BLM instillation to determine when Corisin concentrations are highest. It seems the Authors are suggesting that the effects are most apparent in the late stages of fibrosis, however, neutralization of Corisin immediately after BLM instillation rescued fibrosis suggesting that Corisin is most important during the early stage, i.e. inflammatory phase, of fibrosis.

Response

As suggested, we measured longitudinally corisin after bleomycin administration through osmotic minipumps. We found that the level of corisin increases gradually from early stages, reaches a peak on day 14 and 18 and then gradually decreases.

Please see Supplementary Fig. 26, and the result description on pages 20 and 21 (pdf file), lines 496 to 502 in the revised manuscript.

“.....**Longitudinal changes of circulating native corisin during BLM-induced pulmonary fibrosis.** Mice were infused BLM (n=5) or saline (n=4) through osmotic mini-pumps for 7 days, and blood samples were collected from the mouse tail vein on days 0, 3, 6, 11, 14, 18, 21, and 25 to measure the concentration of corisin. The plasma corisin levels gradually increased from day 3 to reach significant levels on days 14 and 18, suggesting that circulating corisin increases during the acute phase and remains high during the chronic phase of lung injury (**Supplementary Fig. 26a,b**).....”

Comment 8:

c. Why does treatment with anti-Corisin antibody show a decrease in the amount of detectable Corisin only after BLM instillation (Fig. S29C) when there is a demonstrable amount of Corisin present in the lungs even in saline treated mice?

Response

We believe the explanation was the low sensitivity of our corisin immunoassay. Therefore, we developed a new immunoassay using anti-transglycosylase as first or coating antibody followed by a biotinylated anti-corisin mAb as second antibody, and re-measured corisin in all samples described in **Supplementary Fig 35 of the revised manuscript**. The sensitivity of the immunoassay improved, and the level of corisin decreased even in the WT/SAL/corisin group, although the decrease did not reach statistically significant level.

Please see Supplementary Fig 35c in the revised manuscript.

Comment 9:

d. Even 300ug of purified Corisin resulted in only modest increases in cell apoptosis and fibrosis area in the lungs of BLM treated mice. Is the amount of exogenous Corisin given in figures 4 and 5 physiologically relevant given that there is likely sub-microgram levels of Corisin present in an entire mouse lung?

Response

We performed additional experiments to demonstrate the biologically active concentration of native corisin. We collected the bacterial supernatant of *S. nepalensis*, added the bacterial supernatant to the medium (DMEM) of A549 cells at a 1:10 dilution, and after 48h culture evaluated apoptosis by flow cytometry. The bacterial supernatant showed strong apoptotic activity on A549 cells, and this apoptotic activity was almost completely inhibited by the anticorisin neutralizing antibody. We then measured the concentration of native corisin by ELISA in the bacterial supernatant and found that the average concentration of native corisin in the undiluted bacterial supernatant from *S. nepalensis* was 1042 ± 174.6 pg/ml. As we observed that the bacterial supernatant at 1/10 dilution induces apoptosis of cultured A549 cells, the active biological concentration of native corisin is approximately 100 pg/ml. Therefore, the biological activity of corisin in its natural or native form is significantly (100 pg/ml versus 10 µg/ml: → 10^5 times) stronger than the synthetic peptide.

We also showed that the anticorisin neutralizing antibody significantly inhibits the proapoptotic activity of diluted (1:2 dilution) bronchoalveolar lavage fluid (BALF) from

patients with idiopathic pulmonary fibrosis with acute exacerbation on A549 alveolar epithelial cells. The immunoassay showed that the average concentration of native corisin in undiluted BALF from the IPF patients is 650.9 ± 58.3 pg/ml. As we observed that the BALF from IPF patients at 1/2 dilution induces apoptosis of cultured A549 cells, the active biological concentration of native corisin is approximately 325 pg/ml. Therefore, the biological activity of corisin in its natural or native form in BALF is also significantly (325 pg/ml versus 10 μ g/ml: $\rightarrow 3 \times 10^4$ times) stronger than the synthetic peptide.

We have also measured the half-life of (synthetic corisin) in the A459 cell culture system and found that the *in vitro* half-life of corisin is less than 1 hour. Rapid penetration of corisin into cells, its rapid metabolism or rapid binding to other proteins in solution may explain this result. Based on this finding, we speculate that the concentration of corisin under *in vivo* conditions may vary depending on the timing of sample collection.

We explained this in the revised version of the manuscript.

Please see figure 4 and the result description on pages 14 and 15 (pdf file), lines 339 to 369 in the revised manuscript:

“...Significant biological activity of native corisin in bacterial culture supernatant and BALF from IPF patients. We then evaluated the concentration of native corisin that is biologically active in the bacterial supernatant with an ELISA system prepared using rabbit antitransglycosylase polyclonal antibody and biotinylated anticorisin mAtb clone 9A. We first confirmed the apoptotic activity of the bacterial supernatant. Bacterial supernatant from *S. nepalensis* was added to the medium of A549 alveolar epithelial cells at 1:10 dilution in the presence or absence of the neutralizing anticorisin A21 mAtb (20 μ g/ml). A549 cells were cultured for 48h, and then apoptosis was evaluated by flow cytometry. Apoptosis induced by the diluted bacterial supernatant was significantly and almost completely inhibited by anticorisin mAtb (**Figure 4a,b; Supplementary Fig. 14d**). Immunoassay of corisin showed that the mean concentration of native corisin in undiluted bacterial supernatants from *S. nepalensis* was 1042 ± 174.6 pg/ml and from *S. haemolyticus* was 802.4 ± 57.7 pg/ml (**Figure 4c**). As the bacterial supernatant was

used at 1:10 dilution in the apoptosis assay, we estimate that native corisin has strong biological activity at a concentration of 100 pg/ml.

We also conducted a similar experiment to evaluate the proapoptotic activity of native corisin present in BALF from IPF patients (**Supplementary Table 1**). The BALF from IPF patients has been shown to cause apoptosis of lung epithelial cells²⁶, and we have also demonstrated that corisin levels are increased in BALF from IPF patients with AE compared to patients with stable disease¹⁷. BALF collected from IPF patients with AE (n=14), and healthy subjects (n=5) was added to the medium of A549 alveolar epithelial cells at 1:2 dilution in the presence of anticorisin mAb clone A21 or control IgG (20 µg/ml), and apoptosis was assessed after 48h. Apoptosis of A549 cells in the presence of BALF from IPF patients was significantly inhibited by the anticorisin mAb (**Figure 4d,e**). The mean concentration of native corisin in undiluted BALF from IPF patients (n=14) was 650.9 ± 58.3 pg/ml and from healthy subjects (n=5) was 420.1 ± 10.7 pg/ml, suggesting that native corisin also has strong biological activity at low concentrations in BALF from IPF patients with AE (**Figure 4f**). However, it is worth noting that the half-life of (synthetic) corisin in solution is less than 1h in a human alveolar cell culture system (**Figure 4g**). Therefore, the concentration of native corisin in vivo during an acute exacerbation event may differ depending on the sample collection time.....”

Native corisin has potent proapoptotic activity probably because its folded structure is conserved after being cleaved from the transglycosylase. However, high concentration of synthetic corisin is needed probably because a significant number of the synthetic corisin peptides fail to achieve the native (normal) folded state. We have investigated this hypothesis by collaborating with an expert in the field of molecular dynamics simulations and confirmed that the probability of the synthetic peptide adopting the native structure is very low (5.8×10^{-4}).

The results have been added in the supplementary data. Please see Supplementary Fig. 18 and the description of the results on pages 15, 16 and 17 (pdf file), lines 370 to 407 in the revised manuscript.

“..... **Molecular Dynamics simulations of synthetic corisin.** When the peptide is processed by the bacterium it is active at 100 pg/ml, however, the synthetic peptide

requires 10 $\mu\text{g/ml}$ to observe similar level of activity. To explain the large difference between the activity of the synthetic peptide and the native peptide from the bacterial cell supernatant, we hypothesize that the synthetic peptide is not well-folded in solution which leads to a loss of activity whereas the cleaved peptide from the native polypeptide maintains a folded state. Due to loss of potential stabilizing interactions in the solution, the synthetic peptide conformational ensemble is likely a heterogeneous mixture of peptides in the folded or unfolded states, with a lower population in the native (folded structure) required to induce apoptosis.

To test the proposed hypothesis, we performed Molecular Dynamics (MD) simulations of synthetic peptide to observe its conformational ensemble in the solution. In order to obtain the conformational ensemble of the peptide effectively, we run simulations from two starting points: the native folded state and the unfolded state. We constructed Markov State Model (MSM)²⁸⁻³⁰ to capture the thermodynamics and kinetics of the distinctly folded states of the synthetic peptide. MSMs are kinetic network models that represent the conformational ensembles of proteins in the form of different conformational states and the rates of transition between them.

The free energy landscape that describes energy barriers among the different conformational states of peptide is shown in **Supplementary Fig. 18a**. In the native structure of peptide, the only one secondary structure is an α -helix located between Val 2 and Ser 6, so the distance between these two residues was chosen as the metric to characterize the peptide folding process. We also found that the formation of the α -helix is the slowest process in the folding of the synthetic peptide. Another important metric used for representing the folding process of the peptide is the root-mean-square deviation (RMSD) of the peptide from its native structure with low RMSD values representing the folded state. Therefore, these two metrics were used for describing the free energy landscape of the peptide folding process. In the native structure of the peptide, the actual distance between Val 2 and Ser 6 is $\sim 3.2 \text{ \AA}$. Based on the free energy landscape, the predicted free energy of the synthetic peptide adopting the native structure (i.e., an RMSD of $< 2 \text{ \AA}$ and $\sim 3 \text{ \AA}$ distance between Val2 & Ser6) is $\sim 5 \text{ Kcal/mol}$, and this is very high to attain ($> 5 \text{ kcal/mol}$) (**Supplementary Fig. 18a**). The structure of the native state is shown in **Supplementary Fig. 18c**. In addition, we

observe three different stable unfolded conformational states in the free energy landscape (**Supplementary Fig. 18a,d,e,f**). These landscapes reveal that the synthetic peptide conformational ensemble is a mixture of different conformations of the peptide, and the native conformation only has a small population. To quantify the population of native conformation in the synthetic peptide, we calculate the probability density of each RMSD value (**Supplementary Fig. 18b**). The probability of the synthetic peptide adopting the native structure is 5.8×10^{-4} , and thus validates our proposed hypothesis....."

We also described in the new version of the manuscript how we calculated the dose of synthetic corisin for inducing acute exacerbation in mouse models of pulmonary fibrosis. We based our calculation on the biologically active concentration of synthetic corisin in the *in vitro* cell experiments.

Synthetic corisin showed potent proapoptotic activity on A549 alveolar epithelial cells at concentration of 10 µg/ml (5 µM). Based on this concentration, we calculated the dose of synthetic corisin that would be biologically active in the mouse models of fibrosis using the molarity equation and a software.

Please see page 17 (pdf file), lines 408 to 420 in the revised manuscript:

“.....**Dose of synthetic corisin for in vivo experiments.** Unlike native corisin, a high concentration of synthetic corisin (10 µg/ml or 5 µM) was required to induce significant apoptosis in A549 cells (current study **Supplementary Fig. 14b**)¹⁷. While native corisin peptides probably conserve their folded structure after cleavage from transglycosylase, it is likely that a significant number of the synthetic corisin peptides fail to attain the native state, and therefore high concentrations of synthetic corisin are required to induce biological activity²⁷. We used the biologically active *in vitro* concentration (5 µM) of the synthetic peptide as a reference to estimate the approximate dose of synthetic corisin required to induce AE in our disease mouse models using a molarity equation and a molarity calculator software as described under Methods. The estimated dose was about 240 µg, and in a preliminary study using varying intranasal doses of synthetic corisin (100, 150, and 300 µg), we found that a dose of 300 µg induces significant lung

radiological changes. Therefore, we used 300 µg of synthetic corisin to induce AE in our subsequent in vivo experiments.....”

Comment 10:

3)I am curious as to why the Authors chose a multiple comparison statistical method for the comparisons in figure 4 (ANOVA, 3 groups compared) but chose to do a single comparison method in figure 5 (T-test, 4 groups split into two comparisons). Figure 5 makes it very difficult to determine if the group of mice treated with BLM are significantly different in comparison to the saline controls. I would strongly suggest using a multiple comparison method such as ANOVA on this figure.

Response

As suggested, we used ANOVA with a post hoc test to re-evaluate the statistical differences between 4 groups shown in the previously submitted Figure 5 (**Supplementary Fig 23, and Supplementary Fig 24 in the revised manuscript**). We have also re-done all statistics and presented the results in a separate file (please see the supplementary **Statistical analysis** file).

Please see Supplementary material-Statistical results file, the revised Supplementary Fig. 23 and Supplementary Fig. 24.

Please also see the description of the statistical analysis on page 52 (pdf file), lines 1286 to 1293:

“..... **Statistical analysis.** We expressed the data as the mean ± standard deviation of the means (S.D.). The statistical difference between the two variables with normal distribution was assessed by the two-tailed unpaired Student's t-test and the difference between three or more variables with normal distribution by one-way analysis of variance (ANOVA) using the Newman-Keuls' test for post hoc analysis. Variables with skewed distribution were evaluated by the two-tailed Mann-Whitney U test or the Kruskal-Wallis ANOVA with Dunn's test. P-value <0.05 was considered statistically significant. We used GraphPad Prism vs. 7 (GraphPad Software, Inc., San Diego, CA) to perform the statistical analysis.....”

Comment 11:

5) Figure 5g and 6j show relatively small increases / decreases in the area of fibrosis between treated and control groups, however, histology images in 5g and 6i seem to show massive changes in lung architecture. Are these images representative? I would suggest showing lower power magnifications or choosing images that accurately reflect the quantified data. Furthermore, please add the magnification and scale bar information for the histology figures to either the materials and methods or the figure legend.

Response

As suggested, we have presented more representative images of the lungs stained with trichrome in the previously submitted Figure 5f (**Supplementary Fig 24c in the revised manuscript**) and Figure 6j (**Figure 7e,f in the revised manuscript**).

Please see Supplementary Fig 24c and Figure 7e,f in the revised manuscript.

Comment 12:

6) The amount of data presented, especially, supplementary data is excessive and needs to be trimmed substantially. For instance, figures S19 through S25 are largely redundant with figure 5 and 6 in the manuscript. If the Authors could pick just the most important data from these and create a new supplemental figure it would substantially improve manuscript readability.

Response

As suggested, we have reduced the number of figures previously shown. Only the representative CT were presented (the CT of each mouse was presented in the previous original version of the manuscript). Figure 5 was transferred to the Supplementary information file, and some panels of Figure 6 were separated to show them in the Supplementary information file.

Comment 13:

Figure 9, while a nice graphical depiction of how lung fibrosis occurs does not really illustrate what Corisin might be doing to alleviate acute exacerbations. Many of the

things displayed in this figure are not explicitly covered in the manuscript while the topics of the manuscript, such as the proteolytic degradation of bacterial proteins, are not. This figure would be well suited for a review on fibrosis / acute exacerbations, but I would like to see a more mechanistic depiction of how Corisin is exacerbating fibrosis here.

Response

We have modified the previous Figure 9 (now **Figure 10a,b**) to show the effect of corisin and the way corisin is cleaved by the protease to increase in the intraalveolar space to induce acute exacerbation of the disease.

Please see, Figure 10a, b in the revised version of the manuscript.

REVIEWERS' COMMENTS

Reviewer #1 (Remarks to the Author):

The authors responses to my "big picture" comments are very satisfactory. I have nothing further to add.

Reviewer #2 (Remarks to the Author):

The authors addressed my comments sufficiently. I recommend that this manuscript be approved for publication by the journal.

Reviewer #3 (Remarks to the Author):

I am writing in response to the revised manuscript submitted by D'Alessandro-Gabazza et al. in regard to their original manuscript detailing the effects of the pro-apoptotic peptide corisin and its effects on lung epithelial cells and acute exacerbations of pulmonary fibrosis. First, I would like to commend the Authors on their thorough, and thoughtful revision. Based on the complexity and length of the original manuscript, the number of comments were extensive. However, I feel the authors have done a more than adequate job in addressing my initial concerns with their findings.

I am pleased that the authors incorporated nearly all of my suggestions from the initial review including looking more deeply into the effects of corisin on TLR signaling and the physiologic concentrations at which corisin is bioactive. Further, the authors added a very nice rework of the model slide (Fig. 10) that presents a much more mechanistic picture of how corisin is modulating its apoptotic effects.

I do find it somewhat entertaining that the Authors state in their rebuttal, in response to my comment that supplemental data needed to be trimmed, that they had deleted some figures to streamline the manuscript when in fact the supplemental figure count has grown from 31 in the original manuscript to 36 figures in the current revision. While I personally find this amount of supplemental data unwieldy and hard to digest, I understand that the authors had to include much of the revised data as supplement due to space restrictions in the main manuscript.

Reviewer #4 (Remarks to the Author):

The authors have performed MD simulations to explore the conformational space of the peptide corisin in solution in absence of binders.

They have performed a substantial sampling and have extracted states probabilities using a Markov State Model to calculate a weighted free energy landscape projected on two collective variables. RMSD between conformers and distances between two residues that have a short-range interaction in the folded state have been used for this plotting. The simulations parameters and conditions used in the MD protocol are correct in general. The authors could have completed the analysis calculating transition state probabilities and performing flux analysis to get a more complete characterisation of the conformational states and kinetics in solution, but as I will clarify later, this detailed picture is not needed here.

I am unclear on what is the 'active' (folded) state of the peptide. The authors mention modelling the folded and unfolded states, is there any structure available of the folded state? It should be better clarified how they modelled the folded 'active' conformation

There are also reservations on the choice of RMSD as a metric in selecting conformations similar or dissimilar from native ones, as this parameter is not too sensitive to conformational changes for flexible peptides, complicating the assessment of the sampling of low free energy regions far from the native structure. One could use as alternative DRMSD, the differences between atomic distances within structures, or a contact map matrix, or better (if one is sure about the active state conformation) the fraction of native contacts preserved in the sampled conformers.

But all these criticisms are, in fact, details as I think these simulations do not tell us anything about the concentration range to be used in the experiment, because the authors simulate the peptide alone in solution without any binder present. Impact of re-equilibration on the experimentally observed binding event could be claimed only if one would be able to observe very slow transitions between peptide conformers that would be a bottleneck to re-equilibration after the 'active' fraction has bound and is therefore depleted in the ensemble of 'free' conformers. However, it is extremely unlikely that the comparatively (to experimental binding studies) very short simulation times would sample such conformational states. These are kinetically difficult to be accessed (by definition) and therefore not likely to be visited by in-silico sampling.

I suggest that the authors remove the MD simulations from the paper unless they can demonstrate that the isolated peptide story is useful to support the paper independently from the concentration story.

RESPONSE TO REVIEWERS' QUERIES

Reviewer #1 :

Comment

The authors responses to my "big picture" comments are very satisfactory. I have nothing further to add.

Reviewer #2:

Comment

The authors addressed my comments sufficiently. I recommend that this manuscript be approved for publication by the journal.

Reviewer #3:

Comment

I am writing in response to the revised manuscript submitted by D'Alessandro-Gabazza et al. in regard to their original manuscript detailing the effects of the pro-apoptotic peptide corisin and its effects on lung epithelial cells and acute exacerbations of pulmonary fibrosis. First, I would like to commend the Authors on their thorough, and thoughtful revision. Based on the complexity and length of the original manuscript, the number of comments were extensive. However, I feel the authors have done a more than adequate job in addressing my initial concerns with their findings.

I am pleased that the authors incorporated nearly all of my suggestions from the initial review including looking more deeply into the effects of corisin on TLR signaling and the physiologic concentrations at which corisin is bioactive. Further, the authors added a very nice rework of the model slide (Fig. 10) that presents a much more mechanistic picture of how corisin is modulating its apoptotic effects.

I do find it somewhat entertaining that the Authors state in their rebuttal, in response to my comment that supplemental data needed to be trimmed, that they had deleted some figures to streamline the manuscript when in fact the supplemental figure count has grown from 31 in the original manuscript to 36 figures in the current revision. While I personally find this amount of supplemental data unwieldy and hard to digest, I understand that the authors had to include much of the revised data as supplement due to space restrictions in the main manuscript.

Response to Reviewer #1, Reviewer #2 and Reviewer #3

We very much appreciate the favorable response from the Reviewers on the revised (R1) version of the manuscript. Their comments have helped us to significantly improve our work and its presentation.

Reviewer #4:

Comments

The authors have performed MD simulations to explore the conformational space of the peptide corisin in solution in absence of binders.

They have performed a substantial sampling and have extracted states probabilities using a Markov State Model to calculate a weighted free energy landscape projected on two collective variables. RMSD between conformers and distances between two residues that have a short-range interaction in the folded state have been used for this plotting. The simulations parameters and conditions used in the MD protocol are correct in general. The authors could have completed the analysis calculating transition state probabilities and performing flux analysis to get a more complete characterisation of the conformational states and kinetics in solution, but as I will clarify later, this detailed picture is not needed here.

I am unclear on what is the 'active' (folded) state of the peptide. The authors mention modelling the folded and unfolded states, is there any structure available of the folded state? It should be better clarified how they modelled the folded 'active' conformation

There are also reservations on the choice of RMSD as a metric in selecting conformations similar or dissimilar from native ones, as this parameter is not too sensitive to conformational changes for flexible peptides, complicating the assessment of the sampling of low free energy regions far from the native structure. One could use as alternative DRMSD, the differences between atomic distances within structures, or a contact map matrix, or better (if one is sure about the active state conformation) the fraction of native contacts preserved in the sampled conformers.

But all these criticisms are, in fact, details as I think these simulations do not tell us anything about the concentration range to be used in the experiment, because the authors simulate the peptide alone in solution without any binder present. Impact of re-equilibration on the experimentally observed binding event could be claimed only if one would be able to observe very slow transitions between peptide conformers that would be a bottleneck to re-equilibration after the 'active' fraction has bound and is therefore depleted in the ensemble of 'free' conformers. However, it is extremely unlikely that the comparatively (to experimental binding studies) very short simulation

times would sample such conformational states. These are kinetically difficult to be accessed (by definition) and therefore not likely to be visited by in-silico sampling.

I suggest that the authors remove the MD simulations from the paper unless they can demonstrate that the isolated peptide story is useful to support the paper independently from the concentration story.

Response

We thank very much the insight and detailed comments of the Reviewer on the MD simulations described in the previous version of the manuscript.

Following the suggestion of the Reviewer, we have removed the MD simulations from the paper. Instead, we added in the results and discussion sections a potential explanation for the high concentration of synthetic corisin required to show activity in the *in vitro* and *in vivo* experiments.

Please see page 15, lines 370 to 373 (pdf file of the manuscript), and pages 25 and 26, lines 622 to 634 in the re-revised manuscript.